# SAMoSSA: Multivariate Singular Spectrum Analysis with Stochastic Autoregressive Noise

**Abdullah Alomar**
MIT
aalomar@mit.edu

**Munther Dahleh**
MIT
dahleh@mit.edu

**Sean Mann**
MIT
seanmann@mit.edu

**Devavrat Shah**
MIT
devavrat@mit.edu

## Abstract

The well-established practice of time series analysis involves estimating deterministic, non-stationary *trend* and *seasonality* components followed by learning the residual stochastic, stationary components. Recently, it has been shown that one can learn the deterministic non-stationary components accurately using multivariate Singular Spectrum Analysis (mSSA) in the absence of a correlated stationary component; meanwhile, in the absence of deterministic non-stationary components, the Autoregressive (AR) stationary component can also be learnt readily, e.g. via Ordinary Least Squares (OLS). However, a theoretical underpinning of multi-stage learning algorithms involving both deterministic and stationary components has been absent in the literature despite its pervasiveness. We resolve this open question by establishing desirable theoretical guarantees for a natural two-stage algorithm, where mSSA is first applied to estimate the non-stationary components despite the presence of a correlated stationary AR component, which is subsequently learned from the residual time series. We provide a finite-sample forecasting consistency bound for the proposed algorithm, SAMoSSA, which is data-driven and thus requires minimal parameter tuning. To establish theoretical guarantees, we overcome three hurdles: (i) we characterize the spectra of Page matrices of stable AR processes, thus extending the analysis of mSSA; (ii) we extend the analysis of AR process identification in the presence of arbitrary bounded perturbations; (iii) we characterize the out-of-sample or forecasting error, as opposed to solely considering model identification. Through representative empirical studies, we validate the superior performance of SAMoSSA compared to existing baselines. Notably, SAMoSSA's ability to account for AR noise structure yields improvements ranging from 5% to 37% across various benchmark datasets.

## 1 Introduction

**Background.** Multivariate time series have often been modeled as mixtures of stationary stochastic processes (e.g. AR process) and deterministic non-stationary components (e.g. polynomial and harmonics). To handle such mixtures, classical time series forecasting algorithms first attempt to estimate and then remove the non-stationary components. For example, before fitting an Autoregressive Moving-average (ARMA) model, polynomial trend and seasonal components must be estimated and then removed from the time series. Once the non-stationary components have been eliminated[1], the ARMA model is learned. This estimation procedure often relies on domain knowledge and/or fine-tuning, and theoretical analysis for such multiple-stage algorithms is limited in the literature.

Prior work presented a solution for estimating non-stationary deterministic components without domain knowledge or fine-tuning using mSSA [3]. This framework systematically models a wide

---

[1]Note that Autoregressive Integrated Moving-average (ARIMA) and Seasonal ARIMA models use differencing in conjunction with unit root tests to remove *stochastic* non-stationary components, and are not suited for the model we consider.

37th Conference on Neural Information Processing Systems (NeurIPS 2023).

class of (deterministic) non-stationary multivariate time series as linear recurrent formulae (LRF), encompassing a wide class of spatio-temporal factor models that includes harmonics, exponentials, and polynomials. However, mSSA, both algorithmically and theoretically, does not handle additive correlated stationary noise, an important noise structure in time series analysis. Indeed, all theoretical results of mSSA are established in the noiseless setting [16] or under the assumption of independent and identically distributed (i.i.d.) noise [4, 3].

On the other hand, every stable stationary process can be approximated as a finite order AR process [8, 26]. The classical OLS procedure has been shown to accurately learn finite order AR processes [17]. However, the feasibility of identifying AR processes in the presence of a non-stationary deterministic component has not been addressed.

In summary, despite the pervasive practice of first estimating non-stationary deterministic components and then learning the stationary residual component, neither an elegant unified algorithm nor associated theoretical analyses have been put forward in the literature.

A step towards resolving these challenges is to answer the following questions: (i) Can mSSA consistently estimate non-stationary deterministic components in the presence of correlated stationary noise? (ii) Can the AR model be accurately identified using the residual time series, after removing the non-stationary deterministic components estimated by mSSA, potentially with errors? (iii) Can the out-of-sample or forecasting error of such a multi-stage algorithm be analyzed?

In this paper, we resolve all three questions in the affirmative: we present SAMoSSA, a two-stage procedure, where in the first stage, we apply mSSA on the observations to extract the non-stationary components; and in the second stage, the stationary AR process is learned using the residuals.

**Setup.** We consider the discrete time setting where we observe a multivariate time series $Y(t) :=$ $[y_1(t), \ldots, y_N(t)] \in \mathbb{R}^N$ at each time index $t \in [T] := \{1, \ldots, T\}$ where $T \geq N^2$. For each $n \in [N]$, and each timestep $t \in [T]$, the observations take the form

$$y_n(t) = f_n(t) + x_n(t), \tag{1}$$

where $f_n : \mathbb{Z}^+ \to \mathbb{R}$ denotes the non-stationary deterministic component, and $x_n(t)$ is a stationary AR noise process of order $p_n$ ($\text{AR}(p_n)$). Specifically, each $x_n(t)$ is governed by

$$x_n(t) = \sum_{i=1}^{p_n} \alpha_{ni} x_n(t-i) + \eta_n(t), \tag{2}$$

where $\eta_n(t)$ refers to the per-step noise, modeled as mean-zero i.i.d. random variables, and $\alpha_{ni} \in \mathbb{R} \ \forall \ i \in [p_n]$ are the parameters of the $n$-th $\text{AR}(p_n)$ process.

**Goal.** For each $n \in [N]$, our objective is threefold. The first is estimating $f_n(t)$ from the noisy observations $y_n(t)$ for all $t \in [T]$. The second is identifying the AR process' parameters $\alpha_{ni} \ \forall \ i \in [p_n]$. The third is out-of-sample forecasting of $y_n(t)$ for $t > T$.

**Contributions.** The main contributions of this work is SAMoSSA, an elegant two-stage algorithm, which manages to learn both non-stationary deterministic and stationary stochastic components of the underlying time series. A detailed summary of the contributions is as follows.

*(a) Estimating non-stationary component with mSSA under AR noise.* The prior theoretical results for mSSA are established in the deterministic setting [16] or under the assumption of i.i.d. noise [4, 3]. In Theorem 4.1, we establish that the mean squared estimation error scales as $\sim 1/\sqrt{NT}$ under AR noise – the same rate was achieved by [3] under i.i.d. noise. Key to this result is establishing spectral properties of the "Page" matrix of AR processes, which may be of interest in its own right (see Lemma A.2).

*(b) Estimating AR model parameters under bounded perturbation.* We bound the estimation error for the OLS estimator of AR model parameters under *arbitrary and bounded* observation noise, which could be of independent interest. Our results build upon the recent work of [17] which derives similar results but *without* any observation noise. In that sense, ours can be viewed as a robust generalization of [17].

---

[2]if $T < N$, we divide the $N$ time series into $\lceil N/T \rceil$ sets of time series where this condition will hold.

*(c) Out-of-sample finite-sample consistency of SAMoSSA.* We provide a finite-sample analysis for the forecasting error of SAMoSSA– such analysis of two-stage procedures in this setup is nascent despite its ubiquitous use in practice. Particularly, we establish in Theorem 4.4 that the out-of-sample forecasting error for SAMoSSA for the $T$ time-steps ahead scales as $\sim \frac{1}{T} + \frac{1}{\sqrt{NT}}$ with high probability.

*(d) Empirical results.* We demonstrate superior performance of SAMoSSA using both real-world and synthetic datasets. We show that accounting for the AR noise structure, as implemented in SAMoSSA, consistently enhances the forecasting performance compared to the baseline presented in [3]. Specifically, these enhancements range from 5% to 37% across benchmark datasets.

**Related Work.** Time series analysis is a well-developed field. We focus on two pertinent topics.

*mSSA.* Singular Spectrum Analysis (SSA), and its multivariate extension mSSA, are well-studied methods for time series analysis which have been used heavily in a wide array of problems including imputation [4, 3], forecasting [18, 2], and change point detection [22, 6]. Refer to [16, 15] for a good overview of the SSA literature. The classical SSA method consists of the following steps: (1) construct a Hankel matrix of the time series of interest; (2) apply Singular Value Decomposition (SVD) on the constructed Hankel matrix, (3) group the singular triplets to separate the different components of the time series; (4) learn a linear model for each component to forecast. mSSA is an extension of SSA which handles multiple time series simultaneously and attempts to exploit the shared structure between them [18]. The only difference between mSSA and SSA is in the first step, where Hankel matrices of individual series are "stacked" together to create a single stacked Hankel matrix. Despite its empirical success, mSSA's classical analysis has mostly focused on identifying which time series have a low-rank Hankel representation, and defining sufficient *asymptotic* conditions for signal extraction, i.e., when the various time series components are separable. All of the classical analysis mostly focus on the deterministic case, where no observation noise is present.

Recently, a variant of mSSA was introduced for the tasks of forecasting and imputation [3]. This variant, which we extend in this paper, uses the Page matrix representation instead of the Hankel matrix, which enables the authors to establish finite-sample bounds on the imputation and forecasting error. However, their work assumes the observation noise to be i.i.d., and does not accommodate correlated noise structure, which is often assumed in the time series literature [26]. Our work extends the analysis in Agarwal et al. [3] to observations under AR noise structure. We also extend the analysis of the forecasting error by studying how well we can learn (and forecast) the AR noise process with perturbed observations.

*Estimating AR parameters.* AR processes are ubiquitous and of interest in many fields, including time series analysis, control theory, and machine learning. In these fields, it is often the goal to estimate the parameters of an AR process from a sample trajectory. Estimation is often carried out through OLS, which is asymptotically optimal [11]. The asymptotic analysis of the OLS estimator is well established, and recently, given the newly developed statistical tools of high dimensional probability, cf. [31, 29], various recent works have tackled its finite-time analysis as well. For example, several works established results for the finite-time identification for general first order vector AR systems [20, 19, 27, 24, 21]. Further, González et al. [17] provides a finite-time bound on the deviation of the OLS estimate for general $\mathrm{AR}(p)$ processes. To accommodate our setting, we extend the results of [17] in two ways. First, we extend the analysis to accommodate any sub-gaussian noise instead of assuming gaussianity; Second, and more importantly, we extend it to handle arbitrary bounded observation errors in the sampled trajectory, which can be of independent interest.

## 2 Model

**Deterministic Non-Stationary Component.** We adopt the spatio-temporal factor model of [3, 6] described next. The spatio-temporal factor model holds when two assumptions are satisfied: the first assumption concerns the spatial structure, i.e., the structure across the $N$ time series $f_1, \ldots, f_N$; the second assumption pertains to the "temporal" structure. Before we describe the assumptions, we first define a key time series representation: the Page matrix.

**Definition 2.1** (Page Matrix). *Given a time series $f : \mathbb{Z}^+ \to \mathbb{R}$, and an initial time index $t_0 > 0$, the Page matrix representation over the $T$ entries $f(t_0), \ldots, f(t_0 + T - 1)$ with parameter $1 \leq L \leq T$ is given by the matrix $\mathbf{Z}(f, L, T, t_0) \in \mathbb{R}^{L \times \lfloor T/L \rfloor}$ with $\mathbf{Z}(f, L, T, t_0)_{ij} = f(t_0 + i - 1 + (j - 1) \times L)$ for $i \in [L]$, $j \in [\lfloor T/L \rfloor]$.*

In words, to construct the $L \times \lfloor T/L \rfloor$ Page matrix of the entries $f(t_0), \ldots, f(t_0 + T - 1)$, partition them into $\lfloor T/L \rfloor$ segments of $L$ contiguous entries and concatenate these segments column-wise (see Figure 1). Now we introduce the main assumptions for $f_1, \ldots, f_N$.

**Assumption 2.1** (Spatial structure). *For each* $n \in [N]$, $f_n(t) = \sum_{r=1}^{R} u_{nr} w_r(t)$ *for some* $u_{nr} \in \mathbb{R}$ *and* $w_r : \mathbb{Z}^+ \to \mathbb{R}$.

Assumption 2.1 posits that each time series $f_n(\cdot) \ \forall n \in [N]$ can be described as a linear combination of $R$ "fundamental" time series. The second assumption relates to the temporal structure of the fundamental time series $w_r(\cdot) \ \forall r \in [R]$, which we describe next.

**Assumption 2.2** (Temporal structure). *For each* $r \in [R]$ *and for any* $T > 1$, $1 \le L \le T, t_0 > 0$, $rank(\mathbf{Z}(w_r, L, T, t_0)) \le G$.

Assumption 2.2 posits that the Page matrix of each "fundamental" time series $w_r$ has finite rank. While this imposed temporal structure may seem restrictive at first, it has been shown that many standard functions that model time series dynamics satisfy this property [4, 3]. These include any finite sum of products of harmonics, low-degree polynomials, and exponential functions (refer to Proposition 2.1 in [3]).

**Stochastic Stationary Component.** We adopt the following two assumptions about the AR processes $x_n \ \forall n \in [N]$. We first assume that these AR processes are stationary (see Definition H.4 in Appendix H).

**Assumption 2.3** (Stationarity and distinct roots). $x_n(t)$ *is a stationary* $\mathrm{AR}(p_n)$ *process* $\forall n \in [N]$. *That is, let* $\lambda_{ni} \in \mathbb{C}$, $i \in [p_n]$ *denote the roots of* $g_n(z) := z^{p_n} - \sum_{i=1}^{p_n} \alpha_{ni} z^{p_n-i}$. *Then,* $|\lambda_{ni}| < 1$ $\forall i \in [p_n]$. *Further,* $\forall n \in [N]$, *the roots of* $g_n(z)$ *are distinct.*

Further, we assume the per-step noise $\eta_n(t)$ are i.i.d. sub-gaussian random variables (see Definition H.1 in Appendix H).

**Assumption 2.4** (Sub-gaussian noise). *For* $n \in [N], t \in [T]$, $\eta_n(t)$ *are zero-mean i.i.d. sub-gaussian random variables with variance* $\sigma^2$.

**Model Implications.** In this section, we state two important implications of the model stated above. The first is that the stacked Page matrix defined as

$$\mathbf{Z}_f(L, T, t_0) = [\mathbf{Z}(f_1, L, T, t_0) \quad \mathbf{Z}(f_2, L, T, t_0) \quad \ldots \quad \mathbf{Z}(f_n, L, T, t_0)],$$

is low-rank. Precisely, we recall the following Proposition stated in [3].

**Proposition 2.1** (Proposition 2 in [3]). *Let Assumptions 2.1 and 2.2 hold. Then for any* $L \le \lfloor \sqrt{T} \rfloor$ *with any* $T \ge 1$, $t_0 > 0$, *the rank of the Page matrix* $\mathbf{Z}(f_n, L, T, t_0)$ *for* $n \in [N]$ *is at most* $R \times G$. *Further, the rank of the stacked Page matrix* $\mathbf{Z}_f(L, T, t_0)$ *is* $k \le R \times G$.

Throughout, we will use the shorthand $\mathbf{Z}_f := \mathbf{Z}_f(L, T, 1)$ and $\mathbf{Z}_f(t_0) := \mathbf{Z}_f(L, T, t_0)$ [3]. The second implication is that there exists a linear relationship between the last row of $\mathbf{Z}_f$ and its top $L - 1$ rows. The following proposition establishes this relationship. First, Let $[\mathbf{Z}_f]_{L\cdot}$ denote the $L$-th row of $\mathbf{Z}_f$ and let $\mathbf{Z}'_f \in \mathbb{R}^{(L-1)\times(N\lfloor T/L \rfloor)}$ denote the sub-matrix that consist of the top $L - 1$ rows of $\mathbf{Z}_f$.

**Proposition 2.2** (Proposition 3 in [3]). *Let Assumptions 2.1 and 2.2 hold. Then, for* $L > RG$, *there exists* $\beta^* \in \mathbb{R}^{L-1}$ *such that* $[\mathbf{Z}_f]_{L\cdot}^\top = \mathbf{Z}'_f{}^\top \beta^*$. *Further,* $\|\beta^*\|_0 \le RG$.

## 3 Algorithm

The proposed algorithm provides two main functionalities. The first one is *decomposing* the observations $y_n(t)$ into an estimate of the non-stationary and stationary components for $t \le T$. The second is forecasting $y_n(t)$ for $t > T$, which involves learning a forecasting model for both $f_n(t)$ and $x_n(t)$.

**Univariate Case.** For ease of exposition, we will first describe the algorithm for the univariate case ($N = 1$). The algorithm has the following parameters: $1 < L \le \sqrt{T}$, and $1 \le \hat{k} \le L$ (refer

---

[3]We will use the same shorthand for the stacked page matrices of $y$ and $x$: namely, $\mathbf{Z}_y$ and $\mathbf{Z}_x$.

to Appendix B.2 for how to choose these parameters). For clarity, we will assume, without loss of generality, that $L$ is chosen such that $T/L$ is an integer[4]. In the first step of the algorithm, we transform the observations $y_1(t), t \in [T]$ into the $L \times T/L$ Page matrix $\mathbf{Z}(y_1, L, T, 1)$. We will use the shorthand $\mathbf{Z}_{y_1} \coloneqq \mathbf{Z}(y_1, L, T, 1)$ henceforth.

*Decomposition.* We compute the SVD $\mathbf{Z}_{y_1} = \sum_{\ell=1}^{L} s_\ell u_\ell v_\ell^\top$, where $s_1 \geq s_2 \cdots \geq s_L \geq 0$ denote its ordered singular values, and $u_\ell \in \mathbb{R}^L, v_\ell \in \mathbb{R}^{T/L}$ denote its left and right singular vectors, respectively, for $\ell \in [L]$. Then, we obtain $\widehat{\mathbf{Z}}_{f_1}$ by retaining the top $\hat{k}$ singular components of $\mathbf{Z}_{y_1}$ (i.e., by applying Hard Singular Value Thresholding (HSVT) with threshold $\hat{k}$). That is, $\widehat{\mathbf{Z}}_{f_1} = \sum_{\ell=1}^{\hat{k}} s_\ell u_\ell v_\ell^\top$.

We denote by $\widehat{f}_1(t)$ and $\widehat{x}_1(t)$ the *estimates* of $f_1(t)$ and $x_1(t)$ respectively. We read off the estimates $\widehat{f}_1(t)$ directly from the matrix $\widehat{\mathbf{Z}}_{f_1}$ using the entry that corresponds to $t \in [T]$. More precisely, for $t \in [T]$, $\widehat{f}_1(t)$ equals the entry of $\widehat{\mathbf{Z}}_{f_1}$ in row $(t - 1 \mod L) + 1$ and column $\lceil t/L \rceil$, while $\widehat{x}_1(t) = y_1(t) - \widehat{f}_1(t)$.

*Forecasting.* To forecast $y_1(t)$ for $t > T$, we produce a forecast for both $\widehat{f}_1(t)$ and $\widehat{x}_1(t)$. Both forecasts are performed through linear models ($\widehat{\beta}$ for $\widehat{f}_1(t)$ and $\widehat{\alpha}_1$ for $\widehat{x}_1(t)$.) For $\widehat{f}_1(t)$, we first learn a linear model $\widehat{\beta}$ defined as

$$\widehat{\beta} = \operatorname{argmin}_{\beta \in \mathbb{R}^{L-1}} \sum_{m=1}^{T/L} (y_1(Lm) - \beta^\top \widehat{F}_m)^2, \tag{3}$$

where $\widehat{F}_m = [\widehat{f}_1(L(m-1)+1), \dots, \widehat{f}_1(L \times m - 1)]$ for $m \in [T/L]$ [5] We then use $\widehat{\beta}$ and the $L - 1$ lagged observations to produce $\widehat{f}_1(t)$. That is $\widehat{f}_1(t) = \widehat{\beta}^\top Y_1(t-1)$, where $Y_1(t-1) = [y_1(t-1), \dots, y_1(t-L)]$.

For $\widehat{x}_1(t)$, we first estimate the parameters for the $\text{AR}(p_1)$ process. Specifically, define the $p_1$-dimensional vectors $\widehat{X}_1(t) \coloneqq [\widehat{x}_1(t), \dots, \widehat{x}_1(t - p_1 + 1)]$ [6]. Then, define the OLS estimate as,

$$\widehat{\alpha}_1 = \operatorname{argmin}_{\alpha \in \mathbb{R}^{p_1}} \sum_{t=p_1}^{T-1} (\widehat{x}_1(t+1) - \alpha^\top \widehat{X}_1(t)) \tag{4}$$

Then, let $\widetilde{x}_1(t') = y_1(t') - \widehat{f}_1(t')$ for any $t' < t$, [7]. We then use $\widehat{\alpha}_1$ and the $p_1$ lagged entries of $\widetilde{x}_1(\cdot)$ to produce $\widehat{x}_1(t)$. That is $\widehat{x}_1(t) = \widehat{\alpha}_1^\top \widetilde{X}_1(t-1)$, where $\widetilde{X}_1(t-1) = [\widetilde{x}_1(t-1), \dots, \widetilde{x}_1(t - p_1)]$. Finally, produce the forecast $\widehat{y}_1(t) = \widehat{f}_1(t) + \widehat{x}_1(t)$.

**Multivariate Case.** The key change for the case $N > 1$ is the use of the stacked Page matrix $\mathbf{Z}_y$, which is the column-wise concatenation of the Page matrices induced by individual time series. Specifically, consider the stacked Page matrix $\mathbf{Z}_y \in \mathbb{R}^{L \times NT/L}$ defined as

$$\mathbf{Z}_y = [\mathbf{Z}(y_1, L, T, 1) \quad \mathbf{Z}(y_2, L, T, 1) \quad \dots \quad \mathbf{Z}(y_n, L, T, 1)]. \tag{5}$$

*Decomposition.* The procedure for learning the non-stationary component $f_1, \dots, f_N$ for $t \leq T$ is similar to that of the univariate case. Specifically, we perform HSVT with threshold $\hat{k}$ on $\mathbf{Z}_y$ to produce $\widehat{\mathbf{Z}}_y$. Then, we read off the *estimate* of $f_n(t)$ from $\widehat{\mathbf{Z}}_y$. Specifically, for $t \in [T]$, and $n \in [N]$, let $\widehat{\mathbf{Z}}_{f_n}$ refer to sub-matrix of $\widehat{\mathbf{Z}}_f$ induced by selecting only its $[(n-1) \times (T/L) + 1, \dots, n \times T/L]$

---

[4]Otherwise, one can apply this algorithm to the two ranges $\{1, \dots, L \times \lfloor T/L \rfloor\}$ and $\{(T \mod L) + 1, \dots, T\}$.

[5]In the forecasting algorithm, the estimates $[\widehat{f}_1(L(m-1)+1), \dots, \widehat{f}_1(L \times m - 1)]$ are obtained by applying HSVT on a sub-matrix of $\mathbf{Z}_{y_1}$ which consists of its first $L - 1$ rows. This is done to establish the theoretical results as it helps us avoid dependencies in the noise between $y_1(Lm)$ and $\widehat{F}_m$ for $m \in [T/L]$.

[6]Note that we assume knowledge of the true parameter $p_1$ (and $p_n$ in the multivariate case).

[7]Note the subtle difference between $\widehat{x}_1(t)$ and $\widetilde{x}_1(t)$ – precisely, for $t > T$, $\widehat{x}_1(t)$ is an estimate of $x_1(t)$ before observing $y_1(t)$ (i.e., a forecast), whereas $\widetilde{x}_1(t)$ is an estimate of $x_1(t)$ after observing $y_1(t)$.

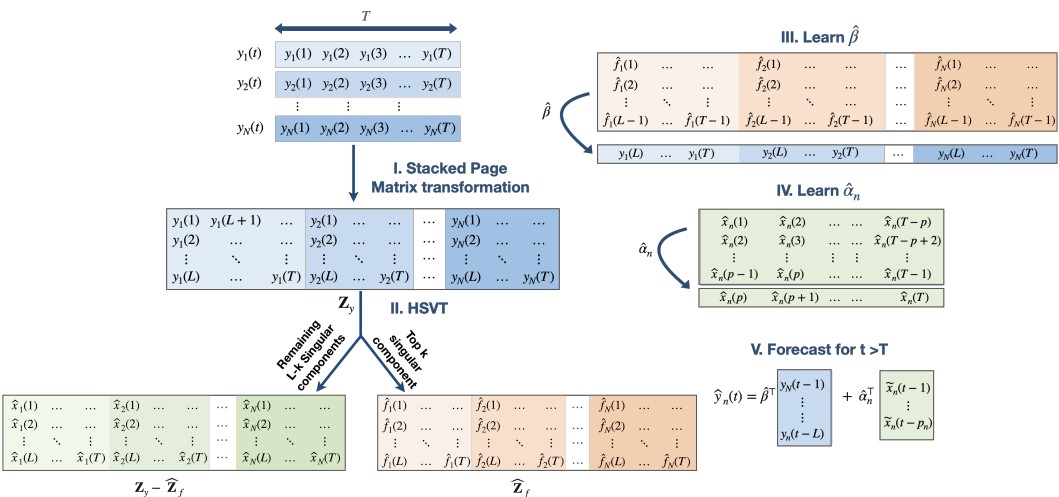

Figure 1: A visual depiction of SAMoSSA. The algorithm five steps are (i) transform the time series into its stacked Page matrix representation; (ii) decompose time series into non-stationary and stationary components (iii) estimate $\widehat{\beta}$; (iv) estimates $\widehat{\alpha}_n \ \forall n \in [N]$; (v) produce the forecast $\widehat{y}_n(t)$ for $t > T$.

columns. Then for $t \in [T]$, $\widehat{f}_n(t)$ equals the entry of $\widehat{\mathbf{Z}}_{f_n}$ in row $(t - 1 \mod L) + 1$ and column $\lceil t/L \rceil$. We then produce an *estimate* for $x_n(t)$ as $\widehat{x}_n(t) = y_n(t) - \widehat{f}_n(t)$.

*Forecasting.* To forecast $y_n(t)$ for $t > T$, as in the univariate case, we learn a linear model $\widehat{\beta}$ defined as

$$\widehat{\beta} = \text{argmin}_{\beta \in \mathbb{R}^{L-1}} \quad \sum_{m=1}^{N \times T/L} (y_m - \beta^\top \widehat{F}_m)^2, \tag{6}$$

where $y_m$ is the $m$-th component of $[y_1(L), \ y_1(2 \times L), \ldots, y_1(T), \ y_2(L), \ldots, y_2(T), \ldots, y_N(T)] \in \mathbb{R}^{N \times T/L}$, and $\widehat{F}_m \in \mathbb{R}^{L-1}$ corresponds to the vector formed by the entries of the first $L - 1$ rows in the $m$th column of $\widehat{\mathbf{Z}}_f$ [8] for $m \in [N \times T/L]$. We then use $\widehat{\beta}$ to produce $\widehat{f}_n(t) = \widehat{\beta}^\top Y_n(t-1)$, where again $Y_n(t-1)$ is the vector of the $L-1$ lags of $y_n(t-1)$. That is $Y_n(t-1) = [y_n(t-1)) \ldots y_n(t-L)]$.

Then, for each $n \in [N]$, we estimate $\alpha_{ni} \ \forall i \in [p_n]$, the parameters for the $n$-th $\text{AR}(p_n)$ process. Let $\widehat{\alpha}_n$ denote the OLS estimate defined as

$$\widehat{\alpha}_n = \text{argmin}_{\alpha \in \mathbb{R}^{p_n}} \quad \sum_{t=p_n}^{T-1} (\widehat{x}_n(t+1) - \alpha^\top \widehat{X}_n(t))^2. \tag{7}$$

Then, produce a forecast for $x_n$ as $\widehat{x}_n(t) = \widehat{\alpha}_n^\top \widetilde{X}_n(t-1)$, where again $\widetilde{X}_n(t-1) = [\widetilde{x}_n(t-1), \ldots, \widetilde{x}_n(t-p_n)]$. Finally, produce the forecast for $y_n$ as $\widehat{y}_n(t) = \widehat{f}_n(t) + \widehat{x}_n(t)$. For a visual depiction of the algorithm, refer to Figure 1.

## 4 Results

In this section, we provide finite-sample high probability bounds on the following quantities:

**1. Estimation error of non-stationary component .** First, we give an upper bound for the estimation error of each one of $f_1(t), \ldots, f_N(t)$ for $t \in [T]$. Specifically, we upper bound the following metric

$$\text{EstErr}(N, T, n) = \frac{1}{T} \sum_{t=1}^{T} (\widehat{f}_n(t) - f_n(t))^2, \tag{8}$$

---

[8]To be precise, $\widehat{\mathbf{Z}}_f$ here is the truncated SVD of a sub-matrix of $\mathbf{Z}_y$ which consist of its first $L - 1$ rows.

where $\widehat{f}_n(t)$ $n \in [N], t \in [T]$ are the estimates produced by the algorithm we proposed in Section 3.

**2. Identification of AR parameters.** Second, we analyze the accuracy of the estimates $\widehat{\alpha}_n \ \forall n \in [N]$ produced by the proposed algorithm. Specifically, we upper bound the following metrics

$$\|\widehat{\alpha}_n - \alpha_n\|_2 \qquad \forall n \in [N], \tag{9}$$

where $\alpha_n = [\alpha_{n1}, \ldots, \alpha_{np_n}]$.

**3. Out-of-sample forecasting error.** Finally, we provide an upper bound for the forecasting error of $y_1(t), \ldots, y_N(t)$ for $t \in \{T+1, \ldots, 2T\}$. Specifically, we upper bound the following metric

$$\mathsf{ForErr}(N,T) = \frac{1}{NT} \sum_{n=1}^{N} \sum_{t=T+1}^{2T} \left( \widehat{y}_n(t) - \mathbb{E}\left[ y_n(t) \mid y_n(t-1), \ldots, y_n(1) \right] \right)^2, \tag{10}$$

where $\widehat{y}_n(\cdot) \ \forall n \in [N], t \in \{T+1, \ldots, 2T\}$ are the forecasts produced by the algorithm we propose in Section 3. Before stating the main results, we state key additional assumptions.

**Assumption 4.1** (Balanced spectra). *Let $\mathbf{Z}_f(t_0)$ denote the $L \times NT/L$ stacked Page matrix associated with all $N$ time series $f_1(\cdot), \ldots, f_N(\cdot)$ for their $T$ consecutive entries starting from $t_0$. Let $k = rank(\mathbf{Z}_f(t_0))$, and let $\sigma_k(\mathbf{Z}_f(t_0))$ denote the $k$-th singular value for $\mathbf{Z}_f(t_0)$. Then, for any $t_0 > 0$, $\mathbf{Z}_f(t_0)$ is such that $\sigma_k(\mathbf{Z}_f(t_0)) \geq \gamma \sqrt{NT}/\sqrt{k}$ for some absolute constant $\gamma > 0$.*

This assumption holds whenever the the non-zero singular values are "well-balanced", a standard assumption in the matrix/tensor estimation literature [3, 6]. Note that this assumption, as stated, ensures balanced spectra for the stacked Page matrix of *any* set of $T$ consecutive entries of $f_1(\cdot), \ldots, f_N(\cdot)$.

Finally, we will impose an additional necessary restriction on the complexity of the $N$ time series $f_1(t), \ldots, f_N(t)$ for $t > T$ (as is done in [3, 5]). Let $\mathbf{Z}'_f$ denote the $(L-1) \times (NT/L)$ matrix formed using the top $L-1$ rows of $\mathbf{Z}_f$. Further, for any $t_0 \in [T+1]$, let $\mathbf{Z}'_f(t_0)$ denote the $(L-1) \times (NT/L)$ matrix formed using the top $L-1$ rows of $\mathbf{Z}_f(t_0)$. For any matrix $M$, let colspan$(M)$ denote the subspace spanned by the its columns. We assume the following property.

**Assumption 4.2** (Subspace inclusion). *For any $t_0 \in [T+1]$, $colspan(\mathbf{Z}'_f(t_0)) \subseteq colspan(\mathbf{Z}'_f)$.*

This assumption is necessary as it requires the stacked Page matrix of the out-of-sample time series $\mathbf{Z}'_f(t_0)$ to be only as "rich" as that of the stacked Page matrix of the "in-sample" time series $\mathbf{Z}'_f$.

## 4.1 Main Results

First, recall that $R$ is defined in Assumption 2.1, $G$ in Assumption 2.2, $k$ is in Proposition 2.1, while $\gamma$ is defined in Assumption 4.1. Further, recall that $\lambda_{ni}$ for $i \in [p_n]$ are the roots of the characteristic polynomial of the $n$-th AR process, as defined in Assumption 2.3. Throughout, let $c$ and $C$ be absolute constants, $f_{\max} := \max_{n, t \leq 2T} |f_n(t)|$, $p := \max_n p_n$, $\alpha_{\max} = \max_n \|\alpha_n\|_2$, and $C(f_{\max}, \gamma)$ denote a constant that depends only (polynomially) on model parameters $f_{\max}$ and $\gamma$. Last but not least, define the key quantity

$$\sigma_x := \frac{c_\lambda \sigma}{(1 - \lambda_\star)}$$

where $\lambda_\star = \max_{i,n} |\lambda_{ni}|$ and $c_\lambda$ is a constant that depends only on $\lambda_{ni}$[9]. Note that $\sigma_x^2$ is a key quantity that we use to bound important spectral properties of AR processes, as Lemma A.2 establishes.

### 4.1.1 Estimation Error of Non-Stationary Component

**Theorem 4.1** (Estimation error of $f$). *Assume access to the observations $y_1(t), \cdots, y_N(t)$ for $t \in [T]$ as defined in (1). Let assumptions 2.1, 2.2, 2.3, 2.4 and 4.1 hold. Let $L = \sqrt{NT}$ and $\hat{k} = k$, then, for any $n \in [N]$, with probability of at least $1 - \frac{c}{(NT)^{10}}$*

$$\mathsf{EstErr}(N,T,n) \leq C(f_{\max}, \gamma) \left( \frac{\sigma_x^4 G R \log(NT)}{\sqrt{NT}} \right). \tag{11}$$

---

[9]See Appendix A for an explicit expression of $c_\lambda$.

This theorem implies that the mean squared error of estimating $f$ scales as $\tilde{O}\left(\frac{1}{\sqrt{NT}}\right)$[10] with high probability. The proof of Theorem 4.1 is in Appendix C.

### 4.1.2 Identification of AR Processes

**Theorem 4.2** (AR Identification). *Let the conditions of Theorem 4.1 hold. Let $\widehat{\alpha}_n$ be as defined in (7), and $\alpha_n = [\alpha_{n1}, \dots, \alpha_{np_n}]$ as defined in (2). Then, for a sufficiently large $T$ such that $\frac{\log(T)}{\sqrt{T}} < \frac{C\sigma^2}{p\sigma_x^2\alpha_{\max}^2}\left(\frac{\lambda_{\min}(\Psi)}{\lambda_{\max}(\Gamma)}\right)$, and for any $n \in [N]$ where $\mathsf{EstErr}(N, T, n) \leq \frac{\sigma^2\lambda_{\min}(\Psi)}{6p}$, we have with probability of at least $1 - \frac{c}{T^{10}}$,*

$$\|\widehat{\alpha}_n - \alpha_n\|_2^2 \leq \frac{Cp}{\lambda_{\min}(\Psi)}\left(\frac{p}{T}\log\left(\frac{T\lambda_{\max}(\Psi)}{\lambda_{\min}(\Psi)}\right) + \frac{\sigma_x^2\mathsf{EstErr}(N, T, n)\log(T)}{\sigma^2\min\{1, \sigma^2\lambda_{\min}(\Psi)\}}\right). \tag{12}$$

Recall that $\widehat{\alpha}_n$ is estimated using $\widehat{x}_n(\cdot)$, a perturbed version of the true AR process $x_n(\cdot)$. This perturbation comes from the estimation error of $x_n(\cdot)$, which is a consequence of $\mathsf{EstErr}(N, T, n)$. Hence, it is not surprising that $\mathsf{EstErr}(N, T, n)$ shows up in the upper bound. Indeed, the upper bound we provide here consists of two terms: the first, which scales as $\tilde{O}\left(\frac{1}{T}\right)$ is the bound one would get with access to the true AR process $x_n(\cdot)$, as shown in [17]. The second term characterizes the contribution of the perturbation caused by the estimation error, and it scales as $O\left(\mathsf{EstErr}(N, T, n)\right)$, which we show is $\tilde{O}\left(\frac{1}{\sqrt{NT}}\right)$ with high probability in Theorem 4.1. Also, note that the theorem holds when the estimation error is sufficiently small ($\mathsf{EstErr}(N, T, n) \leq \frac{\sigma^2\lambda_{\min}(\Psi)}{6p}$). This condition ensures that the perturbation arising from the estimation error remains below the lower bound for the minimum eigenvalue of the sample covariance matrix associated with the autoregressive processes. Note that $\lambda_{\max}(\Psi), \lambda_{\min}(\Psi)$ and $\lambda_{\max}(\Gamma)$ are quantities that relate to the parameters of the $\mathrm{AR}(p)$ processes and their "controllability Gramian" as we detail in Appendix A.1. The proof of Theorem 4.2 is in Appendix D.

### 4.1.3 Out-of-sample Forecasting Error

We first establish an upper bound on the error of our estimate $\widehat{\beta}$.

**Theorem 4.3** (Model Identification ($\beta^*$)). *Let the conditions of Theorem 4.1 hold. Let $\widehat{\beta}$ be defined as in (6), and $\beta^*$ as defined in Proposition 2.2. Then, with probability of at least $1 - \frac{c}{(NT)^{10}}$*

$$\|\widehat{\beta} - \beta^*\|_2^2 \leq C(f_{\max}, \gamma)\left(\frac{\sigma_x^4 G^2 R^2 \log(NT)}{\sqrt{NT}}\right)\max\{\|\beta^*\|_1^2, 1\}. \tag{13}$$

This shows that the error of estimating $\beta^*$ (in squared euclidean norm) scales as $\tilde{O}\left(\frac{1}{\sqrt{NT}}\right)$.

**Theorem 4.4.** *Let the conditions of Theorem 4.2 and Assumption 4.2 hold. Then, with probability of at least $1 - \frac{c}{T^{10}}$*

$$\mathsf{ForErr}(N, T) \leq \tilde{C}G^3R^3p^2\sigma_x^6\left(\frac{p\sigma^2\log(T)}{T} + \frac{GR\sigma_x^6}{\min\{\sigma^2, \sigma^4\}}\frac{\log(NT)^2}{\sqrt{NT}}\right),$$

*where $c$ is an absolute constant, and $\tilde{C}$ denotes a constant that depends only (polynomially) on $f_{\max}, \gamma, \lambda_{\min}(\Psi), \lambda_{\max}(\Psi), \beta^*$ and $\alpha_{\max}$.*

This theorem establishes that the forecasting error for the next $T$ time steps scales as $\tilde{O}\left(\frac{1}{T} + \frac{1}{\sqrt{NT}}\right)$ with high probability. Note that the $\tilde{O}\left(\frac{1}{T}\right)$ term is a function of $T$ only, as it is a consequence of the error incurred when we estimate each $\alpha_n$. Recall that we estimate $\alpha_n$ separately for $n \in [N]$ using the available (perturbed) $T$ observation of each process. The $\tilde{O}\left(\frac{1}{\sqrt{NT}}\right)$ term on the other hand is a consequence of the error incurred when learning and forecasting $f_1, \dots, f_n$, which is done collectively across the $N$ time series. Finally, Theorem 4.4 implies that when $N = \Theta(T)$, the forecasting error scales as $\tilde{O}\left(\frac{1}{T}\right)$. The proof of Theorems 4.3 and 4.4 are in Appendix F and G, respectively.

---

[10]The $\tilde{O}(\cdot)$ notation is analogous to the standard $O(\cdot)$ while ignoring log dependencies.

Table 1: Performance of algorithms on various datasets, measured by mean $R^2$.

|  | Traffic | Electricity | Exchange | Synthetic |
|---|---|---|---|---|
| SAMoSSA | 0.776 | **0.829** | 0.731 | **0.476** |
| mSSA | 0.747 | 0.605 | 0.674 | 0.366 |
| ARIMA | 0.723 | <-10 | **0.756** | 0.305 |
| Prophet | 0.462 | 0.197 | <-10 | -0.445 |
| DeepAR | **0.824** | 0.764 | 0.579 | 0.323 |
| LSTM | 0.821 | -1.261 | -1.825 | 0.381 |

## 5 Experiments

In this section, we support our theoretical results through several experiments using synthetic and real-world data. In particular, we draw the following conclusions:

1. Our results align with numerical simulations concerning the estimation of non-stationary components under AR stationary noise and the accuracy of AR parameter estimation (Section 5.1).
2. Modeling and learning the autoregressive process in SAMoSSA led to a consistent improvement over mSSA. The improvements range from 5% to 37% across standard datasets (Section 5.2).

### 5.1 Model Estimation

**Setup.** We generate a synthetic multivariate time series ($N = 10$) that is a mixture of both harmonics and an AR noise process. The AR process is stationary, and its parameter is chosen such that $\lambda^* = \max_{n,i} |\lambda_{ni}|$ is one of three values $\{0.3, 0.6, 0.95\}$. Refer to Appendix B.1 for more details about the generating process. We then evaluate the estimation error $\mathsf{EstErr}(N, T, 1)$ and the AR parameter estimation error $\|\alpha_1 - \widehat{\alpha}_1\|_2$ as we increase $T$ from 200 to 500000.

**Results.** Figure 2a visualizes the mean squared estimation error of $f_1$, while Figure 2b shows the AR parameter estimation error. The solid lines in both figures indicate the mean across ten trials, whereas the shaded areas cover the minimum and maximum error across the ten trials. We find that, as the theory suggests, the estimation error for both the non-stationary component and the AR parameter decay to zero as $NT$ increases. We also see that the estimation error is inversely proportional to $(1 - \lambda^*)$, which is being reflected in the AR parameter estimation error as well.

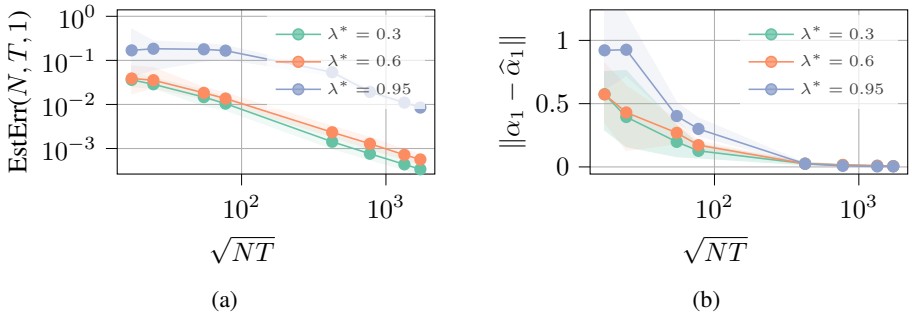

      (a)                                    (b)

Figure 2: The error in SAMoSSA's estimation of the non-stationary components and the AR parameters decays to zero as $NT$ increases as the theorem suggests.

### 5.2 Forecasting

We showcase SAMoSSA's forecasting performance relative to standard algorithms. Notably, we compare SAMoSSA to the mSSA variant in [3] to highlight the value of learning the autoregressive process.

**Setup.** The forecasting ability of SAMoSSA was compared against (i) mSSA [3], (ii) ARIMA, a very popular classical time series prediction algorithm, (iii) Prophet [28], (iv) DeepAR [23] and (v) LSTM [14]on three real-life datasets and one synthetic dataset containing both deterministic trend/seasonality and a stationary noise process (see details in Appendix B). Each dataset was split into train, validation, and test sets (see Appendix B.1). Each dataset has multiple time series, so we

aggregate the performance of each algorithm using the mean $R^2$ score. We use the $R^2$ score since it is invariant to scaling and gives a reasonable baseline: a negative value indicates performance inferior to simply predicting the mean.

**Results.** In Table 1 we report the mean $R^2$ for each method on each dataset. We highlight that, across all datasets, SAMoSSA consistently performs the best or is otherwise very competitive with the best. The results also underscore the significance of modeling and learning the autoregressive process in real-world datasets. Notably, learning the autoregressive model in SAMoSSA consistently led to an improvement over mSSA, with the increase in $R^2$ values ranging from 5% to 37%. We note that this improvement is due to the fact that mSSA, as described in [3], overlooks any potential structure in the stochastic processes $x_1(\cdot), \ldots, x_N(\cdot)$ and assumes i.i.d.mean-zero noise process. While in SAMoSSA, we attempt to capture the structure of $x_1(\cdot), \ldots, x_N(\cdot)$ through the learned AR process.

# 6  Discussion and Limitations

We presented SAMoSSA, a two-stage procedure that effectively handles mixtures of deterministic non-stationary and stationary AR processes with minimal model assumptions. We analyze SAMoSSA's ability to estimate non-stationary components under stationary AR noise, the error rate of AR system identification via OLS under observation errors, and a finite-sample forecast error analysis.

We note that our results can be readily adapted to accommodate (i) *approximate low-rank* settings (as in the model by Agarwal et al [3]); and (ii) scenarios with incomplete data. We do not discuss these settings to focus on our core contributions, but they represent valuable directions for future studies.

Our analysis reveals some limitations, providing avenues for future research. One limitation of our model is that it only considers stationary stochastic processes. Consequently, processes with non-stationary stochastic trends, such as a random walk, are not incorporated. Investigating the inclusion of such models and their interplay with the SSA literature is a worthy direction for future work. Second, our model assume non-interaction between the $N$ stationary processes $x_1, \ldots, x_N$. Yet, it might be plausible to posit the existence of interactions among them, possibly through a vector AR model (VAR). Examining this setting represents another compelling direction for future work.

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

# A  Concentration inequalities for AR processes

In this section, we present two important lemmas that are central to the main results. Before stating the two lemmas, we start with key important quantities of AR Processes.

## A.1  Controllability Gramian of AR Processes

In this section, we explain the notation of key quantities used in the main results. First, note that the process in (2) can be written in matrix-vector form by defining a transition matrix and input vector

$$\boldsymbol{A}_n = \begin{bmatrix} \alpha_{n1:n(p_n-1)}^\top & \alpha_{np_n} \\ I_{p_n-1} & 0_{p_n-1} \end{bmatrix}, \qquad B_n = \begin{bmatrix} 1 \\ 0_{p_n-1} \end{bmatrix},$$

where $0_{p_n-1}$ is the column vector of $p_n - 1$ zeroes. Then, with $X_n(t) := [x_n(t), \dots, x_n(t - p_n + 1)]$,

$$X_n(t) = \boldsymbol{A}_n X_n(t-1) + B_n \eta_{nt}.$$

With this setup, we define two important matrices,

$$\Gamma_n = \sum_{t=0}^\infty \boldsymbol{A}_n^t (\boldsymbol{A}_n^\top)^t, \qquad \Psi_n = \sum_{t=0}^\infty \boldsymbol{A}_n^t B_n B_n^\top (\boldsymbol{A}_n^\top)^t, \tag{14}$$

where $\Psi_n$ is known as the *controllability Gramian* of the $n$-th $\mathrm{AR}(p_n)$ process $x_n$. For a positive semi-definite matrix $\boldsymbol{M}$, let $\lambda_{\max}(\boldsymbol{M}) \geq 0$ and $\lambda_{\min}(\boldsymbol{M}) \geq 0$ be its maximum and minimum eigenvalues, respectively. We define $\lambda_{\max}(\Psi) := \max_n \lambda_{\max}(\Psi_n)$ and $\lambda_{\min}(\Psi) := \min_n \lambda_{\min}(\Psi_n)$. We also define $\lambda_{\max}(\Gamma)$ and $\lambda_{\min}(\Gamma)$ analogously for $\Gamma$.

## A.2  Key Lemmas

Now, we present two important lemmas that are central to the main results. The first lemma states that a vector of consecutive entries of a stationary autoregressive process is a sub-gaussian vector with variance proxy $\sigma_x^2$.

**Lemma A.1.** *Let assumptions 2.3 and 2.4 hold, then, for any $n \in [N], t \geq 1, K \geq 1$, the vector $[x_n(t), \dots, x_n(t + K)]$ is a sub-gaussian vector with variance proxy $\sigma_x^2$.*

This lemma, along with the well known bound on the norm of sub-gaussian vectors, is key to the bounds presented in Section 4.1. Further, this Lemma, along with an $\varepsilon$-net argument, allows us to bound the operator norm of the stacked Page matrix of the AR processes $x_1(t), \dots, x_N(t)$. Specifically, we establish next that its operator norm grows as $O(\sqrt{NT/L})$ with high probability.

**Lemma A.2.** *For any $T > 1$, $1 \leq L \leq \sqrt{NT}$, and $t_0 > 0$, let*

$$\mathbf{Z}_x(t_0) = [\mathbf{Z}(x_1, L, T, t_0) \quad \mathbf{Z}(x_2, L, T, t_0) \quad \dots \quad \mathbf{Z}(x_n, L, T, t_0)]$$

*be the $L \times NT/L$ stacked Page matrix of the AR processes $x_1(t), \dots, x_N(t)$, as defined in (2). Let Assumptions 2.3 and 2.4 hold. Then, each row $[\mathbf{Z}_x(t_0)]_i$ is a sub-gaussian vector with variance proxy $\sigma_x^2$. Further, With probability $1 - \exp(-cNT/L)$,*

$$\|\mathbf{Z}_x(t_0)\|_2 \leq 2c\sigma_x \sqrt{NT/L}.$$

Next, we prove Lemma A.1 and Lemma A.2. Before stating the proofs, we start with the important helper Lemma A.3.

**Lemma A.3** (MA($\infty$) representation of an $\mathrm{AR}(p)$ process)**.** *Let $x(t)$ be a stationary $\mathrm{AR}(p)$ process defined as*

$$x(t) = \sum_{i=1}^p \alpha_i x(t-i) + \eta(t).$$

*Assume that the roots $(\lambda_1, \dots, \lambda_p)$ of its characteristic polynomial $g(z) := z^p - \sum_{i=1}^p \alpha_i z^{p-i}$ are distinct. Then, $x(t)$ has the MA($\infty$) representation*

$$x(t) = \sum_{k=0}^\infty \beta_k \eta(t-k),$$

*with $\beta_k := \sum_{i=1}^p a_i \lambda_i^k$, $a_i := \prod_{1 \leq j \leq p, \ j \neq i} \left(1 - \frac{\lambda_j}{\lambda_i}\right)^{-1}$.*

*Proof.* First, note that that we can write $x(t)$ in lag operator form as

$$f(L)\, x(t) \triangleq \left(1 - \sum_{i=1}^{p} \alpha_i L^i\right) x(t) = \eta(t). \qquad (15)$$

Here, $L$ is a lag operator such that $Lx(t) = x(t-1)$. Note that since we have assumed the AR process to be stationary, all the roots of $g(z)$ lie inside the unit circle [26]. Using the roots of $g(z)$, (15) can be written as,

$$\prod_{i=1}^{p} (1 - \lambda_i L)\, x(t) = \eta(t).$$

Further, using the partial fraction decomposition we can write the process as

$$x_t = \frac{1}{\prod_{i=1}^{p}(1 - \lambda_i L)} \eta(t)$$

$$= \sum_{i=1}^{p} \frac{a_i}{1 - \lambda_i L} \eta(t)$$

where,

$$a_i = \prod_{1 \le j \le p,\ j \ne i} \left(1 - \frac{\lambda_j}{\lambda_i}\right)^{-1}$$

Recall that we can expand each fraction as

$$\frac{a_i}{1 - \lambda_i L} = a_i \left(\sum_{k=0}^{\infty} \lambda_i^k L^k\right).$$

Thus, finally, we can write $x(t)$ as,

$$x(t) = \sum_{k=0}^{\infty} \beta_k \eta(t-k),$$

with $\beta_k := \sum_{i=1}^{p} a_i \lambda_i^k$. $\qquad \square$

## A.3 Proof of Lemma A.1

*Proof.* First, let $X_n(t, K) = [x_n(t), \ldots, x_n(T+K)]$ be the vector of interest. Let $E_n(t, K) = [\eta_n(t), \ldots, \eta_n(T+K)]$, then using Lemma A.3, we can write $X_n(t, K)$ as

$$X_n(t, K) = \sum_{k=0}^{\infty} \beta_{nk} E_n(t-k, K)$$

Note that $\beta_{nk} E_n(t-k, K)$ is clearly a sub-gaussian vector with variance proxy $\beta_{nk}^2 \sigma^2$. Further note that the sum of two sub-gaussian random vectors, not necessarily independent, is also a sub-gaussian random vector. To see this, suppose $\mathbf{x}$ and $\mathbf{y}$ are both sub-gaussian vectors with variance proxies $\sigma_x^2$ and $\sigma_y^2$. Then for any $\mathbf{v} \in \mathcal{S}^K$,

$$\mathbf{v}^T(\mathbf{x} + \mathbf{y}) = \mathbf{v}^T \mathbf{x} + \mathbf{v}^T \mathbf{y}$$
$$\sim \text{subG}\left((\sigma_x + \sigma_y)^2\right).$$

Thus, $X_n(t, K)$, for any $n$, is also a sub-gaussian random vector with variance proxy

$$\left(\sum_{k=0}^{\infty} |\beta_{nk}| \sigma\right)^2 \le \sigma^2 \left(\sum_{k=0}^{\infty} \max_n |\beta_{nk}|\right)^2.$$

Now we are interested in bounding $\sum_{k=0}^{\infty} \max_n |\beta_{nk}|$. To do so, first recall that $\beta_{nk} := \sum_{i=1}^{p_n} a_{ni} \lambda_{ni}^k$ where $\lambda_{ni} \in \mathbb{C}$, $i \in [p_n]$ denote the roots of $g_n(z)$ and $a_i = \prod_{1 \leq j \leq p_n, \ j \neq i} \left( 1 - \frac{\lambda_{nj}}{\lambda_{ni}} \right)^{-1}$. Let $\lambda_\star = \max_{i,n} |\lambda_{ni}|$ and $c_{\lambda_n} = \sum_{i=1}^{p_n} \left| \prod_{1 \leq j \leq p, \ j \neq i} \left( 1 - \frac{\lambda_{nj}}{\lambda_{ni}} \right)^{-1} \right|$. Then, we can bound $\beta_{nk}$ as

$$|\beta_{nk}| \leq \lambda_\star^k \sum_{i=1}^{p_n} |a_i|$$
$$\leq c_{\lambda_n} \lambda_\star^k.$$

Thus, with $c_\lambda = \max_n c_{\lambda_n}$

$$\sum_{k=0}^{\infty} \max_n |\beta_{nk}| \leq \frac{c_\lambda}{1 - \lambda_\star}. \tag{16}$$

Hence, $X_n(t, K)$ is a sub-gaussian random vector with variance proxy $\sigma_x^2 := \frac{c_\lambda^2 \sigma^2}{(1 - \lambda_\star)^2}$. $\qquad \square$

## A.4 Proof of Lemma A.2

*Proof.* Let $q = NT/L$. We will find an upper bound on the operator norm by following two steps: (i) we will represent each $x_n(t) \ \forall n \in [N]$ as an MA($\infty$) process; then, (ii) we will bound the operator norm of the stacked Page matrix of this MA($\infty$) processes.

**Step 1: MA($\infty$) representation for AR($p$).**

As a direct application of Lemma A.3, we can write $x_n(t)$ as,

$$x_n(t) = \sum_{k=0}^{\infty} \beta_{nk} \eta_n(t - k),$$

with $\beta_{nk} := \sum_{i=1}^{p} a_{ni} \lambda_{ni}^k$, $a_{ni} := \prod_{1 \leq j \leq p, \ j \neq i} \left( 1 - \frac{\lambda_{nj}}{\lambda_{ni}} \right)^{-1}$

**Step 2: Bound the Page matrix for the MA($\infty$) Page matrix.**

First, let's consider, without loss of generality, the case when $t_0 = 1$. That is, let's consider $\mathbf{Z}_x := \mathbf{Z}_x(1)$. To get the desired bound, we will first obtain a high probability bound on the Euclidean norm of $\mathbf{Z}_x^\top \mathbf{v}$ for any vector $\mathbf{v} \in \mathcal{S}^{L-1} := \{\mathbf{v} \in \mathbb{R}^L : \|\mathbf{v}\|_2 = 1\}$. Then, we will bound the operator norm using this and an $\varepsilon$-net argument.

Note that by step (1), we have

$$x_n(t) = \sum_{k=0}^{\infty} \beta_{nk} \eta_n(t - k).$$

Now define $\boldsymbol{E}_i$ as the analogous stacked Page matrix for $\eta_1(t), \ldots, \eta_N(t)$ for $t \in \{i, \ldots, i + T - 1\}$. Observe that $\boldsymbol{E}_i$ are simply identically distributed (but not independent) matrices of i.i.d. sub-gaussian random variables. Then we can write

$$\mathbf{Z}_x = \sum_{k=0}^{\infty} \boldsymbol{E}_{1-k} \boldsymbol{B}_k, \tag{17}$$

where $\boldsymbol{B}_k \in \mathbb{R}^{q \times q}$ is a diagonal matrix with the $i$-th diagonal $[\boldsymbol{B}_k]_{ii} = \beta_{\lceil \frac{iN}{q} \rceil k}$. Then

$$\left\| \mathbf{Z}_x^\top \mathbf{v} \right\|_2 = \left\| \sum_{k=0}^{\infty} \boldsymbol{B}_k \boldsymbol{E}_{1-k}^\top \mathbf{v} \right\|_2.$$

It is easy to verify that $\boldsymbol{E}_i^\top \mathbf{v}$ is a *sub-gaussian random vector* (see Definition H.3) with variance proxy $\sigma^2$, because for any $\mathbf{u} \in \mathcal{S}^{q-1}$ we have (using $\boldsymbol{E}$ as a shorthand for $\boldsymbol{E}_i$)

$$\mathbf{v}^T \boldsymbol{E} \mathbf{u} = \sum_{i=1}^{L} \sum_{j=1}^{q} v_i u_j E_{ij}$$

$$\sim \mathrm{subG}\left( \sum_{i=1}^{L} \sum_{j=1}^{q} v_i^2 u_j^2 \sigma^2 \right)$$

$$\sim \mathrm{subG}(\sigma^2).$$

Using the above, $\boldsymbol{B}_k \boldsymbol{E}_i^\top \mathbf{v}$ is $\sim \mathrm{subG}(\max_n |\beta_{nk}|^2 \sigma^2)$. Since $\mathbf{Z}_x \mathbf{v}$ is a sum of sub-gaussian random vectors, it is also a sub-gaussian random vector with variance proxy $\sigma_x^2$ as we showed in the proof of Lemma A.1.

Using Lemma H.2, we get, with probability at least $1 - \delta$,

$$\left\| \mathbf{Z}_x^\top \mathbf{v} \right\|_2 \le 4\sigma_x \sqrt{q} + 2\sigma_x \sqrt{\log\left(\frac{1}{\delta}\right)}$$

Equivalently, for $t \ge 0$, it holds that

$$\mathbb{P}\left( \left\| \mathbf{Z}_x \mathbf{v} \right\|_2 - 4\sigma_x \sqrt{q} > t \right) \le \exp\left( -\frac{t^2}{4\sigma_x^2} \right).$$

Now we have shown that for any *fixed* $\mathbf{v} \in \mathcal{S}^{L-1}$, the quantity $\left\| \mathbf{Z}_x \mathbf{v} \right\|_2$ grows as $O\left(\sqrt{q}\right)$ with high probability. What remains is to apply the union bound over an $\varepsilon$-net to extend this to the *maximum* over $\mathcal{S}^{L-1}$, i.e. the operator norm of $\mathbf{Z}_x$.

Let $\Sigma$ be a maximal $1/2$-net of $\mathcal{S}^{L-1}$, i.e. a set of points in $\mathcal{S}^{L-1}$ spaced at least $1/2$ from each other such that no other point can be added without violating this property. Now consider $\mathbf{v}_\star$ such that $\left\| \mathbf{Z}_x^\top \mathbf{v}_\star \right\|_2 = \left\| \mathbf{Z}_x \right\|_2$. Since $\Sigma$ is maximal, there is some $\mathbf{v} \in \Sigma$ within $1/2$ of $\mathbf{v}_\star$. Since

$$\left\| \mathbf{Z}_x^\top (\mathbf{v} - \mathbf{v}_\star) \right\|_2 \le \left\| \mathbf{Z}_x \right\|_2 \left\| \mathbf{v} - \mathbf{v}_\star \right\|_2 \le \left\| \mathbf{Z}_x \right\|_2 / 2,$$

and

$$\left\| \mathbf{Z}_x^\top (\mathbf{v} - \mathbf{v}_\star) \right\|_2 \ge \left\| \mathbf{Z}_x^\top \mathbf{v}_\star \right\|_2 - \left\| \mathbf{Z}_x^\top \mathbf{v} \right\|_2 = \left\| \mathbf{Z}_x \right\|_2 - \left\| \mathbf{Z}_x^\top \mathbf{v} \right\|_2,$$

we have that

$$\left\| \mathbf{Z}_x^\top \mathbf{v} \right\|_2 \ge \left\| \mathbf{Z}_x \right\|_2 / 2,$$

i.e. taking the maximum over $\Sigma$ is equivalent to doing so over $\mathcal{S}^{L-1}$ up to a factor of two. Using the fact that $|\Sigma| \le C^L$ for some absolute constant $C$, the fact that $L < q$, and using a sufficiently large absolute constant $c > 0$ we have

$$\mathbb{P}\left( \left\| \mathbf{Z}_x \right\|_2 > 2c\sigma_x \sqrt{q} \right) \le \sum_{\mathbf{v} \in \Sigma} \mathbb{P}\left( \left\| \mathbf{Z}_x \mathbf{v} \right\|_2 > c\sigma_x \sqrt{q} \right)$$

$$\le 2C^L \exp\left( -c'q \right)$$

$$= \exp\left( -c''q \right).$$

where $c'' > 0$ and $c' > 0$ are absolute constants. Recalling that $q = NT/L$ concludes the proof for the operator norm.

Finally, we show that each row $[\mathbf{Z}_x(t_0)]_{i\cdot}$ in the stacked page matrix is a sub-gaussian vector with variance proxy $\sigma_x^2$. Consider, without loss of generality, the first row $[\mathbf{Z}_x]_{1\cdot}$. Using (17), we have for any $\mathbf{v} \in \mathcal{S}^{L-1}$

$$\mathbf{v}^\top [\mathbf{Z}_x]_{1\cdot} = \sum_{k=0}^{\infty} \mathbf{v}^\top [\boldsymbol{E}_{1-k} \boldsymbol{B}_k]_{1\cdot},$$

note that $\mathbf{v}^\top [\boldsymbol{E}_{1-k} \boldsymbol{B}_k]_{1\cdot}$ is $\sim \mathrm{subG}(\max_n \beta_{nk}^2 \sigma^2)$, and hence, using the same argument for the sum of dependent sub-gaussian random variables used in the proof above, we have $\mathbf{v}^\top [\mathbf{Z}_x]_{1\cdot} \sim \mathrm{subG}(\sigma_x^2)$. $\qquad\square$

## B   Experiment details

In Appendix B.1, we detail the datasets used. In Appendix B.2, we explain the implementations and hyperparameter choices for our method and benchmark algorithms. In Appendix B.3, we describe our hyperparameter tuning strategy.

### B.1   Datasets

In the model estimation experiments, we generate a synthetic multivariate time series ($N = 10$) that is a mixture of both harmonics and an $\mathrm{AR}(2)$ process. The harmonics are generated as follow: we first generate $R = 3$ fundamental time series of the form $g_k(t) = \sin(\omega_k t + \phi_k)$ for $k \in [R]$, where $\omega_k$ and $\phi_k$ are uniformly randomly sampled on the ranges $[2\pi/100, 2\pi/50]$, $[0, 2\pi]$ respectively. We then sample the $N = 10$ series using $f_k(t) = \sum_{i=1}^{R} c_{ki} g_k(t)$ for $k \in [N]$, where the mixture coefficients $c_{ki}$ are independent standard normal random variables. The AR process is stationary, and its parameter is chosen such that $\lambda^* = \max_{n,i} |\lambda_{ni}|$ is set to one of $\{0.3, 0.6, 0.95\}$. The noise of the process has a variance $\sigma^2 = 0.2$.

In the forecasting experiments, we evaluate our method and benchmark algorithms on three real-world datasets and one synthetic dataset. The preprocessing steps and setup of each dataset are described below.

**Traffic Dataset.** This public dataset obtained from the UCI repository shows the occupancy rate of traffic lanes in San Francisco. The data is sampled every 15 minutes but to be consistent with previous work, we aggregate the data into hourly data and use the first 10248 time-points for training, the next 48 points for validation, and another 48 points for testing in the forecasting experiments. Specifically, in our testing period, we do 1-hour ahead forecasts for the next 48 hours.

**Electricity Dataset.** This is a public dataset obtained from the UCI repository which shows the 15-minutes electricity load of 370 households. We aggregate the data into hourly intervals and use the first 25824 time-points for training, the next 48 points for validation, and another 48 points for testing in the forecasting experiments. Specifically, in our testing period, we do 1-hour ahead forecasts for the next 48 hours.

**Exchange Dataset.** This is a dataset containing the daily exchange rates of eight foreign currencies, including those of Australia, the UK, Canada, Switzerland, China, Japan, New Zealand and Singapore, between 1990 and 2016. We standardize each time series to have zero mean and unit variance since the range of typical exchange rates varies greatly across the currencies. We use the first 7528 time-points for training, the next 30 for validation, and the final 30 for testing. All forecasts are made 1-day ahead.

**Synthetic Dataset.** We generate $R$ fundamental time series of the form $g_k(t) = \sin(\omega_k t + \phi_k) + m_k t$ for $k \in [R]$, where $\omega_k$, $\phi_k$, and $m_k$ are uniformly randomly sampled on the ranges $[2\pi/100, 2\pi/10]$, $[0, 2\pi]$, and $[-5 \times 10^{-4}, 5 \times 10^{-4}]$ respectively. We then sample $N$ mixtures of the form $f_k(t) = \sum_{i=1}^{R} c_{ki} g_k(t)$ for $k \in [N]$, where the mixture coefficients $c_{ki}$ are independent standard normals. We then inject $\mathrm{AR}(1)$ autoregressive noise to each series to produce $y_k(t) = f_k(t) + x_k(t)$ for $k \in [N]$, where $\{x_k(t)\}_{k \in [N]}$ are independent processes, each with a fixed autoregressive coefficient $\alpha = -0.5$ and noise variance $\sigma^2 = 1$. We fix the coefficient to ensure that each time series has significantly autocorrelated noise that can be exploited to improve predictions.

Table 2: Dataset and training/validation/test split details.

| Dataset | No. time series | Training period | Validation period | Test period |
|---|---|---|---|---|
| Traffic | 963 | 1 to 10248 | 10249 to 10296 | 10296 to 10344 |
| Electricity | 370 | 1 to 25824 | 25825 to 25872 | 25872 to 25919 |
| Exchange | 8 | 1 to 7528 | 7529 to 7558 | 7559 to 7588 |
| Synthetic | 25 | 1 to 10000 | 10001 to 10025 | 10026 to 10050 |

### B.2 Algorithms

In this section, we discuss the implementations used and hyperparameter tuning details for each algorithm that was evaluated.

**SAMoSSA & mSSA.** Since mSSA is a special case of SAMoSSA where the second stage – fitting autoregressive models – is skipped, we use one in-house implementation to evaluate both algorithms. The relevant hyperparameters are as follows:

1. *The number of retained singular values, $k$.* This hyperparameter is relevant for both algorithms. We select $k$ using one of three methods: (i) the data-driven procedure suggested in [13] which computes a threshold based on the shape and median singular value of the page matrix; (ii) picking $k$ as the minimum number of singular values required to capture $90\%$ of the spectral energy of the page matrix; (iii) picking a fixed low rank, specifically $k = 5$.

2. *The shape parameter of the stacked Page matrix.* This is the ratio between the number of columns and rows of the stacked Page matrix used to carry out the matrix estimation procedure. While our theoretical analysis fixes this value to 1 (recall that we set $L = \sqrt{NT}$), in practice we have found that a slightly wider Page matrix can improve performance. This parameter is chosen from $\{1, 3, 5\}$.

3. *The number of autoregressive lag coefficients, $p$.* This hyperparameter is only relevant for SAMoSSA. While each time series does not necessarily need to use the same value, we enforce this to be the case for computational efficiency. We choose $p \in \{0, 1, 2, 3\}$, i.e., the model is allowed to not fit a residual model if it brings no apparent benefit, corresponding to the case of $p = 0$.

**Prophet.** We use Prophet's Python library with the parameters selected using a grid search of the following parameters as suggested in [12]:

1. *Changepoint prior scale.* This parameter determines how much the trend changes at the detected trend changepoints. We choose this parameter from $\{0.001, 0.05, 0.2\}$.

2. *Seasonality prior scale.* This parameter controls the magnitude of the seasonality. We choose this parameter from $\{0.01, 10\}$.

3. *Seasonality Mode.* We choose between an "additive" and "multiplicative" seasonality term.

The parameters for each time series are chosen independently.

**ARIMA.** We used the ARIMA implementation of the Python library `statsmodels` [25]. The hyperparameters are grid searched independently for each time series, and are as follows:

1. *Autoregressive order.* This is the number of autoregressive lag coefficients fitted, chosen from $\{1, 2, 3\}$.

2. *Differencing order.* This denotes the number of times the time series is differenced before a model is fitted, and we choose between 0 (no differencing) and 1.

3. *Moving average order.* This is the number of moving average lag coefficients, chosen from $\{1, 2, 3\}$.

**LSTM.** We use Keras implementation [9]. We perform a grid search on the number of layers $\{2, 3, 4\}$.

**DeepAR.** We use the implementation provided by the GluonTS package [7]. We use the default parameters.

### B.3 Parameter Selection

We tune hyperparameters for all algorithms evaluated using rolling cross validation. During the validation step, each model is fitted only once on the training set; then, it repeatedly makes one-step-ahead forecasts and is afterwards provided with the realized value, in order to use the updated data to make the next forecast. During test time, the model is fitted on the training *and* validation sets, and similarly repeatedly makes one-step-ahead forecasts with the fitted parameters. For the models that choose one set of hyperparameters for all $N$ time series, the mean $R^2$ score over all time series on the validation set predictions is the metric of choice.

# C  Proof of Theorem 4.1

In this section, we provide the proof for Theorem 4.1. While the analysis generally follows the argument for the imputation error in [3], we adapt it for our different assumptions and metric of interest. **Metric**. First, recall that our metric of interest is

$$\mathsf{EstErr}(N, T, n) = \frac{1}{T} \sum_{t=1}^{T} (\widehat{f}_n(t) - f_n(t))^2. \tag{18}$$

Recall that $\widehat{\mathbf{Z}}_f$ is the stacked Page matrix of $\widehat{f}_1, \ldots, \widehat{f}_N$, and $\mathbf{Z}_f$ is the stacked Page matrix of $f_1, \ldots, f_N$. To bound this error, we first establish a deterministic bound through a general lemma for Hard Singular Value Thresholding (HSVT).

## C.1  Deterministic Bound

We state the following result, a more general version of which is stated in [3].

**Lemma C.1** ([3]). *For $k \geq 1$, let $\mathbf{Y} = \mathbf{M} + \mathbf{E} \in \mathbb{R}^{q \times p}$ with rank$(\mathbf{M}) = k$. Let $\mathbf{U}_k \mathbf{\Sigma}_k \mathbf{V}_k^\top$ and $\widehat{\mathbf{U}}_k \widehat{\mathbf{\Sigma}}_k \widehat{\mathbf{V}}_k^\top$ denote the top $k$ singular components of the SVD of $\mathbf{M}$ and $\mathbf{Y}$ respectively. Then, the HSVT estimate $\widehat{\mathbf{M}} = \widehat{\mathbf{U}}_k \widehat{\mathbf{\Sigma}}_k \widehat{\mathbf{V}}_k^\top$ is such that for all $j \in [q]$,*

$$\|\widehat{\mathbf{M}}_{j \cdot}^\top - \mathbf{M}_{j \cdot}^\top\|_2^2 \leq 2 \frac{\|\mathbf{E}\|_2^2}{\sigma_k(\mathbf{M})^2} \left( \left\| \mathbf{E}_{j \cdot}^\top \right\|_2^2 + \left\| \mathbf{M}_{j \cdot}^\top \right\|_2^2 \right) + 2 \left\| \mathbf{V}_k \mathbf{V}_k^\top \mathbf{E}_{j \cdot}^\top \right\|_2^2,$$

*Proof.* First, note that

$$\widehat{\mathbf{M}}_{j \cdot}^\top - \mathbf{M}_{j \cdot}^\top = \left( \widehat{\mathbf{V}}_k \widehat{\mathbf{V}}_k^\top \mathbf{Y}_{j \cdot}^\top - \widehat{\mathbf{V}}_k \widehat{\mathbf{V}}_k^\top \mathbf{M}_{j \cdot}^\top \right) + \left( \widehat{\mathbf{V}}_k \widehat{\mathbf{V}}_k^\top \mathbf{M}_{j \cdot}^\top - \mathbf{M}_{j \cdot}^\top \right),$$

where $\widehat{\mathbf{M}}_{j \cdot}^\top = \widehat{\mathbf{V}}_k \widehat{\mathbf{V}}_k^\top \mathbf{Y}_{j \cdot}^\top$ by definition. Note that the vector $\left( \widehat{\mathbf{V}}_k \widehat{\mathbf{V}}_k^\top \mathbf{M}_{j \cdot}^\top - \mathbf{M}_{j \cdot}^\top \right)$ is in the span of a subspace orthogonal to $\widehat{\mathbf{V}}_k \widehat{\mathbf{V}}_k^\top$, and hence we have by the Pythagorean theorem,

$$\left\| \widehat{\mathbf{M}}_{j \cdot}^\top - \mathbf{M}_{j \cdot}^\top \right\|_2^2 = \left\| \widehat{\mathbf{V}}_k \widehat{\mathbf{V}}_k^\top \mathbf{Y}_{j \cdot}^\top - \widehat{\mathbf{V}}_k \widehat{\mathbf{V}}_k^\top \mathbf{M}_{j \cdot}^\top \right\|_2^2 + \left\| \widehat{\mathbf{V}}_k \widehat{\mathbf{V}}_k^\top \mathbf{M}_{j \cdot}^\top - \mathbf{M}_{j \cdot}^\top \right\|_2^2$$

$$= \left\| \widehat{\mathbf{V}}_k \widehat{\mathbf{V}}_k^\top \mathbf{E}_{j \cdot}^\top \right\|_2^2 + \left\| \widehat{\mathbf{V}}_k \widehat{\mathbf{V}}_k^\top \mathbf{M}_{j \cdot}^\top - \mathbf{M}_{j \cdot}^\top \right\|_2^2. \tag{19}$$

The first term can be futher decomposed as

$$\left\| \widehat{\mathbf{V}}_k \widehat{\mathbf{V}}_k^\top \mathbf{E}_{j \cdot}^\top \right\|_2^2 \leq 2 \left\| \widehat{\mathbf{V}}_k \widehat{\mathbf{V}}_k^\top \mathbf{E}_{j \cdot}^\top - \mathbf{V}_k \mathbf{V}_k^\top \mathbf{E}_{j \cdot}^\top \right\|_2^2 + 2 \left\| \mathbf{V}_k \mathbf{V}_k^\top \mathbf{E}_{j \cdot}^\top \right\|_2^2$$

$$\leq 2 \left\| \widehat{\mathbf{V}}_k \widehat{\mathbf{V}}_k^\top - \mathbf{V}_k \mathbf{V}_k^\top \right\|_2^2 \left\| \mathbf{E}_{j \cdot}^\top \right\|_2^2 + 2 \left\| \mathbf{V}_k \mathbf{V}_k^\top \mathbf{E}_{j \cdot}^\top \right\|_2^2 \tag{20}$$

Next, we bound the first term on the right hand side of (20). To that end, by Wedin $\sin \Theta$ Theorem (see [10, 32]) and recalling rank$(\mathbf{M}) = k$,

$$\left\| \widehat{\mathbf{V}}_k \widehat{\mathbf{V}}_k^\top - \mathbf{V}_k \mathbf{V}_k^\top \right\|_2 \leq \frac{\|\mathbf{E}\|_2}{\sigma_k(\mathbf{M})}$$

Then it follows that

$$\left\| \widehat{\mathbf{V}}_k \widehat{\mathbf{V}}_k^\top \mathbf{E}_{j \cdot}^\top \right\|_2^2 \leq 2 \frac{\|\mathbf{E}\|_2^2}{\sigma_k(\mathbf{M})^2} \left\| \mathbf{E}_{j \cdot}^\top \right\|_2^2 + 2 \left\| \mathbf{V}_k \mathbf{V}_k^\top \mathbf{E}_{j \cdot}^\top \right\|_2^2 \tag{21}$$

Then, we turn to bounding the second term from (19).

$$\left\| \widehat{\mathbf{V}}_k \widehat{\mathbf{V}}_k^\top \mathbf{M}_{j \cdot}^\top - \mathbf{M}_{j \cdot}^\top \right\|_2^2 = \left\| \widehat{\mathbf{V}}_k \widehat{\mathbf{V}}_k^\top \mathbf{M}_{j \cdot}^\top - \mathbf{V}_k \mathbf{V}_k^\top \mathbf{M}_{j \cdot}^\top \right\|_2^2$$

$$\leq \left\| \widehat{\mathbf{V}}_k \widehat{\mathbf{V}}_k^\top - \mathbf{V}_k \mathbf{V}_k^\top \right\|_2^2 \left\| \mathbf{M}_{j \cdot}^\top \right\|_2^2$$

$$\leq \frac{\|\mathbf{E}\|_2^2}{\sigma_k(\mathbf{M})^2} \left\| \mathbf{M}_{j \cdot}^\top \right\|_2^2. \tag{22}$$

Using (19), (21), and (22), we have,

$$\left\|\widehat{\boldsymbol{M}}_{j\cdot}^{\top} - \boldsymbol{M}_{j\cdot}^{\top}\right\|_2^2 \le 2\frac{\|\boldsymbol{E}\|_2^2}{\sigma_k(\boldsymbol{M})^2}\left(\left\|\boldsymbol{E}_{j\cdot}^{\top}\right\|_2^2 + \left\|\boldsymbol{M}_{j\cdot}^{\top}\right\|_2^2\right) + 2\left\|\mathbf{V}_k\mathbf{V}_k^{\top}\boldsymbol{E}_{j\cdot}^{\top}\right\|_2^2, \quad (23)$$

which completes the proof. $\qquad\square$

Now, Let's apply Lemma C.1 to our setting. Let $\mathbf{Z}_x$ be the stacked page matrix for $x_1(t), \ldots, x_n(t)$. Using Lemma C.1, and setting $\boldsymbol{Y} = \mathbf{Z}_y^{\top}$, $\boldsymbol{M} = \mathbf{Z}_f^{\top}$, $\boldsymbol{E} = \mathbf{Z}_x^{\top}$, and reusing $\mathbf{U}_k\boldsymbol{\Sigma}_k\mathbf{V}_k^{\top}$ to represent the top $k$ singular components of the SVD of $\mathbf{Z}_f$, results in the following deterministic bound for HSVT with rank set to $k$,

$$\left\|[\widehat{\mathbf{Z}}_f]_{\cdot j} - [\mathbf{Z}_f]_{\cdot j}\right\|_2^2 \le 2\frac{\|\mathbf{Z}_x\|_2^2}{\sigma_k(\mathbf{Z}_f)^2}\left(\left\|[\mathbf{Z}_x]_{\cdot j}\right\|_2^2 + \left\|[\mathbf{Z}_f]_{\cdot j}\right\|_2^2\right) + 2\left\|\mathbf{U}_k\mathbf{U}_k^{\top}[\mathbf{Z}_x]_{\cdot j}\right\|_2^2, \quad (24)$$

## C.2 Deterministic To High-Probability

Next, we convert the bound in (24) to a bound in expectation (as well as one in high-probability) for $\|\widehat{\mathbf{Z}}_f - \mathbf{Z}_f\|_{2,\infty}$. In particular, we establish

**Theorem C.1.** *Let assumptions 2.1, 2.2, 2.3, and 2.4 hold. Let* $\sigma_x = \frac{c_\lambda \sigma}{(1-\lambda_\star)}$, *where* $\lambda_\star = \max_{i,n}|\lambda_{ni}|$ *and* $c_\lambda = \max_n \sum_{i=1}^{p_n}\left|\prod_{1\le j\le p,\, j\ne i}\left(1 - \frac{\lambda_{nj}}{\lambda_{ni}}\right)^{-1}\right|$. *Then, the HSVT estimate* $\widehat{\mathbf{Z}}_f$ *with parameter $k$ is such that*

$$\max_{j\in[q]}\frac{1}{L}\left\|[\widehat{\mathbf{Z}}_f]_{\cdot j} - [\mathbf{Z}_f]_{\cdot j}\right\|_2^2 \le \frac{C\sigma_x^2(NT)^2}{L^3\sigma_k(\mathbf{Z}_f)^2}\left(\sigma^2 + f_{\max}^2\right) + \frac{C\sigma_x^2 k\log(NT/L)}{L}, \quad (25)$$

*with probability of at least* $1 - \frac{c}{(NT)^{11}}$.

*Proof.* We start by identifying certain high probability events. Subsequently, using these events and (24), we will conclude the proof.

**High Probability Events.** Let $q := NT/L$ be the number of columns in the stacked page matrix $\widehat{\mathbf{Z}}_f$, and let $C > 0$ be some positive absolute constant. Define

$$\begin{aligned}
E_1 &:= \left\{\|\mathbf{Z}_x\|_2 \le C\sigma_x\sqrt{q}\right\}, \\
E_2 &:= \left\{\|\mathbf{Z}_x\|_{\infty,2}, \|\mathbf{Z}_x\|_{2,\infty} \le C\sigma_x\sqrt{q}\right\}, \\
E_3 &:= \left\{\max_{j\in[q]}\|\mathbf{U}_k\mathbf{U}_k^{\top}[\mathbf{Z}_x]_{\cdot j}\|_2^2 \le C\sigma_x^2 k\log(q)\right\},
\end{aligned}$$

**Lemma C.2.** *For some positive constant $c_1 > 0$ and $C > 0$ large enough in definitions of $E_1, E_2$, and $E_3$,*

$$\begin{aligned}
\mathbb{P}(E_1) &\ge 1 - 2e^{-c_1 q}, \\
\mathbb{P}(E_2) &\ge 1 - 2e^{-c_1 q}, \\
\mathbb{P}(E_3) &\ge 1 - \frac{c_1}{(NT)^{11}}.
\end{aligned}$$

*Proof.* We bound the probability of events above below.

**Bounding $E_1$.** This is an immediate consequence of Lemma A.2.

**Bounding $E_2$.** Recall that we assume $L \le q$. Observe that for any matrix $A \in \mathbb{R}^{L\times q}$, $\|A\|_{\infty,2}, \|A\|_{2,\infty} \le \|A\|_2$. Thus using the argument to bound $E_1$, concludes the proof.

**Bounding $E_3$.** Let $u_1, \ldots, u_k$ be the orthonormal basis for $\mathbf{U}_k$. Then, consider for $j \in [q]$,

$$\|\mathbf{U}_k\mathbf{U}_k^\top [\mathbf{Z}_x]_{\cdot j}\|_2^2 = \sum_{i=1}^k \|u_i u_i^\top [\mathbf{Z}_x]_{\cdot j}\|_2^2 \leq \sum_{i=1}^k \left(u_i^\top [\mathbf{Z}_x]_{\cdot j}\right)^2 = \sum_{i=1}^k Z_i^2,$$

where $Z_i = u_i^\top [\mathbf{Z}_x]_{\cdot j}$. As shown in Lemma A.1, $[\mathbf{Z}_x]_{\cdot j}$ is a sub-gaussian vector with variance proxy $\sigma_x^2$. Using Lemma H.4, we have

$$\mathbb{P}\Big(\sum_{i=1}^k Z_i^2 > t\Big) \leq c\exp\left(-\frac{t}{16k\sigma_x^2}\right).$$

Therefore, for choice of $t = C\sigma_x^2 k \log(q)$ with large enough constant $C > 368$, we have,

$$\mathbb{P}\Big(\sum_{i=1}^k Z_i^2 > C\sigma_x^2 k \log(q)\Big) \leq \frac{c_1}{q^{23}}.$$

Recalling that $L \leq q$, and taking a union bound over all $j \in [q]$, we have that

$$\mathbb{P}\Big(E_4^c\Big) \leq \frac{c_1}{(qL)^{11}}.$$

$\square$

The following is an immediate corollary of the above stated bounds.

**Corollary C.1.** *Let $E := E_1 \cap E_2 \cap E_3$. Then,*

$$\mathbb{P}(E^c) \leq \frac{C_1}{(NT)^{11}}, \tag{26}$$

*where $C_1$ is an absolute positive constant.*

Thus, under event $E$, and using (24), we have, with probability $1 - \frac{c}{(NT)^{10}}$,

$$\max_{j \in [q]} \left\|[\widehat{\mathbf{Z}}_f]_{\cdot j} - [\mathbf{Z}_f]_{\cdot j}\right\|_2^2 \leq \frac{C\sigma_x^2 q}{\sigma_k(\mathbf{Z}_f)^2}\left(q\sigma_x^2 + qf_{\max}^2\right) + C\sigma_x^2 k \log(q)$$

Recall that $q = NT/L$ then,

$$\max_{j \in [q]} \frac{1}{L}\left\|[\widehat{\mathbf{Z}}_f]_{\cdot j} - [\mathbf{Z}_f]_{\cdot j}\right\|_2^2 \leq \frac{C\sigma_x^2 (NT)^2}{L^3\sigma_k(\mathbf{Z}_f)^2}\left(\sigma_x^2 + f_{\max}^2\right) + \frac{C\sigma_x^2 k \log(NT/L)}{L} \tag{27}$$

This completes the proof of Theorem C.1. Finally, now we are ready to bound $\mathsf{EstErr}(N, T)$. First, let $\Omega_n = \{\frac{(n-1)T}{L} + 1, \ldots, \frac{nT}{L}\}$ be the set of columns in the stacked Page matrix $\mathbf{Z}_f$ that belongs to the $n$-th time series $f_n(t)$. Note that

$$\frac{1}{T}\sum_{t=1}^T (\widehat{f}_n(t) - f_n(t))^2 = \frac{1}{T}\sum_{j \in \Omega_n}\left\|[\widehat{\mathbf{Z}}_f]_{\cdot j} - [\mathbf{Z}_f]_{\cdot j}\right\|_2^2$$

$$\leq \frac{1}{T}\sum_{j \in \Omega_n}\frac{C\sigma_x^2 (NT)^2}{L^2\sigma_k(\mathbf{Z}_f)^2}\left(\sigma_x^2 + f_{\max}^2\right) + C\sigma_x^2 k \log(NT/L)$$

$$\leq \frac{C\sigma_x^2 (NT)^2}{L^3\sigma_k(\mathbf{Z}_f)^2}\left(\sigma_x^2 + f_{\max}^2\right) + \frac{C\sigma_x^2 k \log(NT/L)}{L}.$$

Choosing $L = \sqrt{NT}$, and using $\sigma_k(\mathbf{Z}_f)^2 \geq \frac{\gamma^2 NT}{k}$, we get,

$$\frac{1}{T}\sum_{t=1}^T (\widehat{f}_n(t) - f_n(t))^2 \leq \frac{Cf_{\max}^2 \gamma^2 \sigma_x^4 k \log(NT)}{\sqrt{NT}}.$$

Finally and setting $k = RG$ concludes the proof. $\square$

# D Proof of Theorem 4.2

In this section, we prove Theorem 4.2, which bounds the estimation error $\|\widehat{\alpha}_n - \alpha_n\|_2 \ \forall n \in [N]$. First, recall the OLS estimate $\widehat{\alpha}_n = [\widehat{\alpha}_{n,1}, \widehat{\alpha}_{n,2}, \ldots \widehat{\alpha}_{n,p}]$ defined as

$$\widehat{\alpha}_n = \operatorname{argmin}_{\alpha \in \mathbb{R}^p} \sum_{t=p}^{T-1} (\widehat{x}_n(t+1) - \alpha^\top \widehat{X}_n(t))^2, \tag{28}$$

where $\widehat{X}_n(t) := [\widehat{x}_n(t), \ldots, \widehat{x}_n(t-p+1)]$. Alternatively, the OLS estimate can be defined as,

$$\widehat{\alpha}_n = (\widehat{\boldsymbol{X}}_n^\top \widehat{\boldsymbol{X}}_n)^{-1} \widehat{\boldsymbol{X}}_n^\top \widehat{Y}_n, \tag{29}$$

where $\widehat{\boldsymbol{X}}_n \in \mathbb{R}^{(T-p) \times p}$ is the row-wise concatenation of $\widehat{X}_n(t)$ for $t \in \{p, p+1, \ldots, T-1\}$, and $\widehat{Y}_n = [\widehat{x}_n(p+1), \ldots, \widehat{x}_n(T)]$. Further, consider the OLS estimate with access to the true $\mathrm{AR}(p)$ series $x_n(\cdot)$. Precisely, let $\bar{\alpha}_n$ defined as

$$\bar{\alpha}_n = (\boldsymbol{X}_n^\top \boldsymbol{X}_n)^{-1} \boldsymbol{X}_n^\top Y_n, \tag{30}$$

where $\boldsymbol{X}_n \in \mathbb{R}^{(T-p) \times p}$ is the row-wise concatenation of $X_n(t) := [x_n(t), \ldots, x_n(t-p+1)]$ for $t \in \{p, p+1, \ldots, T-1\}$, and $Y_n = [x_n(p+1), \ldots, x_n(T)]$.

In what follows, as we do the analysis for one process $x_n$, we drop the subscript $n$. We note that the analysis holds for all $n \in [N]$. First, note that,

$$\|\widehat{\alpha} - \alpha\|_2^2 \le 2 \|\widehat{\alpha} - \bar{\alpha}\|_2^2 + 2 \|\bar{\alpha} - \alpha\|_2^2 \tag{31}$$

To bound $\|\bar{\alpha} - \alpha\|_2^2$, we use Corollary E.2 which states that with probability $1 - \frac{c}{T^{11}}$,

$$\|\bar{\alpha} - \alpha\|_2^2 \le C \frac{p^2}{T \lambda_{\min}(\Psi)} \log\left(\frac{T \lambda_{\max}(\Psi)}{\lambda_{\min}(\Psi)}\right). \tag{32}$$

Next, to bound $\|\widehat{\alpha} - \bar{\alpha}\|_2^2$, let $M^\dagger := (M^\top M)^{-1} M^\top$ denote the Moore–Penrose inverse of a matrix $M$. Further, let $\delta(t) = x(t) - \widehat{x}(t) = \widehat{f}(t) - f(t)$, and $\boldsymbol{\Delta} = \boldsymbol{X} - \widehat{\boldsymbol{X}}$, i.e., $\boldsymbol{\Delta}$ is the row-wise concatenation of $\Delta(t) := [\delta(t), \ldots, \delta(t-p+1)]^\top$ for $t \in \{p, p+1, \ldots, T-1\}$. Then, with $\Delta_Y := [\delta(p+1), \ldots, \delta(T)]$, we have

$$\begin{aligned}
\|\widehat{\alpha} - \bar{\alpha}\|_2 &= \left\|\widehat{\boldsymbol{X}}^\dagger \widehat{Y} - \boldsymbol{X}^\dagger Y\right\|_2 \\
&= \left\|\widehat{\boldsymbol{X}}^\dagger \Delta_Y + (\boldsymbol{X}^\dagger - \widehat{\boldsymbol{X}}^\dagger) Y\right\|_2 \\
&\le \left\|\widehat{\boldsymbol{X}}^\dagger \Delta_Y\right\|_2 + \left\|(\boldsymbol{X}^\dagger - \widehat{\boldsymbol{X}}^\dagger) Y\right\|_2 \\
&\le \left\|\widehat{\boldsymbol{X}}^\dagger\right\|_2 \|\Delta_Y\|_2 + \left\|\boldsymbol{X}^\dagger - \widehat{\boldsymbol{X}}^\dagger\right\|_2 \|Y\|_2 \\
&\le \left\|\widehat{\boldsymbol{X}}^\dagger\right\|_2 \|\Delta_Y\|_2 + 2 \max\left\{\left\|\boldsymbol{X}^\dagger\right\|_2^2, \left\|\widehat{\boldsymbol{X}}^\dagger\right\|_2^2\right\} \|\boldsymbol{\Delta}\|_2 \|Y\|_2
\end{aligned} \tag{33}$$

Where Lemma H.6 is used in the last inequality. Note that,

$$\left\|\boldsymbol{X}^\dagger\right\|_2^{-1} = \inf_{v \in \mathbb{R}^p : \|v\|_2 = 1} \|\boldsymbol{X} v\|_2 = \sqrt{\lambda_{\min}(\boldsymbol{X}^\top \boldsymbol{X})} \tag{34}$$

$$\begin{aligned}
\left\|\widehat{\boldsymbol{X}}^\dagger\right\|_2^{-1} &= \inf_{v \in \mathbb{R}^p : \|v\|_2 = 1} \left\|\widehat{\boldsymbol{X}} v\right\|_2 \\
&= \inf_{v \in \mathbb{R}^p : \|v\|_2 = 1} \|(\boldsymbol{X} - \boldsymbol{\Delta}) v\|_2 \\
&\ge \sqrt{\lambda_{\min}(\boldsymbol{X}^\top \boldsymbol{X})} - \|\boldsymbol{\Delta}\|_2
\end{aligned} \tag{35}$$

Using (33), (34), (35), and the theorem conditions, we get

$$\|\widehat{\alpha} - \bar{\alpha}\|_2 \leq \frac{\|\Delta_Y\|_2}{\sqrt{\lambda_{\min}(\boldsymbol{X}^\top \boldsymbol{X})} - \|\boldsymbol{\Delta}\|_2} + \frac{2\|\boldsymbol{\Delta}\|_2 \|Y\|_2}{\min\left\{\lambda_{\min}(\boldsymbol{X}^\top \boldsymbol{X}), \left(\sqrt{\lambda_{\min}(\boldsymbol{X}^\top \boldsymbol{X})} - \|\boldsymbol{\Delta}\|_2\right)^2\right\}}.$$

$$(36)$$

Using Corollary E.1 we get that with probability $1 - \frac{c}{T^{11}}$

$$\lambda_{\min}(\boldsymbol{X}^\top \boldsymbol{X}) \geq \frac{1}{2}\sigma^2(T - p)\lambda_{\min}(\Psi). \tag{37}$$

Now, note that $\|\Delta_Y\|_2^2 = \sum_{t=p+1}^{T}(\widehat{f}(t) - f(t))^2$, and

$$\|\boldsymbol{\Delta}\|_2^2 \leq \|\boldsymbol{\Delta}\|_F^2 \leq p \sum_{t=1}^{T}(\widehat{f}(t) - f(t))^2. \tag{38}$$

Recall that $\sum_{t=1}^{T}(\widehat{f}(t) - f(t))^2 = T\mathsf{EstErr}(N, T, n)$, which we will refer to herein as $\mathsf{EstErr}(N, T)$ as we drop the dependence on $n$. Therefore, we have,

$$\|\boldsymbol{\Delta}\|_2^2 \leq pT\mathsf{EstErr}(N, T) \tag{39}$$

$$\|\Delta_Y\|_2^2 \leq T\mathsf{EstErr}(N, T). \tag{40}$$

Finally, using Lemma H.4, with probability at least $1 - \frac{c}{T^{11}}$ it holds for an absolute constant $C > 4$,

$$\|Y\|_2 \leq C\sigma_x \sqrt{T \log(T)}. \tag{41}$$

First, note that, using (37) and (39), we have

$$\sqrt{\lambda_{\min}(\boldsymbol{X}^\top \boldsymbol{X})} - \|\boldsymbol{\Delta}\|_2 \geq \frac{\sigma\sqrt{(T - p)\lambda_{\min}(\Psi)}}{\sqrt{2}} - \sqrt{pT\mathsf{EstErr}(N, T)}$$

$$\geq C\sigma\sqrt{T\lambda_{\min}(\Psi)}, \tag{42}$$

for sufficiently small $\mathsf{EstErr}(N, T)$ such that $\mathsf{EstErr}(N, T) \leq \frac{\sigma^2 \lambda_{\min}(\Psi)}{6p}$. Now using (36), (37), (39), (40), (41), and (42) we have,

$$\|\widehat{\alpha} - \bar{\alpha}\|_2 \leq \frac{C\sigma_x \sqrt{p\mathsf{EstErr}(N, T)\log(T)}}{\min\{\sigma^2 \lambda_{\min}(\Psi), \sigma\sqrt{\lambda_{\min}(\Psi)}\}}$$

Therefore,

$$\|\widehat{\alpha} - \bar{\alpha}\|_2^2 \leq C\frac{p\log(T)}{\lambda_{\min}(\Psi)}\frac{\sigma_x^2}{\sigma^2}\mathsf{EstErr}(N, T)\max\left\{1, \frac{1}{\sigma\lambda_{\min}(\Psi)}\right\} \tag{43}$$

finally, using (31), (32), and (43), we get with probability $1 - \frac{c}{T^{11}}$,

$$\|\widehat{\alpha} - \alpha\|_2^2 \leq C\frac{p\log(T)}{\lambda_{\min}(\Psi)}\frac{\sigma_x^2}{\sigma^2}\mathsf{EstErr}(N, T)\max\left\{1, \frac{1}{\sigma^2\lambda_{\min}(\Psi)}\right\} + \frac{Cp^2}{T\lambda_{\min}(\Psi)}\log\left(\frac{T\lambda_{\max}(\Psi)}{\lambda_{\min}(\Psi)}\right),$$

$$(44)$$

which completes the proof of Theorem 4.2.

# E   Identification of AR processes

In this section, we state key theorems for the identification of AR processes without the perturbation we consider in this paper. The theorems in this section and their proofs follow closely the ones in [17], we modify them to accommodate general sub-gaussian noise and state them fully for completeness.

First, consider the stationary $\mathrm{AR}(p)$ process defined by the transition matrix

$$\boldsymbol{A} = \begin{bmatrix} \alpha_{1:p-1}^\top & \alpha_p \\ I_{p-1} & 0_{p-1} \end{bmatrix}, \tag{45}$$

where $0_{p-1}$ is the column vector of $p-1$ zeroes, and input vector

$$B = \begin{bmatrix} 1 \\ 0_{p-1} \end{bmatrix}. \tag{46}$$

Then, with $X(t) := [x(t), \dots, x(t-p+1)]$, the AR process is described as

$$X(t) = \boldsymbol{A}X(t-1) + B\eta_t. \tag{47}$$

With $\eta_t$ being a sub-gaussian random variable with variance proxy $\sigma^2$ With this setup, we define two important matrices,

$$\Gamma = \sum_{t=0}^{\infty} \boldsymbol{A}^t (\boldsymbol{A}^\top)^t \tag{48}$$

$$\Psi = \sum_{t=0}^{\infty} \boldsymbol{A}^t BB^\top (\boldsymbol{A}^\top)^t, \tag{49}$$

where $\Psi$ is known as the *controllability Gramian*.

**Theorem E.1.** *Consider the* $\mathrm{AR}(p)$ *process described in* (47)*,* $\{\eta_t\}$ *is an i.i.d. mean-zero sub-gaussian noise with variance* $\sigma^2$*. Given* $0 < \epsilon \le 1$*, define the following quantities:*

$$V_\ell := \sigma^2 (T-p) (\Psi - \epsilon\Gamma)$$

$$V_u := \sigma^2 (T-p) (\Psi + \epsilon\Gamma)$$

$$\delta(\epsilon, T) := c \exp\left(-\tilde{c}\frac{\epsilon\sqrt{T}}{p}\right)$$

$$T_0(\epsilon) := \frac{75^2 \sigma_x^4 p^2 \log(p)^2}{\sigma^4 \epsilon^2} \max\{1, \|\alpha\|_2^4\}$$

*where $c$ and $\tilde{c}$ are constants that depends only on the process coefficients $\alpha_1, \dots, \alpha_p$. Then, for $T > T_0(\epsilon)$ and all valid values of $\epsilon$ such that $V_\ell \succ 0$, we have*

$$\mathbb{P}\left(V_\ell \preceq \mathbf{X}^\top \mathbf{X} \preceq V_u\right) \ge 1 - \delta(\epsilon, T).$$

**Remark E.1.** *To get $V_\ell \succ 0$, one must choose a sufficiently small $\epsilon$ such that $(\Psi - \epsilon\Gamma) \succ 0$. Using Weyl's inequality, we have*

$$\lambda_{\min}(\Psi - \epsilon\Gamma) \ge \lambda_{\min}(\Psi) - \epsilon\lambda_{\max}(\Gamma). \tag{50}$$

*This suggest that $\epsilon < \frac{\lambda_{\min}(\Psi)}{\lambda_{\max}(\Gamma)}$ yields $V_\ell \succ 0$. For example, a choice of $\epsilon = \frac{\lambda_{\min}(\Psi)}{2\lambda_{\max}(\Gamma)}$ will yield $V_\ell \succeq \frac{1}{2}\sigma^2(T-p)\lambda_{\min}(\Psi)I_p$. Note that this ensures $\epsilon \le 1/2$, since $\epsilon \le \frac{\lambda_{\max}(\Psi)}{\lambda_{\max}(\Gamma)}$ and $\Gamma \succeq \Psi$ as can be seen from*

$$z_i^\top (\Gamma - \Psi)z = \sum_{t=0}^{\infty} z_i^\top \boldsymbol{A}^t (\boldsymbol{A}^\top)^t z - z_i^\top \boldsymbol{A}^t BB^\top (\boldsymbol{A}^\top)^t z$$

$$= \sum_{t=0}^{\infty} \left\|(\boldsymbol{A}^\top)^t z\right\|_2^2 - \left(B^\top (\boldsymbol{A}^\top)^t z\right)^2$$

$$\ge \sum_{t=0}^{\infty} \left\|(\boldsymbol{A}^\top)^t z\right\|_2^2 - \underbrace{\|B\|_2^2}_{1} \left\|(\boldsymbol{A}^\top)^t z\right\|_2^2$$

$$\ge 0.$$

*Proof.* First, write

$$\mathbf{X}^\top \mathbf{X} = \sum_{t=p}^{T-1} X_t X_t^\top.$$

Expanding the summand using the process dynamics yields

$$\begin{aligned}
X_{t+1}X_{t+1}^\top &= (AX_t + B\eta_{t+1})(AX_t + B\eta_{t+1})^\top \\
&= AX_t X_t^\top A^\top + \eta_{t+1}AX_t B^\top + \eta_{t+1}BX_t^\top A^\top + \eta_{t+1}^2 BB^\top.
\end{aligned}$$

Now, defining $V_T = \mathbf{X}^\top \mathbf{X}/(T-p)$, we can sum the expression above over $t = p-1, \ldots, T-2$ to get

$$V_T = AV_T A^\top + \underbrace{\frac{1}{T-p}\left( A(X_{p-1}X_{p-1}^\top - X_{T-1}X_{T-1}^\top)A^\top + \sum_{t=p-1}^{T-2} (\eta_{t+1}AX_t B^\top + \eta_{t+1}BX_t^\top A^\top + \eta_{t+1}^2 BB^\top) \right)}_{E_T}.$$

This is called a Lyapunov equation, and due to the stability of the process it has solution $V_T = \sum_{t=0}^\infty A^t E_T (A^\top)^t$. The key step is to argue that $E_T$ converges to $\sigma^2 BB^\top$ with high probability, which will allow us to show that $V_T$ converges to $\sigma^2 \Psi$. To do this, we will split the terms in $E_T$ into three groups, handling them separately. For some $\epsilon > 0$, we will bound the probability of the following events:

$$\begin{aligned}
\mathcal{E}_1 &:= \left\{ \rho\left( A(X_{p-1}X_{p-1}^\top - X_{T-1}X_{T-1}^\top)A^\top \right) \leq \epsilon\sigma^2(T-p)/3 \right\}, \\
\mathcal{E}_2 &:= \left\{ \rho\left( \sum_{t=p-1}^{T-2} (\eta_{t+1}^2 BB^\top - \sigma^2 BB^\top) \right) \leq \epsilon\sigma^2(T-p)/3 \right\}, \\
\mathcal{E}_3 &:= \left\{ \rho\left( \sum_{t=p-1}^{T-2} \eta_{t+1}AX_t B^\top + \eta_{t+1}BX_t^\top A^\top \right) \leq \epsilon\sigma^2(T-p)/3 \right\},
\end{aligned}$$

where $\rho(\cdot)$ denotes the spectral radius of a matrix.

**Lemma E.1.** *With the setup above, it holds that*

$$\mathbb{P}(\mathcal{E}_1) \geq 1 - c\exp\left( -\frac{\epsilon\sigma^2\sqrt{T}}{3\sigma_x^2 p(\|\alpha\|_2^2 + 1)} \right)$$

*for some absolute constant $c > 0$.*

*Proof.* Note that

$$\begin{aligned}
&\rho\left( A(X_{p-1}X_{p-1}^\top - X_{T-1}X_{T-1}^\top)A^\top \right) \leq \epsilon\sigma^2(T-p)/3 \\
\Longleftarrow\ &\rho\left( A(X_{p-1}X_{p-1}^\top + X_{T-1}X_{T-1}^\top)A^\top \right) \leq \epsilon\sigma^2(T-p)/3 \\
\Longleftarrow\ &\|AX_{p-1}\|_2^2 + \|AX_{T-1}\|_2^2 \leq \epsilon\sigma^2(T-p)/3 \\
\Longleftarrow\ &\|X_{p-1}\|_2^2 + \|X_{T-1}\|_2^2 \leq \frac{\epsilon\sigma^2(T-p)}{3\|A\|_2} \\
\Longleftarrow\ &\|X_{p-1}\|_2^2 \vee \|X_{T-1}\|_2^2 \leq \frac{\epsilon\sigma^2(T-p)}{6\|A\|_2},
\end{aligned}$$

so, considering the fact that $X_{p-1}$ and $X_{T-1}$ are identically distributed by stationarity, it suffices to upper bound the probability of

$$\mathcal{E}' = \left\{ \|X_{p-1}\|_2^2 > \frac{\epsilon\sigma^2(T-p)}{6\|A\|_2} \right\}.$$

First, recall that $X_{p-1}$ is a sub-gaussian vector with variance proxy $\frac{\sigma^2\gamma^2}{(1-\lambda^*)^2} := \sigma_x^2$ (see Lemma A.2). Thus, using Lemma H.2, for

$$\mathbb{P}\left(\|X_{p-1}\|_2^2 \geq \frac{\epsilon\sigma^2(T-p)}{6\|A\|_2}\right) \leq c\exp\left(-\frac{\epsilon\sigma^2(T-p)}{96\sigma_x^2 p\|A\|_2}\right)$$

$$\leq c\exp\left(-\frac{\epsilon\sigma^2\sqrt{T}}{3\sigma_x^2\|A\|_2}\right),$$

where in the last inequality, we used the assumption $T > T_0(\epsilon) > 75^2 p^2$. Finally, note that $\|A\|_2 \leq \mathrm{Tr}(A) = \|\alpha\|_2^2 + p - 1 \leq p(\|\alpha\|_2^2 + 1)$. Recalling that $\mathbb{P}(\mathcal{E}_1) > 1 - 2\mathbb{P}(\mathcal{E}')$ concludes the proof. $\qquad\square$

**Lemma E.2.** *Continuing with the setup above, it holds that*

$$\mathbb{P}(\mathcal{E}_2) \geq 1 - c\exp\left(\frac{-\epsilon\sqrt{T-p}}{48}\right).$$

*Proof.* Since the spectral radius of $BB^\top$ is unity, $\mathcal{E}_2$ is equivalent to the complement of the event

$$\mathcal{E}' = \left\{\left|(T-p)^{-1}\sum_{t=p-1}^{T-2}\left(\eta_{t+1}^2 - \sigma^2\right)\right| > \epsilon\sigma^2/3\right\}.$$

Since $\eta_{t+1}$ is sub-gaussian with variance proxy $\sigma^2$, $\eta_{t+1}^2 - \sigma^2$ is sub-exponential with parameter $16\sigma^2$. Thus, Lemma H.5 applies, and therefore,

$$\mathbb{P}(\mathcal{E}') \leq c\exp\left(\frac{-\epsilon\sqrt{T-p}}{48}\right). \tag{51}$$

$\qquad\square$

**Lemma E.3.** *With the same setup, we have*

$$\mathbb{P}(\mathcal{E}_3) \geq 1 - c\exp\left(-\frac{\epsilon\sigma^2\sqrt{T}}{75p\sigma_x^2\left(2 + \|\alpha\|_2^2\right)}\right)$$

*as long as*

$$T \geq T_0(\epsilon) := \frac{75^2\sigma_x^4 p^2\log(p)^2}{\sigma^4\epsilon^2}\left(2 + \|\alpha\|_2^2\right)^2.$$

*Proof.* Recall the definition

$$\mathcal{E}_3 := \left\{\rho\left(\sum_{t=p-1}^{T-2}\eta_{t+1}AX_tB^\top + \eta_{t+1}BX_t^\top A^\top\right) \leq \epsilon\sigma^2(T-p)/3\right\}.$$

For the sake of a variational analysis of the spectral radius, consider any vector $q = [q_1\ \tilde{q}^\top]^\top \in \mathbb{R}^p$ such that $\|q\|_2 = 1$. For notational simplicity. Then

$$q^\top\underbrace{\left(\sum_{t=p-1}^{T-2}\eta_{t+1}AX_tB^\top + \eta_{t+1}BX_t^\top A^\top\right)}_{M+M^\top}q \leq 2\|Mq\|_2.$$

Next, we aim to bound $\|Mq\|_2$. First note that only the first column of $M$ is nonzero which evaluates to

$$M_{.,1} = \eta_{t+1}\cdot\begin{bmatrix}\sum_{t=p-1}^{T-2}\alpha^\top X_t\\ \sum_{t=p-1}^{T-2}x_t\\ \vdots\\ \sum_{t=p-1}^{T-2}x_{t-p+1}\end{bmatrix} \tag{52}$$

Therefore, we have $Mq = q_1 M_{.,1}$, which implies that for any unit vector $q$,

$$q^\top(M+M^\top)q \leq 2\sqrt{\sum_{i=1}^{p}M_{i,1}^2}.$$

Therefore, we will bound $\mathbb{P}\left(\mathcal{E}_3^c\right)$ by

$$\mathbb{P}\left(\mathcal{E}_3^c\right) \leq \mathbb{P}\left(4\sum_{i=1}^{p} M_{i,1}^2 \geq \epsilon^2\sigma^4(T-p)^2/9\right)$$

$$\leq \sum_{i=1}^{p}\mathbb{P}\left(|M_{i,1}| \geq \frac{\epsilon\sigma^2(T-p)}{6\sqrt{p}}\right) \tag{53}$$

First note that each term $\mathbb{P}\left(|M_{i,1}| \geq \frac{\epsilon\sigma^2(T-p)}{6\sqrt{p}}\right)$ can be upper bounded by $2\mathbb{P}\left(\sum_{t=p}^{T-1} z_i^\top X_{t-1}\eta_t \geq \frac{\epsilon\sigma^2(T-p)}{6\sqrt{p}}\right)$ where $z_1 = \alpha$ and $z_i = e_{i-1}$ for $i > 1$ where $e_i$ is the $i$-th standard basis vector. Using Lemma H.7 and choosing $Z_t = z_i^\top X_{t-1}$ and $W_t = \eta_t$, we have that

$$\mathbb{P}\left(\left\{\sum_{t=p}^{T-1}\eta_t z_i^\top X_{t-1} \geq \kappa\right\} \cap \left\{\sum_{t=p}^{T-1}(z_i^\top X_{t-1})^2 \leq \lambda\right\}\right) \leq \exp\left(-\frac{\kappa^2}{2\sigma^2\lambda}\right),$$

and thus

$$\mathbb{P}\left(\sum_{t=p}^{T-1}\eta_t z_i^\top X_{t-1} \geq \kappa\right) \leq \exp\left(-\frac{\kappa^2}{2\sigma^2\lambda}\right) + \mathbb{P}\left(\sum_{t=p}^{T-1}(z_i^\top X_{t-1})^2 > \lambda\right)$$

$$\leq \exp\left(-\frac{\kappa^2}{2\sigma^2\lambda}\right) + \mathbb{P}\left(\sum_{t=p}^{T-1}\|X_{t-1}\|_2^2 > \lambda/\|z_i\|_2^2\right).$$

What is left is to bound $\mathbb{P}\left(\sum_{t=0}^{T-2} x_t^2 > \frac{\lambda}{p\|z\|_2^2}\right)$. Using Lemma H.2, we get,

$$\mathbb{P}\left(\sum_{t=0}^{T-2} x_t^2 > \frac{\lambda}{p\|z\|_2^2}\right) \leq c\exp\left(-\frac{\lambda}{16pT\sigma_x^2\|z\|_2^2}\right) \tag{54}$$

Choosing $\kappa = \epsilon\sigma^2(T-p)/(6\sqrt{p})$ and $\lambda = \|z\|_2^2 \epsilon\sigma^2 T^{3/2}$ yields

$$\mathbb{P}\left(\sum_{t=p}^{T-1}\eta_t z_i^\top X_{t-1} \geq \frac{\epsilon\sigma^2(T-p)}{6\sqrt{p}}\right) \leq \exp\left(-\frac{\epsilon(T-p)^2}{72p\|z\|_2^2 T^{3/2}}\right) + c\exp\left(-\frac{\epsilon\sigma^2\sqrt{T}}{16p\sigma_x^2}\right).$$

Which implies, that for $T > 50p$, we have,

$$\mathbb{P}\left(\sum_{t=p}^{T-1}\eta_t z_i^\top X_{t-1} \geq \frac{\epsilon\sigma^2(T-p)}{6\sqrt{p}}\right) \leq c\exp\left(-\frac{\epsilon\sqrt{T}}{p}\min\left\{\frac{1}{75\|z\|_2^2}, \frac{\sigma^2}{16\sigma_x^2}\right\}\right), \tag{55}$$

for some absolute constants $c, C > 0$. Now, note that

$$\min\left\{\frac{1}{75\|z\|_2^2}, \frac{\sigma^2}{16\sigma_x^2}\right\} \geq \frac{1}{75}\min\left\{\frac{1}{\|z_i\|_2^2}, \frac{\sigma^2}{\sigma_x^2}\right\}$$

$$\geq \frac{1}{75}\frac{\sigma^2}{\sigma_x^2}\min\left\{\frac{1}{\|z_i\|_2^2}, 1\right\}$$

$$\geq \frac{1}{75}\frac{\sigma^2}{\sigma_x^2}\frac{1}{1+\|z_i\|_2^2}$$

$$\geq \frac{1}{75}\frac{\sigma^2}{\sigma_x^2}\frac{1}{2+\|\alpha\|_2^2}.$$

Thus, using (55) and (53), we have

$$\mathbb{P}\left(\mathcal{E}_3^c\right) \leq \sum_{i=1}^{p} \mathbb{P}\left(|M_{i,1}| \geq \frac{\epsilon\sigma^2(T-p)}{6\sqrt{p}}\right) \tag{56}$$

$$\leq \sum_{i=1}^{p} 2\mathbb{P}\left(\sum_{t=p}^{T-1} z_i^\top X_{t-1}\eta_t \geq \frac{\epsilon\sigma^2(T-p)}{6\sqrt{p}}\right) \tag{57}$$

$$\leq cp\exp\left(-\frac{\epsilon\sigma^2\sqrt{T}}{75p\sigma_x^2\left(2+\|\alpha\|_2^2\right)}\right). \tag{58}$$

Using the condition $T > T_0(\epsilon)$ concludes the proof. $\qquad\square$

Now we argue that the sum of the probabilities deduced from lemmas (E.1), (E.2), and (E.3) can be expressed in the simple form of $\delta(\epsilon, T)$ as defined above. First, from lemma (E.1) we have

$$\mathbb{P}\left(\mathcal{E}_1'\right) \leq c\exp\left(-\frac{1}{3(\|\alpha\|_2^2+1)}\frac{\sigma^2}{\sigma_x^2}\frac{\epsilon\sqrt{T}}{p}\right).$$

From lemma (E.2), and $T > T_0(\epsilon) > 50p$ we get

$$\mathbb{P}\left(\mathcal{E}_2'\right) \leq c\exp\left(\frac{-\epsilon\sqrt{T-p}}{48}\right) \leq c\exp\left(\frac{-\epsilon\sqrt{T}}{49}\right).$$

Lastly, lemma (E.3) says, for $T \geq T_0(\epsilon)$,

$$\mathbb{P}\left(\mathcal{E}_3'\right) \leq c\exp\left(-\frac{\epsilon\sigma^2\sqrt{T}}{75p\sigma_x^2\left(2+\|\alpha\|_2^2\right)}\right)$$

Thus we can write

$$\mathbb{P}\left(\mathcal{E}_1' \cup \mathcal{E}_2' \cup \mathcal{E}_3'\right) \leq c\exp\left(-\frac{1}{75\left(2+\|\alpha\|_2^2\right)}\frac{\epsilon\sigma^2\sqrt{T}}{p\sigma_x^2}\right) := \delta(\epsilon, T).$$

Where in the statement we set $\tilde{c} := \frac{1}{75\left(2+\|\alpha\|_2^2\right)}\frac{\sigma^2}{\sigma_x^2}$. Using lemmas (E.1), (E.2), and (E.3) and the subadditivity of the spectral radii of symmetric matrices, we get with probability at least $1 - \delta(\epsilon, T)$ that

$$\mathcal{E}_1 \cap \mathcal{E}_2 \cap \mathcal{E}_3 \implies \rho\left(E_T - \sigma^2 BB^\top\right) \leq \epsilon\sigma^2$$
$$\implies \sigma^2(BB^\top - \epsilon I) \preceq E_T \preceq \sigma^2(BB^\top + \epsilon I)$$
$$\implies \sigma^2(T-p)(\Psi - \epsilon\Gamma) \preceq \mathbf{X}^\top\mathbf{X} \preceq \sigma^2(T-p)(\Psi + \epsilon\Gamma).$$

$$\square$$

Finally, we establish the following corollary, which is an immediate consequence of Theorem E.1.

**Corollary E.1.** *Consider the* $\mathrm{AR}(p)$ *process* $x_t$ *described in (A.1). Let* $\mathbf{X} \in \mathbb{R}^{(T-p)\times p}$ *such that* $[\mathbf{X}]_{ij} = x_{i+j-1}$. *Then, for sufficiently large* $T$ *such that* $\frac{\log(T)}{\sqrt{T}} < \frac{\tilde{c}}{22p}\left(\frac{\lambda_{\min}(\Psi)}{\lambda_{\max}(\Gamma)}\right)$, *we have that with probability* $1 - \frac{c}{T^{11}}$,

$$\lambda_{\min}(\mathbf{X}^\top\mathbf{X}) \geq \frac{1}{2}\sigma^2(T-p)\lambda_{\min}(\Psi), \tag{59}$$

$$\lambda_{\max}(\mathbf{X}^\top\mathbf{X}) \leq \frac{3}{2}\sigma^2(T-p)\lambda_{\max}(\Psi), \tag{60}$$

*where* $c, C$ *are constants and* $\tilde{c} := \frac{1}{75\left(2+\|\alpha\|_2^2\right)}\frac{\sigma^2}{\sigma_x^2}$, $\psi$ *(the controllability Gramian) and* $\Gamma$ *are defined in (14).*

*Proof.* Note that according to Theorem E.1, Weyl's inequality (see Lemma H.3), and using $\epsilon = \frac{11p\log(T)}{\tilde{c}\sqrt{T}}$ we have, with probability $1 - \frac{c}{T^D}$,

$$\lambda_{\min}(\boldsymbol{X}^\top \boldsymbol{X}) \geq \sigma^2(T-p)\left(\lambda_{\min}(\Psi) - \epsilon\lambda_{\max}(\Gamma)\right) \tag{61}$$

$$\geq \sigma^2(T-p)\left(\lambda_{\min}(\Psi) - 11\frac{p\log(T)}{\tilde{c}\sqrt{T}}\lambda_{\max}(\Gamma)\right) \tag{62}$$

$$\geq \frac{1}{2}\sigma^2(T-p)\left(\lambda_{\min}(\Psi)\right). \tag{63}$$

Also,

$$\lambda_{\max}(\boldsymbol{X}^\top \boldsymbol{X}) \leq \sigma^2(T-p)\left(\lambda_{\min}(\Psi) + \epsilon\lambda_{\max}(\Gamma)\right) \tag{64}$$

$$\leq \frac{3}{2}\sigma^2(T-p)\left(\lambda_{\min}(\Psi)\right). \tag{65}$$

What is left is to check the condition on $T$ and $\epsilon$. Note that Theorem E.1 requires $T > \frac{75^2\sigma_x^4 p^2\log(p)^2}{\sigma^4\epsilon^2}\max\{1, \|\alpha\|_2^4\}$, which with our choice of $\epsilon$ implies $T > p$. To see that the choice of $\epsilon$ is valid, note that $\frac{11p\log(T)}{\tilde{c}\sqrt{T}} \leq \frac{\lambda_{\min}(\Psi)}{2\lambda_{\max}(\Gamma)}$, which validates the choice of $\epsilon$. $\qquad\square$

**Theorem E.2.** *Consider the* $\mathrm{AR}(p)$ *process described in (47). where* $\{\eta_t\}$ *is an i.i.d. mean-zero sub-gaussian noise with variance* $\sigma^2$. *Let* $\alpha := [\alpha_1, \ldots, \alpha_p]^\top$ *and* $\bar{\alpha}$ *be its least squares estimator. If* $T \geq T_0(\epsilon)$, *then for any unit vector* $w \in \mathcal{S}_p$,

$$
\mathbb{P}\left(\left|w^\top(\bar{\alpha} - \alpha)\right| > 2\sigma \left\|w^\top V_\ell^{-1/2}\right\|_2 \sqrt{\log\left(\frac{\det\left(V_u V_\ell^{-1} + I_p\right)^{1/2}}{\delta(\epsilon, T)}\right)}\right) \leq 2\delta(\epsilon, T),
$$

*where* $V_\ell, V_u \; T_0(\epsilon)$, *and* $\delta(\epsilon, T)$ *are as described in Theorem E.1.*

*Proof.* Write $E = [\eta_{p+1}, \ldots, \eta_T]^\top$. Using Cauchy-Schwarz, we have

$$
\begin{aligned}
|w^\top(\hat{\alpha}(T) - \alpha)| &= |w^\top(\mathbf{X}^\top\mathbf{X})^{-1}\mathbf{X}^\top E| \\
&\leq \left\|w^\top(\mathbf{X}^\top\mathbf{X})^{-1/2}\right\|_2 \left\|(\mathbf{X}^\top\mathbf{X})^{-1/2}\mathbf{X}^\top E\right\|_2.
\end{aligned}
$$

Now we will need to introduce a lemma from [1]:

**Lemma E.4.** *Let* $\{\mathcal{F}_t\}_{t=0}^\infty$ *be a filtration. Let* $\{\eta_t\}_{t=1}^\infty$ *be a real-valued stochastic process such that* $\eta_t$ *is* $\mathcal{F}_t$-*measurable and is conditionally sub-gaussian with variance proxy* $\sigma^2 > 0$. *Further, let* $\{X_t\}_{t=0}^\infty$ *be an* $\mathbb{R}^d$-*valued stochastic process such that* $X_t$ *is* $\mathcal{F}_{t-1}$-*measurable. Consider any positive definite matrix* $V \in \mathbb{R}^{d \times d}$. *For any* $t \geq 1$, *define*

$$
\bar{V}_t = V + \sum_{s=1}^t X_s X_s^\top, \quad S_t = \sum_{s=1}^t \eta_s X_s.
$$

*Then for any* $\delta > 0$, *with probability at least* $1 - \delta$ *for all* $t \geq 1$,

$$
\|S_t\|_{\bar{V}_t^{-1}}^2 \leq 2\sigma^2 \left(\frac{\det\left(\bar{V}_t\right)^{1/2} \det\left(V\right)^{-1/2}}{\delta}\right).
$$

We will apply this result to bound the second term above. Choosing $V = V_\ell$, $\sum_{s=1}^t X_s X_s^\top = \mathbf{X}^\top\mathbf{X}$, and $S_t = \mathbf{X}^\top E$, we have with probability at least $1 - \delta(\epsilon, T)$

$$
\|\mathbf{X}^\top E\|_{(\mathbf{X}^\top\mathbf{X}+V_\ell)^{-1}} \leq \sqrt{2\sigma^2 \log\left(\frac{\det\left(\mathbf{X}^\top\mathbf{X} + V_\ell\right)^{1/2} \det\left(V_\ell\right)^{-1/2}}{\delta(\epsilon, T)}\right)}.
$$

Now further condition on the event of Theorem E.1, which occur together with the condition above with probability at least $1 - 2\delta(\epsilon, T)$. Recall, then, that $\mathbf{X}^\top\mathbf{X} + V_\ell \preceq 2\mathbf{X}^\top\mathbf{X}$, and so $(\mathbf{X}^\top\mathbf{X} + V_\ell)^{-1} \succeq \frac{1}{2}(\mathbf{X}^\top\mathbf{X})^{-1}$. This means that

$$
\begin{aligned}
\left\|(\mathbf{X}^\top\mathbf{X})^{-1/2}\mathbf{X}^\top E\right\|_2 &= \|\mathbf{X}^\top E\|_{(\mathbf{X}^\top\mathbf{X})^{-1}} \\
&\leq \sqrt{2}\|\mathbf{X}^\top E\|_{(\mathbf{X}^\top\mathbf{X}+V_\ell)^{-1}} \\
&\leq 2\sigma \sqrt{\log\left(\frac{\det\left(\mathbf{X}^\top\mathbf{X} + V_\ell\right)^{1/2} \det\left(V_\ell\right)^{-1/2}}{\delta(\epsilon, T)}\right)} \\
&= 2\sigma \sqrt{\log\left(\frac{\det\left(\mathbf{X}^\top\mathbf{X} V_\ell^{-1} + I_p\right)^{1/2}}{\delta(\epsilon, T)}\right)} \\
&\leq 2\sigma \sqrt{\log\left(\frac{\det\left(V_u V_\ell^{-1} + I_p\right)^{1/2}}{\delta(\epsilon, T)}\right)}.
\end{aligned}
$$

Finally, it is clear that under the conditions in Theorem E.1 we have $\left\|w^\top(\mathbf{X}^\top\mathbf{X})^{-1/2}\right\|_2 \leq \left\|w^\top V_\ell^{-1/2}\right\|_2$. $\qquad\square$

**Corollary E.2.** *Let $\epsilon = 12\frac{p\log(T)}{\tilde{c}\sqrt{T}}$, where $\tilde{c}$ is as defined in Theorem* (E.1). *Then, for $T$ such that* $\frac{\log(T)}{\sqrt{T}} < \frac{\tilde{c}}{24p}\left(\frac{\lambda_{\min}(\Psi)}{\lambda_{\max}(\Gamma)}\right)$ *we have*

$$\mathbb{P}\left(\|\bar{\alpha} - \alpha\|_2 > Cp\sqrt{\frac{1}{T\lambda_{\min}(\Psi)}\log\left(\frac{T\lambda_{\max}(\Psi)}{\lambda_{\min}(\Psi)}\right)}\right) \leq \frac{c}{T^{11}},$$

*Proof.* First, note that with the choice of $\epsilon = 12\frac{p\log(T)}{\tilde{c}\sqrt{T}}$ we have,

$$\delta(\epsilon, T) = c\exp(-\log(T^{12}))$$
$$= \frac{c}{T^{12}} \tag{66}$$

Hence, $-\log(\delta(\epsilon, T)) = C\log(T)$. Further,

$$\left\|w^\top V_\ell^{-1/2}\right\|_2 \leq \left\|V_\ell^{-1/2}\right\|_2$$
$$= \sqrt{\lambda_{\max}(V_\ell^{-1})}$$
$$= \frac{1}{\sqrt{\lambda_{\min}(V_\ell)}}$$
$$\leq \frac{1}{\sigma\sqrt{(T-p)\left(\lambda_{\min}(\Psi) - \epsilon\lambda_{\max}(\Gamma)\right)}}$$
$$\leq \frac{2}{\sigma\sqrt{(T-p)\lambda_{\min}(\Psi)}}, \tag{67}$$

where the last inequality holds with probability $1 - \delta(\epsilon, T)$ using Theorem E.1. Finally, consider

$$\log\left(\det\left(V_u V_\ell^{-1} + I_p\right)^{1/2}\right) = \frac{1}{2}\log\left(\frac{\det\left(V_u + V_\ell\right)}{\det\left(V_\ell\right)}\right)$$
$$= \frac{1}{2}\log\left(\frac{2^p\sigma^{2p}(T-p)^p\det\left(\Psi\right)}{\det\left(V_\ell\right)}\right) \tag{68}$$

Note that, again by our assumed condition on $\epsilon$,

$$\det\left(V_\ell\right) \geq \frac{1}{2}\det\left(\Psi\right)$$
$$\geq \frac{1}{2}\sigma^{2p}(T-p)^p\lambda_{\min}(\Psi)^p \tag{69}$$

Using (69) and (68), we have,

$$\log\left(\det\left(V_u V_\ell^{-1} + I_p\right)^{1/2}\right) \leq \frac{1}{2}\log\left(\frac{2^{p+1}\det\left(\Psi\right)}{\lambda_{\min}(\Psi)^p}\right)$$
$$\leq p\log\left(\frac{4\lambda_{\max}(\Psi)}{\lambda_{\min}(\Psi)}\right) \tag{70}$$

From (70), (67) and (66), we have,

$$2\sigma\left\|w^\top V_\ell^{-1/2}\right\|_2\sqrt{\log\left(\frac{\det\left(V_u V_\ell^{-1} + I_p\right)^{1/2}}{\delta(\epsilon, T)}\right)}$$
$$\leq \sqrt{\frac{Cp}{(T-p)\lambda_{\min}(\Psi)}}\sqrt{\log\left(\frac{4\lambda_{\max}(\Psi)}{\lambda_{\min}(\Psi)}\right) + \log(T)}. \tag{71}$$

Using (71) and Theorem E.2, we get that with probability $1 - \frac{2c}{T^{12}}$

$$|\bar{\alpha}_i(T) - \alpha_i| \leq \sqrt{\frac{Cp}{(T-p)\lambda_{\min}(\Psi)}} \sqrt{\log\left(\frac{4\lambda_{\max}(\Psi)}{\lambda_{\min}(\Psi)}\right) + \log(T)}, \qquad (72)$$

For any $i \in [p]$. This implies,

$$\|\bar{\alpha} - \alpha\|_2 \leq \sqrt{\sum_{i=1}^{p} |\hat{\alpha}_i(T) - \alpha_i|^2} \leq \sqrt{\frac{Cp^2}{(T-p)\lambda_{\min}(\Psi)}} \sqrt{\log\left(\frac{4\lambda_{\max}(\Psi)}{\lambda_{\min}(\Psi)}\right) + \log(T)}, \qquad (73)$$

with probability $1 - \frac{c}{T^{11}}$ which completes the proof. $\qquad\square$

# F  Proof of Theorem 4.3

## F.1  Setup

Recall $\beta^*$ as defined in Proposition 2.2, and its estimate $\widehat{\beta}$ defined as

$$\widehat{\beta} = \mathrm{argmin}_{\beta \in \mathbb{R}^{L-1}} \sum_{m=1}^{N \times T/L} (y_m - \beta^\top \widehat{F}_m)^2. \tag{74}$$

To establish this bound, we first define (and recall) the following notations,

- Recall that $\mathbf{Z}_y = [\mathbf{Z}(y_1, L, T, 1) \quad \mathbf{Z}(y_2, L, T, 1) \quad \dots \quad \mathbf{Z}(y_n, L, T, 1)]$ is the stacked Page matrix of $y_1(t), \dots, y_n(t) \ \forall t \in [T]$.

- Let $\mathbf{Z}_x$ be the stacked Page matrix of $x_1(t), \dots, x_n(t) \ \forall t \in [T]$.

- Let $\mathbf{Z}_f$ be the stacked Page matrix of $f_1(t), \dots, f_n(t) \ \forall t \in [T]$. That is, $\mathbf{Z}_y = \mathbf{Z}_f + \mathbf{Z}_x$.

- Let $\mathbf{Z}'_y \in \mathbb{R}^{(L-1) \times (NT/L)}$, be the sub-matrix obtained by dropping the $L$-th row from the stacked Page matrix $\mathbf{Z}_y$. Define $\mathbf{Z}'_f$ and $\mathbf{Z}'_x$ analogously.

- Let $\mathbf{U\Sigma V}^\top$ denote the SVD of $\mathbf{Z}'_f$.

- let $\mathbf{V}^\perp$ and $\mathbf{U}^\perp$ be matrices of orthonormal basis vectors that span the null space of $\mathbf{Z}'_f$ and ${\mathbf{Z}'_f}^\top$, respectively.

- Let $\widetilde{\mathbf{U}}\widetilde{\mathbf{\Sigma}}\widetilde{\mathbf{V}}^\top$ denote the top k singular components of the SVD of $\mathbf{Z}'_y$, while $\widetilde{\mathbf{U}}^\perp \widetilde{\mathbf{\Sigma}}^\perp (\widetilde{\mathbf{V}}^\perp)^\top$ denote the remaining $L - k - 1$ components such that $\mathbf{Z}'_y = \widetilde{\mathbf{U}}\widetilde{\mathbf{\Sigma}}\widetilde{\mathbf{V}}^\top + \widetilde{\mathbf{U}}^\perp \widetilde{\mathbf{\Sigma}}^\perp (\widetilde{\mathbf{V}}^\perp)^\top$.

- Let $\widehat{\mathbf{Z}}'_f$ be the HSVT estimate of $\mathbf{Z}'_f$ with parameter $k$. That is $\widehat{\mathbf{Z}}'_f = \widetilde{\mathbf{U}}\widetilde{\mathbf{\Sigma}}\widetilde{\mathbf{V}}^\top$.

## F.2  Deterministic Bound

First, given the notation above, the solution for (74), can be written as,

$$\widehat{\beta} = \left(\widehat{\mathbf{Z}}'_f\right)^{\top,\dagger} [\mathbf{Z}_y]_{L\cdot} = \widetilde{\mathbf{U}}\widetilde{\mathbf{\Sigma}}^{-1}\widetilde{\mathbf{V}}^\top [\mathbf{Z}_y]_{L\cdot}. \tag{75}$$

Then, let's consider the following expansion for $\|\widehat{\beta} - \beta^*\|_2$,

$$
\begin{aligned}
\|\widehat{\beta} - \beta^*\|_2^2 &= \|\widetilde{\mathbf{U}}^\perp (\widetilde{\mathbf{U}}^\perp)^\top (\widehat{\beta} - \beta^*) + \widetilde{\mathbf{U}}(\widetilde{\mathbf{U}})^\top (\widehat{\beta} - \beta^*)\|_2^2 \\
&= \|\widetilde{\mathbf{U}}^\perp (\widetilde{\mathbf{U}}^\perp)^\top (\widehat{\beta} - \beta^*)\|_2^2 + \|\widetilde{\mathbf{U}}(\widetilde{\mathbf{U}})^\top (\widehat{\beta} - \beta^*)\|_2^2 \\
&= \|\widetilde{\mathbf{U}}^\perp (\widetilde{\mathbf{U}}^\perp)^\top (\widehat{\beta} - \beta^*)\|_2^2 + \|\widetilde{\mathbf{U}}^\top (\widehat{\beta} - \beta^*)\|_2^2 \\
&= \|\widetilde{\mathbf{U}}^\perp (\widetilde{\mathbf{U}}^\perp)^\top \beta^*\|_2^2 + \|\widetilde{\mathbf{U}}^\top (\widehat{\beta} - \beta^*)\|_2^2,
\end{aligned} \tag{76}
$$

where the last equality follow from the fact that $(\widetilde{\mathbf{U}}^\perp)^\top \widehat{\beta} = \left((\widetilde{\mathbf{U}}^\perp)^\top \widetilde{\mathbf{U}}\right) \widetilde{\mathbf{\Sigma}}^{-1} \widetilde{\mathbf{V}}^\top [\mathbf{Z}_y]_{L\cdot} = 0$. Now, consider the first term in (76), and note that $\beta^* = \mathbf{U}\mathbf{U}^\top \beta^*$,

$$
\begin{aligned}
\|\widetilde{\mathbf{U}}^\perp (\widetilde{\mathbf{U}}^\perp)^\top \beta^*\|_2 &= \left\| \mathbf{U}^\perp (\mathbf{U}^\perp)^\top \mathbf{U}\mathbf{U}^\top \beta^* + \left(\widetilde{\mathbf{U}}^\perp (\widetilde{\mathbf{U}}^\perp)^\top - \mathbf{U}^\perp (\mathbf{U}^\perp)^\top\right) \beta^* \right\|_2 \\
&\leq \left\| \left(\widetilde{\mathbf{U}}^\perp (\widetilde{\mathbf{U}}^\perp)^\top - \mathbf{U}^\perp (\mathbf{U}^\perp)^\top\right) \beta^* \right\|_2 \\
&\leq \left\| \widetilde{\mathbf{U}}^\perp (\widetilde{\mathbf{U}}^\perp)^\top - \mathbf{U}^\perp (\mathbf{U}^\perp)^\top \right\|_2 \|\beta^*\|_2 \\
&= \left\| \widetilde{\mathbf{U}}\widetilde{\mathbf{U}}^\top - \mathbf{U}\mathbf{U}^\top \right\|_2 \|\beta^*\|_2. 
\end{aligned} \tag{77}
$$

Next, by Wedin $\sin\Theta$ Theorem (see [10, 32]) we have:

$$\left\|\widetilde{\mathbf{U}}\widetilde{\mathbf{U}}^\top - \mathbf{U}\mathbf{U}^\top\right\|_2 \|\beta^*\|_2 \leq \frac{\|\mathbf{Z}'_y - \mathbf{Z}'_f\|_2}{\sigma_k(\mathbf{Z}'_f)} \|\beta^*\|_2$$

$$= \frac{\|\mathbf{Z}'_x\|_2}{\sigma_k(\mathbf{Z}'_f)} \|\beta^*\|_2. \tag{78}$$

That is,

$$\|\widetilde{\mathbf{U}}^\perp(\widetilde{\mathbf{U}}^\perp)^\top \beta^*\|_2 \leq \frac{\|\mathbf{Z}'_x\|_2}{\sigma_k(\mathbf{Z}'_f)} \|\beta^*\|_2. \tag{79}$$

What is left is bounding $\|\widetilde{\mathbf{U}}^\top(\widehat{\beta} - \beta^*)\|_2^2$. To that end, first consider

$$\|(\widehat{\mathbf{Z}}'_f)^\top(\widehat{\beta} - \beta^*)\|_2^2 \leq 2\|(\widehat{\mathbf{Z}}'_f)^\top\widehat{\beta} - (\mathbf{Z}'_f)^\top\beta^*\|_2^2 + 2\|(\mathbf{Z}'_f)^\top\beta^* - (\widehat{\mathbf{Z}}'_f)^\top\beta^*\|_2^2$$

$$\leq 2\|(\widehat{\mathbf{Z}}'_f)^\top\widehat{\beta} - (\mathbf{Z}'_f)^\top\beta^*\|_2^2 + 2\|\mathbf{Z}'_f - \widehat{\mathbf{Z}}'_f\|_{2,\infty}^2\|\beta^*\|_1^2. \tag{80}$$

Also, consider

$$\|(\widehat{\mathbf{Z}}'_f)^\top(\widehat{\beta} - \beta^*)\|_2^2 = (\widehat{\beta} - \beta^*)^\top\widetilde{\mathbf{U}}\widetilde{\mathbf{\Sigma}}^2\widetilde{\mathbf{U}}^\top(\widehat{\beta} - \beta^*)$$

$$\geq \sigma_k(\widehat{\mathbf{Z}}'_f)^2\|\widetilde{\mathbf{U}}^\top(\widehat{\beta} - \beta^*)\|_2^2. \tag{81}$$

From (81) and (80) we get,

$$\|\widetilde{\mathbf{U}}^\top(\widehat{\beta} - \beta^*)\|_2^2 \leq \frac{2}{\sigma_k(\widehat{\mathbf{Z}}'_f)^2}\left(\|(\widehat{\mathbf{Z}}'_f)^\top\widehat{\beta} - (\mathbf{Z}'_f)^\top\beta^*\|_2^2 + \|\mathbf{Z}'_f - \widehat{\mathbf{Z}}'_f\|_{2,\infty}^2\|\beta^*\|_1^2\right). \tag{82}$$

First, recall that $[\mathbf{Z}_y]_{L\cdot} = [\mathbf{Z}_f]_{L\cdot} + [\mathbf{Z}_x]_{L\cdot}$, and that by definition, $(\mathbf{Z}'_f)^\top\beta^* = [\mathbf{Z}_f]_{L\cdot}$. Then the term $\|(\widehat{\mathbf{Z}}'_f)^\top\widehat{\beta} - (\mathbf{Z}'_f)^\top\beta^*\|_2^2$ can be bounded as follows. First, consider

$$\|(\widehat{\mathbf{Z}}'_f)^\top\widehat{\beta} - (\mathbf{Z}'_f)^\top\beta^*\|_2^2$$

$$= \|(\widehat{\mathbf{Z}}'_f)^\top\widehat{\beta} - [\mathbf{Z}_y]_{L\cdot}\|_2^2 - \|[\mathbf{Z}_x]_{L\cdot}\|_2^2 + 2\left((\widehat{\mathbf{Z}}'_f)^\top\widehat{\beta} - (\mathbf{Z}'_f)^\top\beta^*\right)^\top[\mathbf{Z}_x]_{L\cdot}$$

$$\leq \|(\widehat{\mathbf{Z}}'_f)^\top\beta^* - [\mathbf{Z}_y]_{L\cdot}\|_2^2 - \|[\mathbf{Z}_x]_{L\cdot}\|_2^2 + 2\left((\widehat{\mathbf{Z}}'_f)^\top\widehat{\beta} - (\mathbf{Z}'_f)^\top\beta^*\right)^\top[\mathbf{Z}_x]_{L\cdot}$$

$$= \|(\widehat{\mathbf{Z}}'_f)^\top\beta^* - [\mathbf{Z}_x]_{L\cdot} - [\mathbf{Z}_f]_{L\cdot}\|_2^2 - \|[\mathbf{Z}_x]_{L\cdot}\|_2^2 + 2\left((\widehat{\mathbf{Z}}'_f)^\top\widehat{\beta} - (\mathbf{Z}'_f)^\top\beta^*\right)^\top[\mathbf{Z}_x]_{L\cdot}$$

$$= \left\|\left(\widehat{\mathbf{Z}}'_f - \mathbf{Z}'_f\right)^\top\beta^* - [\mathbf{Z}_x]_{L\cdot}\right\|_2^2 - \|[\mathbf{Z}_x]_{L\cdot}\|_2^2 + 2\left((\widehat{\mathbf{Z}}'_f)^\top\widehat{\beta} - (\mathbf{Z}'_f)^\top\beta^*\right)^\top[\mathbf{Z}_x]_{L\cdot}$$

$$= \left\|\left(\widehat{\mathbf{Z}}'_f - \mathbf{Z}'_f\right)^\top\beta^*\right\|_2^2 + 2\left((\widehat{\mathbf{Z}}'_f)^\top\widehat{\beta} - (\widehat{\mathbf{Z}}'_f)^\top\beta^*\right)^\top[\mathbf{Z}_x]_{L\cdot} \tag{83}$$

Note that $(\widehat{\mathbf{Z}}'_f)^\top\widehat{\beta} = \widetilde{\mathbf{V}}\widetilde{\mathbf{\Sigma}}\widetilde{\mathbf{U}}^\top\widehat{\beta} = \widetilde{\mathbf{V}}\widetilde{\mathbf{\Sigma}}\widetilde{\mathbf{U}}^\top\widetilde{\mathbf{U}}\widetilde{\mathbf{\Sigma}}^\dagger\widetilde{\mathbf{V}}^\top[\mathbf{Z}_y]_{L\cdot} = \widetilde{\mathbf{V}}\widetilde{\mathbf{V}}^\top[\mathbf{Z}_y]_{L\cdot}$, and $(\widehat{\mathbf{Z}}'_f)^\top\beta^* = \widetilde{\mathbf{V}}\widetilde{\mathbf{\Sigma}}\widetilde{\mathbf{U}}^\top\beta^*$, thus,

$$\left((\widehat{\mathbf{Z}}'_f)^\top\widehat{\beta} - (\widehat{\mathbf{Z}}'_f)^\top\beta^*\right)^\top[\mathbf{Z}_x]_{L\cdot} = [\mathbf{Z}_y]_{L\cdot}^\top\widetilde{\mathbf{V}}\widetilde{\mathbf{V}}^\top[\mathbf{Z}_x]_{L\cdot} - \beta^{*\top}\widetilde{\mathbf{U}}\widetilde{\mathbf{\Sigma}}\widetilde{\mathbf{V}}^\top[\mathbf{Z}_x]_{L\cdot}$$

$$= \left([\mathbf{Z}_y]_{L\cdot}^\top\widetilde{\mathbf{V}} - \beta^{*\top}\widetilde{\mathbf{U}}\widetilde{\mathbf{\Sigma}}\right)\widetilde{\mathbf{V}}^\top[\mathbf{Z}_x]_{L\cdot}$$

$$= \left([\mathbf{Z}_f]_{L\cdot}^\top\widetilde{\mathbf{V}} + [\mathbf{Z}_x]_{L\cdot}^\top\widetilde{\mathbf{V}} - \beta^{*\top}\widetilde{\mathbf{U}}\widetilde{\mathbf{\Sigma}}\right)\widetilde{\mathbf{V}}^\top[\mathbf{Z}_x]_{L\cdot}$$

$$= \left(\beta^{*\top}\mathbf{Z}'_f\widetilde{\mathbf{V}} - \beta^{*\top}\widehat{\mathbf{Z}}'_f\widetilde{\mathbf{V}}\right)\widetilde{\mathbf{V}}^\top[\mathbf{Z}_x]_{L\cdot} + \left\|[\mathbf{Z}_x]_{L\cdot}^\top\widetilde{\mathbf{V}}\right\|_2^2$$

$$\leq \left\|\beta^{*\top}\mathbf{Z}'_f - \beta^{*\top}\widehat{\mathbf{Z}}'_f\right\|_2\left\|\widetilde{\mathbf{V}}^\top[\mathbf{Z}_x]_{L\cdot}\right\|_2 + \left\|[\mathbf{Z}_x]_{L\cdot}^\top\widetilde{\mathbf{V}}\right\|_2^2$$

$$\leq \|\mathbf{Z}'_f - \widehat{\mathbf{Z}}'_f\|_{2,\infty}\|\beta^*\|_1\left\|[\mathbf{Z}_x]_{L\cdot}^\top\widetilde{\mathbf{V}}\right\|_2 + \left\|[\mathbf{Z}_x]_{L\cdot}^\top\widetilde{\mathbf{V}}\right\|_2^2 \tag{84}$$

Finally, from (82) and (83) we get

$$\|\widetilde{\mathbf{U}}^\top(\widehat{\beta} - \beta^*)\|_2^2 \le \frac{2}{\sigma_k(\widehat{\mathbf{Z}}_f')^2}\Bigg(2\|\mathbf{Z}_f' - \widehat{\mathbf{Z}}_f'\|_{2,\infty}^2\|\beta^*\|_1^2$$

$$+ \|\mathbf{Z}_f' - \widehat{\mathbf{Z}}_f'\|_{2,\infty}\|\beta^*\|_1\left\|[\mathbf{Z}_x]_{L\cdot}^\top\widetilde{\mathbf{V}}\right\|_2 + \left\|[\mathbf{Z}_x]_{L\cdot}^\top\widetilde{\mathbf{V}}\right\|_2^2\Bigg)$$

$$\le \frac{4}{\sigma_k(\widehat{\mathbf{Z}}_f')^2}\Bigg(2\|\mathbf{Z}_f' - \widehat{\mathbf{Z}}_f'\|_{2,\infty}^2\|\beta^*\|_1^2 + \left\|[\mathbf{Z}_x]_{L\cdot}^\top\widetilde{\mathbf{V}}\right\|_2^2\Bigg).$$

The term $\left\|[\mathbf{Z}_x]_{L\cdot}^\top\widetilde{\mathbf{V}}\right\|_2^2$ can be further decomposed as

$$\left\|[\mathbf{Z}_x]_{L\cdot}^\top\widetilde{\mathbf{V}}\right\|_2^2 \le 2\left\|[\mathbf{Z}_x]_{L\cdot}^\top(\widetilde{\mathbf{V}}\widetilde{\mathbf{V}}^\top - \mathbf{V}\mathbf{V}^\top)\right\|_2^2 + 2\left\|[\mathbf{Z}_x]_{L\cdot}^\top\mathbf{V}\mathbf{V}^\top\right\|_2^2$$

Using Wedin $\sin\Theta$ Theorem,

$$\left\|\widetilde{\mathbf{V}}\widetilde{\mathbf{V}}^\top - \mathbf{V}\mathbf{V}^\top\right\|_2 \le \frac{\|\mathbf{Z}_x'\|_2}{\sigma_k(\mathbf{Z}_f')}$$

Therefore,

$$\|\widetilde{\mathbf{U}}^\top(\widehat{\beta} - \beta^*)\|_2^2 \le \frac{8}{\sigma_k(\widehat{\mathbf{Z}}_f')^2}\Bigg(\|\mathbf{Z}_f' - \widehat{\mathbf{Z}}_f'\|_{2,\infty}^2\|\beta^*\|_1^2 + \frac{\|\mathbf{Z}_x'\|_2^2}{\sigma_k(\mathbf{Z}_f')^2}\|[\mathbf{Z}_x]_{L\cdot}\|_2^2 + \left\|[\mathbf{Z}_x]_{L\cdot}^\top\mathbf{V}\right\|_2^2\Bigg). \tag{85}$$

From (76), (79), and (85) we have

$$\|\widehat{\beta} - \beta^*\|_2^2 \le \frac{\|\mathbf{Z}_x'\|_2^2}{\sigma_k(\mathbf{Z}_f')^2}\Bigg(\|\beta^*\|_2^2 + 8\frac{\|[\mathbf{Z}_x]_{L\cdot}\|_2^2}{\sigma_k(\widehat{\mathbf{Z}}_f')^2}\Bigg)$$

$$+ \frac{8}{\sigma_k(\widehat{\mathbf{Z}}_f')^2}\Bigg(2\|\mathbf{Z}_f' - \widehat{\mathbf{Z}}_f'\|_{2,\infty}^2\|\beta^*\|_1^2 + \left\|[\mathbf{Z}_x]_{L\cdot}^\top\mathbf{V}\right\|_2^2\Bigg). \tag{86}$$

### F.3 High probability bound

Let $q := NT/L$ be the number of columns in the stacked page matrix $\widehat{\mathbf{Z}}_f'$, and let $C > 0$ be some positive absolute constant. Define

$$E_1' := \Big\{\|\mathbf{Z}_x'\|_2 \le C\sigma_x\sqrt{q}\Big\},$$

$$E_2' := \Big\{\|[\mathbf{Z}_x]_{L\cdot}\|_2 \le C\sigma_x\sqrt{q}\Big\},$$

$$E_3' := \Big\{\|\mathbf{Z}_f' - \widehat{\mathbf{Z}}_f'\|_{2,\infty}^2 \le \frac{C\sigma_x^2 q^2}{\sigma_k(\mathbf{Z}_f)^2}\big(\sigma_x^2 + f_{\max}^2\big) + C\sigma_x^2 k\log(q)\Big\}$$

$$E_4' := \Big\{\left\|[\mathbf{Z}_x]_{L\cdot}^\top\mathbf{V}\right\|_2 \le C\sigma_x\sqrt{k\log(q)}\Big\}.$$

**Lemma F.1.** *For some positive constant $c_1 > 0$ and $C > 0$ large enough in definitions of $E_1'$, $E_2'$, $E_3'$ and $E_4'$,*

$$\mathbb{P}(E_1') \ge 1 - 2e^{-c_1 q},$$

$$\mathbb{P}(E_2') \ge 1 - 2e^{-c_1 q},$$

$$\mathbb{P}(E_3') \ge 1 - \frac{c_1}{(NT)^{11}},$$

$$\mathbb{P}(E_4') \ge 1 - \frac{c_1}{(NT)^{11}}.$$

*Proof.* We bound the probability of events above below.

**Bounding $E_1'$ and $E_2'$.** Note that $\mathbf{Z}_x'$ is a sub-matrix of $\mathbf{Z}_x$, and its operator norm is bounded by that of $\mathbf{Z}_x$. Hence, as the probability is an immediate consequence of Lemma A.2. The second event probability follow from the fact that $\|[\mathbf{Z}_x]_{L\cdot}\|_2 \leq \|\mathbf{Z}_x\|_2$.

**Bounding $E_3'$.** This is immediate from Corollary C.1 and its consequence in (27).

**Bounding $E_4'$.** First, recall that $[\mathbf{Z}_x]_{L\cdot}$ is a $q$-dimensional sub-gaussian vector with variance proxy $\sigma_x^2$. That is, for any unit vector $\mathbf{v} \in \mathcal{S}^{q-1}$, $[\mathbf{Z}_x]_{L\cdot}\mathbf{v} \sim \mathrm{subG}(\sigma_x^2)$. Now, recall that $\mathbf{V} \in \mathbb{R}^{q \times k}$ is a set of $k$ orthonormal vectors $\mathbf{V}_i, i \in [k]$. Let $Z_i = [\mathbf{Z}_x]_{L\cdot}\mathbf{V}_i$. Clearly, $Z_i \sim \mathrm{subG}(\sigma_x^2)$. Therefore, $[\mathbf{Z}_x]_{L\cdot}\mathbf{V}$ is a $k$-dimensional vector with dependent sub-gaussian entries with variance proxy $\sigma_x^2$. Using Lemma H.4, we have

$$\mathbb{P}\Big(\|[\mathbf{Z}_x]_{L\cdot}\mathbf{V}\|_2 > t\Big) \leq c\exp\left(-\frac{t^2}{16k\sigma_x^2}\right).$$

Therefore, for choice of $t = C\sigma_x\sqrt{k\log(q)}$ with large enough constant $C \geq 19$, we have,

$$\mathbb{P}\Big(\|[\mathbf{Z}_x]_{L\cdot}\mathbf{V}\|_2 > C\sigma_x\sqrt{k\log(q)}\Big) \leq \frac{c}{q^{22}}.$$

Recalling that $q > L$, and $q = NT/L$, concludes the proof. $\qquad\square$

Let $E' := E_1' \cap E_2' \cap E_3' \cap E_4'$. Then, under event $E$, and using (86), we have, with probability $1 - \frac{c}{(NT)^{11}}$,

$$\|\widehat{\beta} - \beta^*\|_2^2 \leq \frac{C\sigma_x^2}{\sigma_k(\mathbf{Z}_f')^2}\left(q\|\beta^*\|_2^2 + \frac{q}{\sigma_k(\widehat{\mathbf{Z}}_f')^2}\right)$$

$$+ \frac{C\sigma_x^2}{\sigma_k(\widehat{\mathbf{Z}}_f')^2}\left(\left(\frac{q^2}{\sigma_k(\mathbf{Z}_f)^2}\left(\sigma_x^2 + f_{\max}^2\right) + k\log(q)\right)\max\{\|\beta^*\|_1^2, 1\}\right).$$

Choosing $L = \sqrt{NT}$, and $k = RG$, recalling Assumption 4.1, and using Weyl's lemma (H.3) to bound $|\sigma_k(\widehat{\mathbf{Z}}_f')^2 - \sigma_k(\mathbf{Z}_f')^2|$ we get,

$$\|\widehat{\beta} - \beta^*\|_2^2 \leq \frac{C\gamma^2\sigma_x^2 RG}{NT}\left(\sqrt{NT}\|\beta^*\|_2^2 + RG\log(NT)\right.$$

$$+ \left(\gamma^2 RG\left(\sigma_x^2 + f_{\max}^2\right) + RG\log(NT)\right)\|\beta^*\|_1^2\Bigg)$$

$$\leq \frac{C\gamma^4 f_{\max}^2\sigma_x^4 R^2 G^2\log(NT)}{\sqrt{NT}}\max\{\|\beta^*\|_1^2, 1\},$$

which concludes the proof of Theorem 4.3.

# G   Proof of Theorem 4.4

In this section, we prove an upper bound on the forecasting error,

$$\mathsf{ForErr}(N, T) = \frac{1}{NT} \sum_{n=1}^{N} \sum_{t=T+1}^{2T} (\widehat{y}_n(t) - \bar{y}_n(t))^2,$$

where $\bar{y}_n(t) = \mathbb{E}[y_n(t)|y_n(1), \ldots, y_n(t-1)] = f_n(t) + \bar{x}_n(t)$, where $\bar{x}_n(t) := \mathbb{E}[x_n(t)|x_n(1), \ldots, x_n(t-1)]$. First, recall that for $t > T$,

$$\widehat{y}_n(t) = \widehat{f}_n(t) + \widehat{x}_n(t) = \widehat{\beta}^\top Y_n(t-1) + \widehat{\alpha}_n^\top \widetilde{X}_n(t-1),$$

where $Y_n(t-1) = [y_n(t-(L-1)), \ldots, y_n(t-1)]$, and $\widetilde{X}_n(t-1) = [\widetilde{x}_n(t-p), \ldots, \widetilde{x}_n(t-1)]$. Then, we have,

$$\frac{1}{NT} \sum_{n=1}^{N} \sum_{t=T+1}^{2T} (\widehat{y}_n(t) - y_n(t))^2 \le \frac{2}{NT} \sum_{n=1}^{N} \sum_{t=T+1}^{2T} (\widehat{f}_n(t) - f_n(t))^2 + (\widehat{x}_n(t) - \bar{x}_n(t))^2.$$

Now, we will bound each error term separately.

## G.1   Bounding Forecasting Error of $f$

To bound the forecasting error of $f$, we use the following Lemma.

**Lemma G.1.** *Let the conditions of Theorem 4.1 hold. Then, with probability $1 - \frac{c}{(NT)^{11}}$*

$$\frac{1}{NT} \sum_{n=1}^{N} \sum_{t=T+1}^{2T} \left( \widehat{f}_n(t) - f_n(t) \right)^2 \le C(f_{\max}, \gamma) R^3 G^3 \sigma_x^6 \left( \frac{\max\{\|\beta^*\|_1^2, 1\} \log(NT)}{\sqrt{NT}} \right),$$

*where $c$ is an absolute constant, and $C(f_{\max}, \gamma)$ denotes a constant that depends only (polynomially) on model parameters $f_{\max}, \gamma$.*

*Proof.* The proof of this lemma is similar to that in [3], but we adapt it for our different settings. First, note that, where we assume that $NT/L$ is an integer,

$$\frac{1}{NT} \sum_{n=1}^{N} \sum_{t=T+1}^{2T} \left( \widehat{f}_n(t) - f_n(t) \right)^2 = \frac{1}{L} \sum_{\ell=1}^{L} \frac{L}{NT} \sum_{n=1}^{N} \sum_{m=0}^{T/L-1} \left( \widehat{f}_n(T + mL + \ell) - f_n(T + mL + \ell) \right)^2.$$

In what follows, we provide an upper bound for the inner sum $\frac{L}{NT} \sum_{n=1}^{N} \sum_{m=0}^{T/L-1} \left( \widehat{f}_n(T + mL + \ell) - f_n(T + mL + \ell) \right)^2$ for all $\ell \in [L]$. Next, we show, without loss of generality, the case for $\ell = 1$.

To establish this bound, we first define (and recall) the following notations,

- Recall that $\mathbf{Z}_y = [\mathbf{Z}(y_1, L, T) \quad \mathbf{Z}(y_2, L, T) \quad \ldots \quad \mathbf{Z}(y_n, L, T)]$ is the stacked Page matrix of $y_1(t), \ldots, y_n(t)$ for $t \in [T]$.

- Similarly, let $\mathbf{Z}_{y,out}$ denote the stacked Page matrix of the out-of-sample observations $y_1(t), \ldots, y_n(t)$ for $t \in \{T - L + 2, \ldots, 2T - L + 1\}$, i.e., $\mathbf{Z}_{y,out} \in \mathbb{R}^{L \times NT/L}$.

- Let $\mathbf{Z}_x$ be the stacked Page matrix of $x_1(t), \ldots, x_n(t)$ for $t \in [T]$, and $\mathbf{Z}_{x,out}$ be the stacked Page matrix of $x_1(t), \ldots, x_n(t)$ for $t \in \{T - L + 2, \ldots, 2T - L + 1\}$.

- Similarly, let $\mathbf{Z}_f$ and $\mathbf{Z}_{f,out}$ be defined similarly but for $f_1(t), \ldots, f_n(t)$ .

- Let $\mathbf{Z}'_y \in \mathbb{R}^{(L-1) \times (NT/L)}$, be the a sub-matrix obtained by dropping the $L$-th row from the stacked Page matrix $\mathbf{Z}_y$. Define $\mathbf{Z}'_{y,out}, \mathbf{Z}'_f, \mathbf{Z}'_{f,out}, \mathbf{Z}'_x$, and $\mathbf{Z}'_{x,out}$ analogously.

- Let $\mathbf{U}\mathbf{\Sigma}\mathbf{V}^\top$ and $\mathbf{U}_{out}\mathbf{\Sigma}_{out}\mathbf{V}_{out}{}^\top$ denote the SVD of $\mathbf{Z}'_f$ and $\mathbf{Z}'_{f,out}$ respectively.

- let $\mathbf{V}^\perp$ and $\mathbf{U}^\perp$ be matrices of orthonormal basis vectors that span the null space of $\mathbf{Z}'_f$ and $\mathbf{Z'}_f^\top$, respectively. Define $\mathbf{V}_{out}{}^\perp$ and $\mathbf{U}_{out}^\perp$ similarly for the matrix $\mathbf{Z}'_{f,out}$.

- Let $\widetilde{\mathbf{U}}\widetilde{\mathbf{\Sigma}}\widetilde{\mathbf{V}}^\top$ denote the top $k$ singular components of the SVD of $\mathbf{Z}'_y$, while $\widetilde{\mathbf{U}}^\perp\widetilde{\mathbf{\Sigma}}^\perp(\widetilde{\mathbf{V}}^\perp)^\top$ denote the remaining $L - k - 1$ components such that $\mathbf{Z}'_y = \widetilde{\mathbf{U}}\widetilde{\mathbf{\Sigma}}\widetilde{\mathbf{V}}^\top + \widetilde{\mathbf{U}}^\perp\widetilde{\mathbf{\Sigma}}^\perp(\widetilde{\mathbf{V}}^\perp)^\top$.

- Similarly, Let $\widetilde{\mathbf{U}}_{out}\widetilde{\mathbf{\Sigma}}_{out}\widetilde{\mathbf{V}}_{out}^\top$ denote the top k singular components of the SVD of $\mathbf{Z}'_{y,out}$, while $\widetilde{\mathbf{U}}_{out}^\perp\widetilde{\mathbf{\Sigma}}_{out}^\perp(\widetilde{\mathbf{V}}_{out}^\perp)^\top$ denote the remaining $L - k - 1$ components such that $\mathbf{Z}'_{y,out} = \widetilde{\mathbf{U}}_{out}\widetilde{\mathbf{\Sigma}}_{out}\widetilde{\mathbf{V}}_{out}^\top + \widetilde{\mathbf{U}}_{out}^\perp\widetilde{\mathbf{\Sigma}}_{out}^\perp(\widetilde{\mathbf{V}}_{out}^\perp)^\top$.

- Let $\widehat{\mathbf{Z}}_f = \widetilde{\mathbf{U}}\widetilde{\mathbf{\Sigma}}\widetilde{\mathbf{V}}^\top$ and $\widehat{\mathbf{Z}}_{f,out} = \widetilde{\mathbf{U}}_{out}\widetilde{\mathbf{\Sigma}}_{out}\widetilde{\mathbf{V}}_{out}^\top$.

Recall that we are interested in a high-probability bound for the following out-of-sample prediction error:

$$\frac{L}{NT} \sum_{n=1}^{N} \sum_{m=0}^{T/L-1} \left( \widehat{f}_n(T + mL + 1) - f_n(T + mL + 1) \right)^2. \tag{87}$$

Recall that $\widehat{f}_n(t) = \widehat{\beta}^\top Y_n(t-1)$ and hence we can rewrite (87) as

$$\sum_{n=1}^{N} \sum_{m=0}^{T/L} \left( \widehat{f}_n(T + mL + 1) - f_n(T + mL + 1) \right)^2 = \left\| \mathbf{Z}'_{y,out}{}^\top \widehat{\beta} - [\mathbf{Z}_{f,out}]_{L\cdot}^\top \right\|_2^2 \tag{88}$$

$$= \left\| \mathbf{Z}'_{y,out}{}^\top \widehat{\beta} - \mathbf{Z}'_{f,out}{}^\top \beta^* \right\|_2^2 \tag{89}$$

Next, we derive a deterministic upper found for the expression above.

**Deterministic Bound.** Through adding and subtracting $\widehat{\mathbf{Z}}_{f,out}^\top \widehat{\beta}$ and triangle inequality, we have

$$\left\| (\mathbf{Z}'_{y,out})^\top \widehat{\beta} - (\mathbf{Z}'_{f,out})^\top \beta^* \right\|_2 \leq \left\| (\mathbf{Z}'_{y,out} - \widehat{\mathbf{Z}}_{f,out})^\top \widehat{\beta} \right\|_2 + \left\| \widehat{\mathbf{Z}}_{f,out}^\top \widehat{\beta} - (\mathbf{Z}'_{f,out})^\top \beta^* \right\|_2 \tag{90}$$

Next, we proceed to bound each of the two terms on the right hand side.

*First term:* $\left\| (\mathbf{Z}'_{y,out} - \widehat{\mathbf{Z}}_{f,out})^\top \widehat{\beta} \right\|_2$.

$$\left\| (\mathbf{Z}'_{y,out} - \widehat{\mathbf{Z}}_{f,out})^\top \widehat{\beta} \right\|_2^2 = \| \widetilde{\mathbf{V}}_{out}^\perp \widetilde{\mathbf{\Sigma}}_{out}^\perp (\widetilde{\mathbf{U}}_{out}^\perp)^\top \widehat{\beta} \|_2^2$$

$$\leq \| \widetilde{\mathbf{\Sigma}}_{out}^\perp \|_2^2 \| (\widetilde{\mathbf{U}}_{out}^\perp)^\top \widehat{\beta} \|_2^2. \tag{91}$$

Note that $\| \widetilde{\mathbf{\Sigma}}_{out}^\perp \|_2$ equals the $(k+1)$-th singular value of $\mathbf{Z}'_{y,out}$. Using Weyl's inequality (see Lemma H.3), and recalling that $\mathbf{Z}'_{y,out} = \mathbf{Z}'_{f,out} + \mathbf{Z}'_{x,out}$, and that the rank of $\mathbf{Z}'_{f,out}$ is $k$, we can bound the $(k+1)$-th singular value of $\mathbf{Z}'_{y,out}$ by the operator norm of $\mathbf{Z}'_{x,out}$. That is,

$$\| \widetilde{\mathbf{\Sigma}}_{out}^\perp \|_2^2 \leq \| \mathbf{Z}'_{x,out} \|_2^2. \tag{92}$$

Next, we bound the term $\| (\widetilde{\mathbf{U}}_{out}^\perp)^\top \widehat{\beta} \|_2^2$.

$$\|(\widetilde{\mathbf{U}}_{out}^{\perp})^{\top}\widehat{\beta}\|_2^2 = \|\widetilde{\mathbf{U}}_{out}^{\perp}(\widetilde{\mathbf{U}}_{out}^{\perp})^{\top}\widehat{\beta}\|_2^2$$
$$= \|\widetilde{\mathbf{U}}_{out}^{\perp}(\widetilde{\mathbf{U}}_{out}^{\perp})^{\top}\beta^* + \widetilde{\mathbf{U}}_{out}^{\perp}(\widetilde{\mathbf{U}}_{out}^{\perp})^{\top}(\widehat{\beta} - \beta^*)\|_2^2$$
$$\leq 2\|\widetilde{\mathbf{U}}_{out}^{\perp}(\widetilde{\mathbf{U}}_{out}^{\perp})^{\top}\beta^*\|_2^2 + 2\|\widehat{\beta} - \beta^*\|_2^2. \tag{93}$$

Further expanding the first term,

$$\|\widetilde{\mathbf{U}}_{out}^{\perp}(\widetilde{\mathbf{U}}_{out}^{\perp})^{\top}\beta^*\|_2 = \|\widetilde{\mathbf{U}}_{out}^{\perp}(\widetilde{\mathbf{U}}_{out}^{\perp})^{\top}\mathbf{U}_{out}(\mathbf{U}_{out})^{\top}\beta^*\|_2$$
$$\leq \left\|\mathbf{U}_{out}^{\perp}(\mathbf{U}_{out}^{\perp})^{\top}\mathbf{U}_{out}(\mathbf{U}_{out})^{\top}\beta^*\right\|_2$$
$$+ \left\|\left(\widetilde{\mathbf{U}}_{out}^{\perp}(\widetilde{\mathbf{U}}_{out}^{\perp})^{\top} - \mathbf{U}_{out}^{\perp}(\mathbf{U}_{out}^{\perp})^{\top}\right)\mathbf{U}_{out}(\mathbf{U}_{out})^{\top}\beta^*\right\|_2$$
$$\leq \left\|\left(\widetilde{\mathbf{U}}_{out}^{\perp}(\widetilde{\mathbf{U}}_{out}^{\perp})^{\top} - \mathbf{U}_{out}^{\perp}(\mathbf{U}_{out}^{\perp})^{\top}\right)\beta^*\right\|_2$$
$$\leq \left\|\widetilde{\mathbf{U}}_{out}^{\perp}(\widetilde{\mathbf{U}}_{out}^{\perp})^{\top} - \mathbf{U}_{out}^{\perp}(\mathbf{U}_{out}^{\perp})^{\top}\right\|_2 \|\beta^*\|_2$$
$$= \left\|\widetilde{\mathbf{U}}_{out}\widetilde{\mathbf{U}}_{out}^{\top} - \mathbf{U}_{out}\mathbf{U}_{out}^{\top}\right\|_2 \|\beta^*\|_2.$$

Where in the first equality we use the fact that $\beta^* = \mathbf{U}_{out}(\mathbf{U}_{out})^{\top}\beta^*$, i.e., $\beta^*$ lives in the column space of $\mathbf{Z}'_{f,out}$ (Assumption 4.2).

Next, by Wedin $\sin\Theta$ Theorem (see [10, 32]) we bound $\left\|\widetilde{\mathbf{U}}_{out}\widetilde{\mathbf{U}}_{out}^{\top} - \mathbf{U}_{out}\mathbf{U}_{out}^{\top}\right\|_2$ as follows:

$$\left\|\widetilde{\mathbf{U}}_{out}\widetilde{\mathbf{U}}_{out}^{\top} - \mathbf{U}_{out}\mathbf{U}_{out}^{\top}\right\|_2 \|\beta^*\|_2 \leq \frac{\|\mathbf{Z}_{x,out}\|_2}{\sigma_k(\mathbf{Z}_{f,out})}\|\beta^*\|_2 \tag{94}$$

Using this definition, (91), (92), (93), and (94) we have

$$\left\|(\mathbf{Z}'_{y,out} - \widehat{\mathbf{Z}}_{f,out})^{\top}\widehat{\beta}\right\|_2^2 \leq 2\|\mathbf{Z}'_{x,out}\|_2^2\left(\frac{\|\mathbf{Z}_{x,out}\|_2^2\|\beta^*\|_2^2}{\sigma_k(\mathbf{Z}_{f,out})^2} + \|\widehat{\beta} - \beta^*\|_2^2\right). \tag{95}$$

*Second term:* $\left\|\widehat{\mathbf{Z}}_{f,out}^{\top}\widehat{\beta} - (\mathbf{Z}'_{f,out})^{\top}\beta^*\right\|_2$. To bound the second term, we follow a similar proof to that shown in [5].

$$\|\widehat{\mathbf{Z}}_{f,out}^{\top}\widehat{\beta} - (\mathbf{Z}'_{f,out})^{\top}\beta^*\|_2^2 = \|\widehat{\mathbf{Z}}_{f,out}^{\top}\widehat{\beta} - \widehat{\mathbf{Z}}_{f,out}^{\top}\beta^* + \widehat{\mathbf{Z}}_{f,out}^{\top}\beta^* - (\mathbf{Z}'_{f,out})^{\top}\beta^*\|_2^2$$
$$\leq 2\|\widehat{\mathbf{Z}}_{f,out}^{\top}(\widehat{\beta} - \beta^*)\|_2^2 + 2\|(\mathbf{Z}'_{f,out} - \widehat{\mathbf{Z}}_{f,out}))^{\top}\beta^*\|_2^2$$
$$\leq 2\|\widehat{\mathbf{Z}}_{f,out}^{\top}(\widehat{\beta} - \beta^*)\|_2^2 + 2\|\mathbf{Z}'_{f,out} - \widehat{\mathbf{Z}}'_{f,out}\|_{2,\infty}^2\|\beta^*\|_1^2. \tag{96}$$

Next, we bound the term $\|\widehat{\mathbf{Z}}_{f,out}^{\top}(\widehat{\beta} - \beta^*)\|_2^2$.

$$\|\widehat{\mathbf{Z}}_{f,out}^{\top}(\widehat{\beta} - \beta^*)\|_2^2 \leq \|(\widehat{\mathbf{Z}}_{f,out} - \mathbf{Z}'_{f,out} + \mathbf{Z}'_{f,out})^{\top}(\widehat{\beta} - \beta^*)\|_2^2 \tag{97}$$
$$\leq 2\|(\widehat{\mathbf{Z}}_{f,out} - \mathbf{Z}'_{f,out})^{\top}(\widehat{\beta} - \beta^*)\|_2^2 + 2\|\mathbf{Z}'_{f,out}^{\top}(\widehat{\beta} - \beta^*)\|_2^2 \tag{98}$$
$$\leq 2\|\mathbf{Z}'_{x,out}\|_2^2\|\widehat{\beta} - \beta^*\|_2^2 + 2\|\mathbf{Z}'_{f,out}^{\top}(\widehat{\beta} - \beta^*)\|_2^2. \tag{99}$$

Next, we bound $\|\mathbf{Z}'_{f,out}^{\top}(\widehat{\beta} - \beta^*)\|_2^2$. Recall that $\mathbf{U}$ spans the column space of $\mathbf{Z}'_{f,out}$. Thus $\mathbf{Z}'_{f,out}^{\top} = \mathbf{Z}'_{f,out}^{\top}\mathbf{U}\mathbf{U}^{\top}$, therefore,

$$\|\mathbf{Z}_{f,out}^{\top}(\widehat{\beta} - \beta^*)\|_2^2 = \|\mathbf{Z}'_{f,out}^{\top}\mathbf{U}\mathbf{U}^{\top}(\widehat{\beta} - \beta^*)\|_2^2$$
$$\leq \|\mathbf{Z}'_{f,out}\|_2^2\|\mathbf{U}\mathbf{U}^{\top}(\widehat{\beta} - \beta^*)\|_2^2. \tag{100}$$

Now recall that we denote the top $k$ left singular vectors of $\mathbf{Z}'_y$ by $\widetilde{\mathbf{U}}$, then consider

$$\|\mathbf{U}\mathbf{U}^\top(\widehat{\beta} - \beta^*)\|_2^2 = \|(\mathbf{U}\mathbf{U}^\top + \widetilde{\mathbf{U}}\widetilde{\mathbf{U}}^\top - \widetilde{\mathbf{U}}\widetilde{\mathbf{U}}^\top)(\widehat{\beta} - \beta^*)\|_2^2$$
$$\leq 2\|\mathbf{U}\mathbf{U}^\top - \widetilde{\mathbf{U}}\widetilde{\mathbf{U}}^\top\|_2^2\|\widehat{\beta} - \beta^*\|_2^2 + 2\|\widetilde{\mathbf{U}}\widetilde{\mathbf{U}}^\top(\widehat{\beta} - \beta^*)\|_2^2. \quad (101)$$

Recall that, as we prove in (85) in Appendix F,

$$\|\widetilde{\mathbf{U}}\widetilde{\mathbf{U}}^\top(\widehat{\beta} - \beta^*)\|_2^2 \leq \frac{8}{\sigma_k(\mathbf{Z}'_f)^2}\left(\|\mathbf{Z}'_f - \widehat{\mathbf{Z}}'_f\|_{2,\infty}^2\|\beta^*\|_1^2 + \frac{\|\mathbf{Z}'_x\|_2^2}{\sigma_k(\mathbf{Z}'_f)^2}\|[\mathbf{Z}_x]_{L\cdot}\|_2^2 + \left\|[\mathbf{Z}_x]_{L\cdot}^\top\mathbf{V}\right\|_2^2\right). \quad (102)$$

Using (101), (102) and Wedin $\sin\Theta$ Theorem, we obtain,

$$\|\mathbf{U}\mathbf{U}^\top(\widehat{\beta} - \beta^*)\|_2^2 \leq 2\frac{\|\mathbf{Z}'_x\|_2^2}{\sigma_k(\mathbf{Z}'_f)^2}\left(\|\widehat{\beta} - \beta^*\|_2^2 + \frac{8\|[\mathbf{Z}_x]_{L\cdot}\|_2^2}{\sigma_k(\mathbf{Z}'_f)^2}\right)$$
$$+ \frac{8}{\sigma_k(\widehat{\mathbf{Z}}'_f)^2}\left(\|\mathbf{Z}'_f - \widehat{\mathbf{Z}}'_f\|_{2,\infty}^2\|\beta^*\|_1^2 + \left\|[\mathbf{Z}_x]_{L\cdot}^\top\mathbf{V}\right\|_2^2\right). \quad (103)$$

Therefore, using (96), (100), and (103), we have

$$\|\widehat{\mathbf{Z}}_{f,out}^\top\widehat{\beta} - (\mathbf{Z}'_{f,out})^\top\beta^*\|_2^2$$
$$\leq 2\|\widehat{\mathbf{Z}}_{f,out}^\top(\widehat{\beta} - \beta^*)\|_2^2 + 2\|\mathbf{Z}'_{f,out} - \widehat{\mathbf{Z}}'_{f,out}\|_{2,\infty}^2\|\beta^*\|_1^2$$
$$\leq 4\|\mathbf{Z}'_{x,out}\|_2^2\|\widehat{\beta} - \beta^*\|_2^2$$
$$+ 4\|\mathbf{Z}'_{f,out}\|_2^2\frac{\|\mathbf{Z}'_x\|_2^2}{\sigma_k(\mathbf{Z}'_f)^2}\|\left(\|\widehat{\beta} - \beta^*\|_2^2 + \frac{8\|[\mathbf{Z}_x]_{L\cdot}\|_2^2}{\sigma_k(\mathbf{Z}'_f)^2}\right)$$
$$+ 16\|\mathbf{Z}'_{f,out}\|_2^2\frac{1}{\sigma_k(\widehat{\mathbf{Z}}'_f)^2}\left(2\|\mathbf{Z}'_f - \widehat{\mathbf{Z}}'_f\|_{2,\infty}^2\|\beta^*\|_1^2 + \left\|[\mathbf{Z}_x]_{L\cdot}^\top\mathbf{V}\right\|_2^2\right)$$
$$+ 2\|\mathbf{Z}'_{f,out} - \widehat{\mathbf{Z}}'_{f,out}\|_{2,\infty}^2\|\beta^*\|_1^2. \quad (104)$$

*Combining.* Incorporating the two bounds in (95) and (104) yields,

$$\left\|(\mathbf{Z}'_{y,out})^\top\widehat{\beta} - (\mathbf{Z}'_{f,out})^\top\beta^*\right\|_2^2 \leq c\|\mathbf{Z}'_{x,out}\|_2^2\left(\frac{\|\mathbf{Z}_{x,out}\|_2^2\|\beta^*\|_2^2}{\sigma_k(\mathbf{Z}_{f,out})^2} + \|\widehat{\beta} - \beta^*\|_2^2\right)$$
$$+ c\|\mathbf{Z}'_{f,out}\|_2^2\left(\frac{\|\mathbf{Z}'_x\|_2^2}{\sigma_k(\mathbf{Z}'_f)^2}\|\widehat{\beta} - \beta^*\|_2^2 + \frac{\|\mathbf{Z}'_x\|_2^2\|[\mathbf{Z}_x]_{L\cdot}\|_2^2}{\sigma_k(\mathbf{Z}'_f)^4}\right)$$
$$+ c\frac{\|\mathbf{Z}'_{f,out}\|_2^2}{\sigma_k(\widehat{\mathbf{Z}}'_f)^2}\left(2\|\mathbf{Z}'_f - \widehat{\mathbf{Z}}'_f\|_{2,\infty}^2\|\beta^*\|_1^2 + \left\|[\mathbf{Z}_x]_{L\cdot}^\top\mathbf{V}\right\|_2^2\right).$$
$$+ c\|\mathbf{Z}'_{f,out} - \widehat{\mathbf{Z}}'_{f,out}\|_{2,\infty}^2\|\beta^*\|_1^2. \quad (105)$$

For some absolute constant $c > 0$.

**High probability bound.** With our choice $L = \sqrt{NT}$, Let $q := \sqrt{NT}$ be the number of columns in the stacked page matrix $\mathbf{Z}_y$ and subsequently $q + 1$ to be the number of columns in the stacked page matrix $\mathbf{Z}_{y,out}$. Let $C > 0$ be some positive absolute constant, and $C(f_{\max}, \gamma)$ be a constant that

depends only on model parameters $f_{\max}$ and $\gamma$. Define

$$\bar{E}_1 := \left\{ \|\mathbf{Z}'_x\|_2 \leq C\sigma_x \sqrt{q} \right\}, \tag{106}$$

$$\bar{E}_2 := \left\{ \|\mathbf{Z}'_{x,out}\|_2 \leq C\sigma_x \sqrt{q} \right\}, \tag{107}$$

$$\bar{E}_3 := \left\{ \left\| \widehat{\beta} - \beta^* \right\|_2 \leq C(f_{\max}, \gamma) \left( \frac{\sigma_x^4 G^2 R^2 \log(q)}{q} \right) \max\{\|\beta^*\|_1^2, 1\} \right\}, \tag{108}$$

$$\bar{E}_4 := \left\{ \|\mathbf{Z}'_f - \widehat{\mathbf{Z}}'_f\|_{2,\infty}^2 \leq \frac{C\sigma_x^2 q^2}{\sigma_k(\mathbf{Z}_f)^2} \left( \sigma_x^2 + f_{\max}^2 \right) + C\sigma_x^2 k \log(q) \right\} \tag{109}$$

$$\bar{E}_5 := \left\{ \|\mathbf{Z}'_{f,out} - \widehat{\mathbf{Z}}'_{f,out}\|_{2,\infty}^2 \leq \frac{C\sigma_x^2 q^2}{\sigma_k(\mathbf{Z}_f)^2} \left( \sigma_x^2 + f_{\max} \right) + C\sigma_x^2 k \log(q) \right\} \tag{110}$$

$$\bar{E}_6 := \left\{ \left\| [\mathbf{Z}_x]_{L\cdot}^\top \mathbf{V} \right\|_2 \leq C\sigma_x \sqrt{k \log(q)} \right\}, \tag{111}$$

$$\bar{E}_7 := \left\{ \| [\mathbf{Z}_x]_{L\cdot} \|_2 \leq C\sigma_x \sqrt{q} \right\}, \tag{112}$$

First, with $c_1 > 0$ is an absolute constant, recall from Lemma F.1,

$$\mathbb{P}(\bar{E}_1) \geq 1 - 2e^{-c_1 q},$$

$$\mathbb{P}(\bar{E}_4) \geq 1 - \frac{c_1}{(NT)^{11}},$$

$$\mathbb{P}(\bar{E}_6) \geq 1 - \frac{c_1}{(NT)^{11}},$$

$$\mathbb{P}(\bar{E}_7) \geq 1 - 2e^{-c_1 q},$$

Further, note that $\mathbf{Z}'_{x,out}$ is a sub-matrix of $\mathbf{Z}_{x,out}$, hence its operator norm is bounded by that of $\mathbf{Z}_{x,out}$. Therefore, as an immediate consequence of Lemma A.2,

$$\mathbb{P}(\bar{E}_2) \geq 1 - 2e^{-c_1 q}.$$

The probability of event $\bar{E}_3$ is bounded by Theorem 4.3 as

$$\mathbb{P}(\bar{E}_3) \geq 1 - \frac{c_1}{(NT)^{11}}.$$

Finally, as an immediate results of Corollary C.1 and its consequence in (27), we have

$$\mathbb{P}(\bar{E}_5) \geq 1 - \frac{c_1}{(NT)^{11}}.$$

Now consider the event $\bar{E} := \bar{E}_1 \cap \bar{E}_2 \cap \bar{E}_3 \cap \bar{E}_4 \cap \bar{E}_5 \cap \bar{E}_6 \cap \bar{E}_7$. Then, under event $\bar{E}$, and using (105), we have, with probability $1 - \frac{c}{(NT)^{11}}$,

$$\left\| (\mathbf{Z}'_{y,out})^\top \widehat{\beta} - (\mathbf{Z}'_{f,out})^\top \beta^* \right\|_2^2 \leq C(f_{\max}, \gamma)\sigma_x^2 \left( \frac{\sigma_x^2 q^2 \|\beta^*\|_2^2}{\sigma_k(\mathbf{Z}_{f,out})^2} + \left( \sigma_x^4 G^2 R^2 \log(q) \right) \max\{\|\beta^*\|_1^2, 1\} \right)$$

$$+ C(f_{\max}, \gamma)\|\mathbf{Z}'_{f,out}\|_2^2 \left( \frac{\sigma_x^2}{\sigma_k(\mathbf{Z}'_f)^2} \left( \sigma_x^4 G^2 R^2 \log(q) \right) \max\{\|\beta^*\|_1^2, 1\} \right)$$

$$+ c\|\mathbf{Z}'_{f,out}\|_2^2 \frac{\sigma_x^4 q^2}{\sigma_k(\mathbf{Z}'_f)^4}$$

$$+ c\frac{\|\mathbf{Z}'_{f,out}\|_2^2}{\sigma_k(\widehat{\mathbf{Z}}'_f)^2} \|\beta^*\|_1^2 \left( \frac{C\sigma_x^2 q^2}{\sigma_k(\mathbf{Z}_f)^2} \left( \sigma_x^2 + f_{\max}^2 \right) + C\sigma_x^2 k \log(q) \right)$$

$$+ c\frac{\|\mathbf{Z}'_{f,out}\|_2^2}{\sigma_k(\widehat{\mathbf{Z}}'_f)^2} \sigma_x^2 k \log(q)$$

$$+ c\frac{\sigma_x^2 q^2}{\sigma_k(\mathbf{Z}_f)^2} \left( \sigma_x^2 + f_{\max}^2 \right) \|\beta^*\|_1^2 \tag{113}$$

$$+ c\sigma_x^2 k \log(q)\|\beta^*\|_1^2. \tag{114}$$

Recalling $q = \sqrt{NT}$, $k = RG$, and Assumption 4.1, we get,

$$\left\| (\mathbf{Z}'_{y,out})^\top \widehat{\beta} - (\mathbf{Z}'_{f,out})^\top \beta^* \right\|_2^2 \leq C(f_{\max}, \gamma) R^3 G^3 \sigma_x^6 \max\{\|\beta^*\|_1^2, 1\} \log(NT). \tag{115}$$

Therefore, with probability $1 - c/(NT)^{11}$,

$$\frac{L}{NT} \sum_{n=1}^{N} \sum_{m=0}^{T/L-1} \left( \widehat{f}_n(T + mL + 1) - f_n(T + mL + 1) \right)^2 \tag{116}$$

$$\leq C(f_{\max}, \gamma) R^3 G^3 \sigma_x^6 \left( \frac{\max\{\|\beta^*\|_1^2, 1\} \log(NT)}{\sqrt{NT}} \right). \tag{117}$$

Further, note that the same argument can be used to bound the sum $\frac{L}{NT} \sum_{n=1}^{N} \sum_{m=0}^{T/L-1} \left( \widehat{f}_n(T + mL+\ell) - f_n(T+mL+\ell) \right)^2$ for all $\ell \in [L]$. Thus, with probability $1 - cL/(NT)^{11} > 1 - c/(NT)^{10}$, we have,

$$\frac{2}{NT} \sum_{n=1}^{N} \sum_{t=T+1}^{2T} \left( \widehat{f}_n(t) - f_n(t) \right)^2 \leq C(f_{\max}, \gamma) R^3 G^3 \sigma_x^6 \left( \frac{\max\{\|\beta^*\|_1^2, 1\} \log(NT)}{\sqrt{NT}} \right). \tag{118}$$

$\square$

## G.2  Bounding Forecasting Error of $x$

Now, we bound the error of forecasting $x_n(t)$. First, recall $\widetilde{x}_n(t) = y_n(t) - \widehat{f}_n(t)$, and let $\widetilde{X}_n(t) := [\widetilde{x}_n(t), \ldots, \widetilde{x}_n(t - p + 1)]$ and $X_n(t) := [x_n(t), \ldots, x_n(t - p + 1)]$. Now, let $\boldsymbol{X}_n$, $\widetilde{\boldsymbol{X}}_n$ and $\boldsymbol{\Delta}_n$, all in $\mathbb{R}^{(T-p) \times p}$, be the row-wise concatenations of $X_n(t)$, $\widetilde{X}_n(t)$, and $\Delta_n(t) := X_n(t) - \widehat{X}_n(t)$ for $t \in \{T + p, T + L, \ldots, 2T + p - 1\}$.

Then, we have, where $\bar{x}_n(t) = \mathbb{E}\left[x_n(t) | x_n(1), \ldots, x_n(t-1)\right]$

$$\sum_{t=T+1}^{2T} (\widehat{x}_n(t) - \bar{x}_n(t))^2 = \left\| \widehat{\alpha}_n^\top \widetilde{\boldsymbol{X}}_n^\top - \alpha_n^\top \boldsymbol{X}_n^\top \right\|_2^2 \tag{119}$$

$$= \left\| \widehat{\alpha}_n^\top \widetilde{\boldsymbol{X}}_n^\top - \widehat{\alpha}_n^\top \boldsymbol{X}_n^\top + \widehat{\alpha}_n^\top \boldsymbol{X}_n^\top - \alpha_n^\top \boldsymbol{X}_n^\top \right\|_2^2 \tag{120}$$

$$\leq 2 \left( \left\| \widehat{\alpha}_n^\top \widetilde{\boldsymbol{X}}_n^\top - \widehat{\alpha}_n^\top \boldsymbol{X}_n^\top \right\|_2^2 + \left\| \widehat{\alpha}_n^\top \boldsymbol{X}_n^\top - \alpha_n^\top \boldsymbol{X}_n^\top \right\|_2^2 \right) \tag{121}$$

$$\leq 2 \left( \left\| \widehat{\alpha}_n^\top \widetilde{\boldsymbol{X}}_n^\top - \widehat{\alpha}_n^\top \boldsymbol{X}_n^\top \right\|_2^2 + \left\| \widehat{\alpha}_n^\top - \alpha_n^\top \right\|_2^2 \left\| \boldsymbol{X}_n^\top \right\|_2^2 \right) \tag{122}$$

$$\leq 4 \left( \|\boldsymbol{\Delta}_n\|_2^2 \left( \|\alpha_n\|_2^2 + \|\widehat{\alpha}_n - \alpha_n\|_2^2 \right) + \|\boldsymbol{X}_n\|_2^2 \|\widehat{\alpha}_n - \alpha_n\|_2^2 \right). \tag{123}$$

First note that

$$\sum_{n=1}^{N} \|\boldsymbol{\Delta}_n\|_2^2 \leq \sum_{n=1}^{N} \|\boldsymbol{\Delta}_n\|_F^2 \tag{124}$$

$$\leq p \sum_{n=1}^{N} \sum_{t=T-p+1}^{2T-1} (x_n(t) - \widetilde{x}_n(t))^2 \tag{125}$$

$$= p \sum_{n=1}^{N} \sum_{t=T-p+1}^{2T-1} (f_n(t) - \widehat{f}_n(t))^2. \tag{126}$$

Therefore,

$$\sum_{n=1}^{N} \sum_{t=T+1}^{2T} \left( \widehat{x}_n(t) - \bar{x}_n(t) \right)^2 \leq 4p \max_n \left( \|\alpha_n\|_2^2 + \|\widehat{\alpha}_n - \alpha_n\|_2^2 \right) \sum_{n=1}^{N} \sum_{t=T-p+1}^{2T-1} \left( f_n(t) - \widehat{f}_n(t) \right)^2 \tag{127}$$

$$+ \sum_{n=1}^{N} \|\boldsymbol{X}_n\|_2^2 \|\widehat{\alpha}_n - \alpha_n\|_2^2 \tag{128}$$

**High probability bound.** For the high probability bound, consider the event $\tilde{E} = \tilde{E}_1 \cap \tilde{E}_2 \cap \tilde{E}_3$ where

$$\tilde{E}_1 := \bigcap_{n=1}^{N} \left\{ \|\widehat{\alpha}_n - \alpha_n\|_2^2 \leq C(f_{\max}, \gamma) \frac{GRp}{\lambda_{\min}(\Psi)} \frac{\sigma_x^6}{\sigma^2} \frac{\log(NT)^2}{\sqrt{NT}} \max\left\{1, \frac{1}{\sigma^2 \lambda_{\min}(\Psi)}\right\} + \frac{Cp^2}{T\lambda_{\min}(\Psi)} \log\left(\frac{T\lambda_{\max}(\Psi)}{\lambda_{\min}(\Psi)}\right) \right\} \tag{129}$$

$$\tilde{E}_2 := \left\{ \sum_{n=1}^{N} \sum_{t=T-p+1}^{2T-1} \left( f_n(t) - \widehat{f}_n(t) \right)^2 \leq C(f_{\max}, \gamma) R^3 G^3 \sigma_x^6 \left( \max\{\|\beta^*\|_1^2, 1\} \sqrt{NT} \log(NT) \right) \right\} \tag{130}$$

$$\tilde{E}_3 := \bigcap_{n=1}^{N} \left\{ \|\boldsymbol{X}_n\|_2^2 \leq \frac{3}{2} \sigma^2 (T-p) \lambda_{\max}(\Psi) \right\}, \tag{131}$$

Note that $\mathbb{P}(\tilde{E}_1) \geq 1 - \frac{CN}{T^{11}}$ using Theorems 4.2, 4.1 and the the union bound. Further, note that $\mathbb{P}(\tilde{E}_2) \geq 1 - \frac{C}{(NT)^{10}}$ using both Lemma G.1 and Theorem 4.1. Further, note that according to Corollary E.1, with probability $1 - \frac{c}{T^{11}}$, we have

$$\|\boldsymbol{X}_n\|_2^2 = \lambda_{\max}(\boldsymbol{X}_n^\top \boldsymbol{X}_n) \leq \frac{3}{2} \sigma^2 (T-p) \lambda_{\max}(\Psi). \tag{132}$$

Therefore, under $\tilde{E}$, and recalling $N < T$, we have with probability $1 - \frac{C}{T^{10}}$,

$$\frac{1}{NT} \sum_{n=1}^{N} \sum_{t=T+L}^{2T+L} \left( \widehat{x}_n(t) - x_n(t) \right)^2 \tag{133}$$

$$\leq C(f_{\max}, \gamma) G^3 R^3 p^2 \frac{\lambda_{\max}(\Psi)}{\lambda_{\min}(\Psi)} \sigma_x^6 \max\{\|\beta^*\|_1^2, 1\} \left( \frac{p\sigma^2}{T} \log\left(\frac{T\lambda_{\max}(\Psi)}{\lambda_{\min}(\Psi)}\right) \right. \tag{134}$$

$$+ \frac{GR\alpha_{\max}^2}{\min\left\{1, \sigma^2 \lambda_{\min}(\Psi)\right\}} \frac{\sigma_x^6}{\sigma^2} \frac{\log(NT)^2}{\sqrt{NT}} \right). \tag{135}$$

where $\alpha_{\max} = \max_n \|\alpha_n\|_2$ therefore, with probability $1 - \frac{C}{T^{10}}$, we have

$$\frac{2}{NT} \sum_{n=1}^{N} \sum_{t=T+1}^{2T} \left( \widehat{y}_n(t) - y_n(t) \right)^2 \tag{136}$$

$$\leq C(f_{\max}, \gamma) G^3 R^3 p^2 \frac{\lambda_{\max}(\Psi)}{\lambda_{\min}(\Psi)} \sigma_x^6 \max\{\|\beta^*\|_1^2, 1\} \left( \frac{p\sigma^2}{T} \log\left(\frac{T\lambda_{\max}(\Psi)}{\lambda_{\min}(\Psi)}\right) \right. \tag{137}$$

$$+ \frac{GR \|\alpha_n\|_2^2}{\min\left\{1, \sigma^2 \lambda_{\min}(\Psi)\right\}} \frac{\sigma_x^6}{\sigma^2} \frac{\log(NT)^2}{\sqrt{NT}} \right) \tag{138}$$

which completes the proof.

# H Helper Lemmas and Definitions

**Definition H.1** (Sub-gaussian random variable [30, 31]). *A zero-mean random variable $X \in \mathbb{R}$ is said to be sub-gaussian with variance proxy $\sigma^2$ (denoted as $X \sim subG(\sigma^2)$) if its moment generating function satisfies*

$$\mathbb{E}[\exp(sX)] \leq \exp\left(\frac{\sigma^2 s^2}{2}\right) \forall s \in \mathbb{R}.$$

**Definition H.2** (Sub-exponential random variable [30, 31]). *A zero-mean random variable $X \in \mathbb{R}$ is said to be sub-exponential with parameter $\nu$ (denoted as $X \sim subE(\nu)$) if its moment generating function satisfies*

$$\mathbb{E}[\exp(sX)] \leq \exp\left(\frac{\nu^2 s^2}{2}\right) \forall |s| \leq \frac{1}{\nu}$$

**Lemma H.1** ([31, 30, 6]). *Let $X \sim subG(\sigma_s^2)$ and $Z \sim N(0, \sigma_g^2)$. For $i \in [N]$, let $Y_i \sim subE(\nu_i)$. Then:*

1. $Z \sim subG(\sigma_g^2)$

2. $Z \sim subE(\sigma_g)$

3. $X^2 - \mathbb{E}[X^2] \sim subE(16\sigma_s^2)$

4. $\sum_{i=1}^{N} Y_i \sim subE(\sum_{i=1}^{N} \nu_i)$

   *If $Y_i$ are independent:*

5. $\sum_{i=1}^{N} Y_i \sim subE((\sum_{i=1}^{N} \nu_i^2)^{1/2})$

**Definition H.3** (Sub-gaussian vector). *A random vector $X \in \mathbb{R}^d$ is a sub-gaussian random vector with parameter $\sigma^2$ if*

$$v^T X \sim subG(\sigma_s^2), \forall v \in \mathbb{S}^{d-1}$$

*where $\mathbb{S}^{d-1} = \{x \in \mathbb{R}^d : \|x\| = 1\}$.*

**Lemma H.2** (Norm of sub-gaussian vector). *Let $X \in \mathbb{R}^d$ be a sub-gaussian random vector with parameter $\sigma^2$. Then, with probability at least $1 - \delta$ for $\delta \in (0, 1)$ :*

$$\|X\|_2 \leq 4\sigma\sqrt{d} + 2\sigma\sqrt{\log\left(\frac{1}{\delta}\right)}.$$

*This implies that for any $t > 2\sigma$,*

$$\|X\|_2 \leq t,$$

*with probability at least $1 - \exp(-\frac{t^2}{16d\sigma^2})$.*

**Definition H.4** (Stationary process [26]). *A process $x(t) \in \mathbb{R}$ is said to be stationary if its expectation does not depend on $t$ and its autocovariance function depend only on the time difference. That is,*

$$\mathbb{E}[x(t)] = \mu \qquad\qquad \forall t$$
$$\mathbb{E}([x(t) - \mu](x(t+j) - \mu)] = \gamma(j) \qquad\qquad \forall t \text{ and any } j.$$

**Lemma H.3** (**Weyl's inequality**). *Given $A, B \in \mathbb{R}^{m \times n}$, let $\sigma_i$ and $\widehat{\sigma}_i$ be the $i$-th singular values of $A$ and $B$, respectively, in decreasing order and repeated by multiplicities. Then for all $i \in [m \wedge n]$,*

$$|\sigma_i - \widehat{\sigma}_i| \leq \|A - B\|_2.$$

**Lemma H.4.** *Let $X = [X_1, \ldots, X_n]$ where each $X_i, i \in [n]$ is a sub-gaussian (not necessarily independent) random variable with variance proxy $\sigma^2$ and $\mathbb{E}[X_i^2] = \gamma$. Then with probability at least $1 - c\exp\left(-\frac{t^2}{16n\sigma^2}\right)$,*

$$\|X\|_2 \leq t \tag{139}$$

*Proof.* From Lemma H.1, we can see that $\|X\|_2^2 - n\gamma = \sum_{i=1}^n x_i^2 - n\gamma$ is sub-exponential with parameter $16n\sigma^2$. Let $d := \sum_{i=1}^n x_i^2 - n\gamma$, then

$$\mathbb{P}(d > t') \leq \frac{\mathbb{E}[\exp(sd)]}{\exp(st')} \leq \exp\left(\frac{\nu^2 s^2}{2} - st'\right) \qquad \forall |s| \leq \frac{1}{\nu}. \tag{140}$$

Choosing $t' = t^2 - n\gamma$, $\nu = 16n\sigma^2$, and $s = \frac{1}{16n\sigma^2}$, we get,

$$\mathbb{P}\left(d > t^2 - n\gamma\right) = \mathbb{P}\left(\|X\|_2^2 > t^2\right) \leq \exp\left(\frac{1}{2}\right) \exp\left(-\frac{t^2 - n\gamma}{16n\sigma^2}\right) \tag{141}$$

$$\leq \exp\left(\frac{1}{2}\right) \exp\left(\frac{\gamma}{16\sigma^2}\right) \exp\left(-\frac{t^2}{16n\sigma^2}\right) \tag{142}$$

$$\leq c \exp\left(-\frac{t^2}{16n\sigma^2}\right) \tag{143}$$

where the last inequality follows from the fact the second moment of $x_1$ is bounded by its variance proxy up to a multiplicative constant. □

**Lemma H.5.** *Let* $X_1, \ldots, X_n$ *be i.i.d. sub-exponential random variables with parameter* $\nu$. *Let* $S = \sum_{i=1}^n X_i$. *Then*

$$\mathbb{P}(|S| \leq nt) \geq 1 - c \exp\left(-\frac{t\sqrt{n}}{\nu}\right). \tag{144}$$

*Proof.* Recall that the sum of n *i.i.d.* sub-exponential random variables is also a $x$ sub-exponential random variable with parameter $\nu\sqrt{n}$ (see Lemma H.1). Thus, using the definition of sub-exponential random variable, we have

$$\mathbb{P}(S > nt) \leq \frac{\mathbb{E}[\exp(\lambda S)]}{\exp(\lambda nt)} \leq \exp\left(\frac{n\nu^2\lambda^2}{2} - \lambda nt\right) \qquad \forall |s| \leq \frac{1}{\sqrt{n}\nu}. \tag{145}$$

Choosing $\lambda = \frac{1}{\sqrt{n}\nu}$, we get,

$$\mathbb{P}(S > nt) \leq \exp\left(\frac{1}{2}\right) \exp\left(-\frac{nt}{\sqrt{n}\nu}\right) \tag{146}$$

$$\leq 2 \exp\left(\frac{-t\sqrt{n}}{\nu}\right). \tag{147}$$

Applying the same inequality to $-S$ completes the proof. □

**Lemma H.6.** *[[33]] Let* $A \in \mathbb{R}^{m \times n}$ *and* $B = A + E$. *Then*

$$\left\|B^\dagger - A^\dagger\right\|_2 \leqslant 2\max\left\{\left\|A^\dagger\right\|_2^2, \left\|B^\dagger\right\|_2^2\right\} \|E\|_2,$$

*where* $M^\dagger := (M^\top M)^{-1}M^\top$ *denote the Moore–Penrose inverse of* $M$.

**Lemma H.7.** *[Lemma 4.2 of [27]] Let* $\{\mathcal{F}_t\}_{t \geq 0}$ *be a filtration, and* $\{Z_t\}_{t \geq 1}$ *and* $\{W_t\}_{t \geq 1}$ *be real-valued processes adapted to* $\mathcal{F}_t$ *and* $\mathcal{F}_{t+1}$ *respectively. Moreover, assume* $W_t | \mathcal{F}_t$ *is mean zero and* $\sigma^2$-*sub-gaussian. Then, for any positive real numbers* $\alpha, \beta$ *we have*

$$\mathbb{P}\left(\left\{\sum_{t=1}^T Z_t W_t \geq \alpha\right\} \cap \left\{\sum_{t=1}^T Z_t^2 \leq \beta\right\}\right) \leq \exp\left(-\frac{\alpha^2}{2\sigma^2\beta}\right). \tag{148}$$

