# OpenReview forum: "SAMoSSA:  Multivariate Singular Spectrum Analysis with Stochastic Autoregressive Noise"
_NeurIPS.cc/2023/Conference — NeurIPS 2023 poster_

### Official Review · Reviewer_xDu7 · 2023-07-06

**Soundness:** 4 excellent
**Presentation:** 3 good
**Contribution:** 3 good
**Rating:** 7
**Confidence:** 2

**Summary:**

The authors propose SAMoSSA, an algorithm that combines deterministic trend estimation via mSSA with estimation of an autoregressive component of a time series.  They provide error rates for trend estimation, estimation of the AR coefficients, as well as the prediction error.   In addition, they consider real and simulated examples and show that the proposed method offers very competitive performance.

**Strengths:**

- While the time series literature is vast and well-developed, it does appear that theoretical results that analyze the effect of trend estimation on AR estimates and the prediction error were quite limited.  Detrending a time series and then modeling using ARMA or VAR is a common routine so this is a practically important problem.
- The derived error bounds appear to be novel.
- The paper is well-written.  Assumptions and implications of theorems are for the most part explained well.

**Weaknesses:**

- The authors could do a more thorough literature review when it comes to inference for trends.  For example, there is a statistics literature on testing for trends. See Chen and Wu (2019): Testing for Trends in High-Dimensional Time Series and references therein.
- While it is a nice result, the implications for real-world data are questionable due to the conditions on the trend as well as the assumed autoregressive structure.

**Questions:**

The more general ARMA model is first discussed in the introduction.  What challenges do estimating the MA components pose for the analysis?

**Limitations:**

The authors do acknowledge some limitations with the studied framework, including assuming stationarity of the stochastic component.

---

> ### Author Rebuttal · Authors · 2023-08-07
>
> We thank the reviewer for their constructive feedback and valuable questions. Below we address the specific questions and comments they raise.
>
> > The more general ARMA model is first discussed in the introduction. What challenges do estimating the MA components pose for the analysis?
>
> Thank you for this great question. If an ARMA process is stable, then it can be represented as an infinite MA process whose coefficients are absolutely summable. Hence, we believe that the Non-Stationary deterministic component can be learned (with a similar bound to the one in Theorem 4.1). However, more careful analysis is needed to determine how well we can learn the ARMA parameters and how well we can forecast. This is indeed an interesting venue for future work. We will add a discussion about ARMA in the revision.
>
>
> > While it is a nice result, the implications for real-world data are questionable due to the conditions on the trend as well as the assumed autoregressive structure.
>
> We believe that the model we provide is rich. In terms of the deterministic part, many standard functions that model time series dynamics satisfy assumptions 2.1-2.2 either exactly or approximately.  Indeed,  as [3] shows, these functions include any finite sum of products of harmonics, low-degree polynomials, and exponential functions.
>
> In terms of the stochastic part, it is worth noting that, in theory,  stationary stochastic processes can be approximated by AR processes suggesting that AR processes are also quite rich. However, there are valid concerns about the limitations we set in our model. In future work, we aim to also explore when the stochastic part is:
> 1. an ARMA structure.
> 2. an integrated process (e.g., ARIMA), as we discuss in the limitation
>
> These are indeed great extensions to further increase the richness of the model. That being said, the model we’re considering now is indeed rich. The improvements of SAMoSSA over mSSA and other baselines, as well as the analysis we provide in the rebuttal, strongly suggest the richness and practicality of our model.
>
> We again thank the reviewer for their comments and constructive feedback. We hope the reviewer will take these clarifications into account in their revised scores.

---

> > ### Comment · Reviewer_xDu7 · 2023-08-19
> >
> > Thank you for your response.  It is interesting to hear about the potential extension to ARMA models.  While there are results that state that even non-stationary processes may be approximated well by AR processes with growing order under certain conditions, it is likely that certain analyses with autoregressive models cannot be extended using such arguments.  Nonetheless, I believe that this is a nice contribution and in light of the author's comments during the rebuttal/discussion phase, I will raise my score up one point.

---

### Official Review · Reviewer_ki9S · 2023-07-07

**Soundness:** 4 excellent
**Presentation:** 3 good
**Contribution:** 3 good
**Rating:** 6
**Confidence:** 3

**Summary:**

This paper proposed SAMoSSA, a two-stage procedure that effectively handles mixtures of deterministic nonstationary and stationary AR processes with minimal model assumptions. The authors analyze SAMoSSA’s ability to estimate non-stationary components under stationary AR noise, the error rate of AR system identiﬁcation via OLS under observation errors, and a ﬁnite-sample forecast error analysis.

**Strengths:**

1. This paper is well-written and easy to follow.
2. The theoretical results are solid.

**Weaknesses:**

1. The theoretical contributions are somewhat incremental.
2. The experiment are insufficient.


**Questions:**

1. The biggest difference between this paper and [3] lies in the different noise setting, i.e., this paper replaces the i.i.d noise with stationary AR noise. However, the theoretical derivation in this article is almost the same as it in [3]. What technical difficulties did the authors encounter during the proof process?

2. How limited are  Assumptions 2.1 and 4.2 in practice?

3. In experiment, more baselines are needed.

4. How to verify that the dataset (especially the real dataset) satisfies the assumptions?

---

> ### Author Rebuttal · Authors · 2023-08-07
>
> We thank the reviewer for their positive and constructive feedback. Below we address the specific questions and comments they raise.
>
> > The biggest difference between this paper and [3] lies in the different noise setting, i.e., this paper replaces the i.i.d noise with stationary AR noise. However, the theoretical derivation in this article is almost the same as it in [3]. What technical difficulties did the authors encounter during the proof process?
>
> The different noise settings raised the following three difficulties
> - **Establishing spectral properties of AR noise.** Given that the noise is not i.i.d., both reconstructing and forecasting the deterministic part ($f_n(t)$) using mSSA required establishing an upper bound on the operator norm of the “Page” matrix of AR processes; a result that may be of interest in its own right (see Lemma A.2).
> - **Analysis of parameter identification of AR processes under arbitrary noise.** To establish the out-of-sample forecasting results for the observations $y_n$, we had to establish a finite sample bound for forecasting both $f_n$ and $x_n$ (whereas in the i.i.d. noise setting, $f_n$ was the only goal). This requires learning the AR parameters (system identification) under  arbitrary and bounded observation noise.
> - **Dependence between learned parameters and future noise terms.** The forecasting analysis in [3] benefits from the iid assumption. For example,  the fact that $\hat{\beta}$ is independent of future noise terms $x_n(t), t > T$ is key to cancel a few of the terms when proving the forecasting error theorem.
>
> > How limited are Assumptions 2.1 and 4.2 in practice?
> How to verify that the dataset (especially the real dataset) satisfies the assumptions?
>
> Assumption 2.1 implies that if you construct a matrix M where $M_{ij}  = f_i(j)$, then rank($M$) $\leq R$. We would argue that this assumption is actually common in high dimensional time series analysis (see [1]). Indeed, TRMF, and also mSSA, has been shown to perform very well in practice on a variety of high dimensional multivariate time series which suggests that the assumption is not limiting in practice.
>
> To verify whether the spatio-temporal model we assume (assumptions 2.1 and 2.2) hold in practice, one can use the diagnostic test for the Spatio-Temporal Model is furnished in [2]. This test verifies whether mSSA is likely to succeed based on the spectral properties of the observed matrix.  Specifically, the test measures the effective rank of the Page matrix associated with the multivariate time series with parameter $L \sim \sqrt{NT}$. If the effective rank does not scale much slower than $L$ then mSSA is unlikely to be effective. We will add a reference and discussion of this test in the revised version.
>
> Assumption 4.2 will hold in practice if the underlying deterministic part of the time series (i.e., $f_i(\cdot)$) does not change, assuming that we have observed enough samples to capture the time series complexity. We believe this assumption is both necessary and natural for our settings; Note that typically, to establish generalization error, modern statistical estimators assume the data-generating process to be i.i.d. Herein, and in line with the work in [3], we make a less restrictive assumption and rely on a purely linear algebraic condition that is natural with the proposed model.
>
>
> > In experiment, more baselines are needed.
> As other reviewers correctly point out,  the theoretical characterization of multi-stage learning algorithms in this setup is missing in the literature, and this is the main focus and deliverable of our paper.  That said, based on the collective feedback from reviewers, we added a new baseline to our experiments (DeepAR, see attached pdf), and we will attempt to add TRMF [1] in our revised manuscript. Interestingly, DeepAR only outperforms SAMoSSA in the traffic dataset.
>
> We again thank the reviewer for their comments and constructive feedback. We hope the reviewer will take these clarifications into account in their revised scores.
>
> **References**
>
> [1] Yu, Hsiang-Fu, Nikhil Rao, and Inderjit S. Dhillon. "Temporal regularized matrix factorization for high-dimensional time series prediction." Advances in neural information processing systems 29 (2016).
>
> [2] Agarwal, Anish, Abdullah Alomar, and Devavrat Shah. "On multivariate singular spectrum analysis and its variants." ACM SIGMETRICS Performance Evaluation Review 50.1 (2022): 79-80.
>
> [3] Agarwal, Anish, Devavrat Shah, and Dennis Shen. "On Model Identification and Out-of-Sample Prediction of Principal Component Regression: Applications to Synthetic Controls." arXiv preprint arXiv:2010.14449 (2020).

---

### Official Review · Reviewer_KKGZ · 2023-07-10

**Soundness:** 3 good
**Presentation:** 2 fair
**Contribution:** 4 excellent
**Rating:** 7
**Confidence:** 2

**Summary:**

This paper proposes a two-stage approach based on multivariate Singular Spectrum Analysis (mSSA) to  estimate the non-stationary components in a time series in the presence of a correlated stationary AR noise, which is subsequently estimated from the residual time series. Theoretical results on the performance of the algorithm in this novel setting, are established along with a finite-sample forecasting consistency bound. Empirical results demonstrate significant improvements in forecasting performance due to identification of the AR noise structure, across various benchmark datasets.

**Strengths:**

mSSA allows estimation of non-stationary deterministic components without domain knowledge or fine-tuning, however, cannot handle additive correlated stationary noise. The paper deals with the important problem of estimating non-stationary deterministic components in the presence of correlated stationary noise, through a unified approach via mSSA. Theoretical results for the mSSA are established beyond the i.i.d noise setting (specifically with AR noise). The paper also provides theoretical results on out-of-sample forecasting error for the proposed two-step algorithm. Overall, I found the contribution to be sufficiently novel and of high quality; addressing a crucial gap in the literature.

**Weaknesses:**

Overall, the paper is well-written but is difficult to follow in some parts, and is lacking in conveying the key ideas. For example, presentation of the algorithm (Section 3): The details of the algorithm are presented well however, the main idea behind the approach is missing. A few sentences to convey the big picture behind the steps would have been very helpful.

**Questions:**

The abbreviation SAMoSSA is used in the abstract and introduction without being explicitly defined (I can guess what it is through the title but not sure why the order is reversed). can you please state this explicitly?

Assumption 2.1: Please provide some intuitive reasoning behind this assumption and what you mean by `fundamental' time series.

In Assumption 2.2, L<=T. Why does it become L<=sqrt(T) in line 168 on p.4?

Section 3: The paper mentions that the proposed algorithm builds on the work of A. Agarwal, et al. [3]. It is not clear how the algorithm in Section 3 is different from what exists in the literature -it would be helpful to state this clearly.

**Limitations:**

Yes, limitations of the approach are included in the final section.

---

> ### Author Rebuttal · Authors · 2023-08-07
>
>
> We thank the reviewer for their positive and constructive feedback. Below we address the specific questions and comments they raise.
>
> > ... A few sentences to convey the big picture behind the steps would have been very helpful in the algorithm section.
>
> Thank you, we will revise the algorithm description in the revised manuscript to convey the key ideas.
>
> > The abbreviation SAMoSSA is used in the abstract and introduction without being explicitly defined (I can guess what it is through the title but not sure why the order is reversed). can you please state this explicitly?
>
> Thank you for this question. We think of SAMoSSA more like a pseudo-acronym, and not really an abbreviation. As you may probably surmised, "SAMoSSA" is a combination of the terms "Stochastic", "Autoregressive", and "Multivariate Singular Spectrum Analysis" (MSSA).
>
> > Assumption 2.1: Please provide some intuitive reasoning behind this assumption and what you mean by `fundamental' time series.
>
> Another way to state this assumption is: $f_n \forall n \in [N]$ is such that if you construct a matrix $M$ where $M_{ij}  = f_i(j)$, then rank($M$) $\leq R$. That is, you can factorize the matrix M into “channel/time series” factors and temporal factors. These R temporal factors (in $\mathbb{R}^{R\times T}$) are what we call fundamental time series.   This assumption, while worded differently, is quite common in high dimensional time series analysis (e.g., see [1]).
>
> > In Assumption 2.2, L<=T. Why does it become L<=sqrt(T) in line 168 on p.4?
>
> The $L\leq\sqrt{T}$ condition in the algorithm is used to simplify the exposition of the analysis but is not crucial for the results. More precisely, it's included to guarantee that the number of rows in the Page matrix is fewer than the number of columns.  For consistency, we will fix the wording of Assumption 2.2 to be $L\leq\sqrt{T}$.
>
>
> > The paper mentions that the proposed algorithm builds on the work of A. Agarwal, et al. [3]. It is not clear how the algorithm in Section 3 is different from what exists in the literature -it would be helpful to state this clearly.
>
> We will revise the algorithm section to make this clearer. In brief, the first three steps in our algorithm (see figure 1) are the same as [3], but steps 4 and 5 are different. In particular, SAMoSSA, in addition to imputing and forecasting $f(\cdot)$, also learns the AR process $x(\cdot)$, and exploits that learned structure to forecast $x(t)$ for $t>T$.  This additional step, as illustrated in the empirical experiments, helps increase the forecast accuracy.
>
> We would like to note though that a few technical challenges in the analysis arise with our model and algorithm compared to that of [3]. Specifically,
> - Given that we assume the noise process to be an AR process, most of the analysis presented in [3] breaks down. Specifically,  we now have to (1) establish spectral properties of the Page matrix of the correlated AR noise; a result that may be of interest in its own right (see Lemma A.2);  and (2) establish a forecasting error bound when the learned parameters (i.e. $\hat{\beta}$) and the stochastic terms (x_i’s) are correlated.
> - Further, as we attempt to learn the AR structure in the stochastic component, we had to improve over current results in system identification to accommodate for the case of arbitrary and bounded observation noise (due to the two-stage nature of the algorithm). In that sense, ours can be viewed as a robust generalization of previous work in AR system identification.
>
> We again thank the reviewer for their comments and constructive feedback.
>
> **References**
>
> [1] Yu, Hsiang-Fu, Nikhil Rao, and Inderjit S. Dhillon. "Temporal regularized matrix factorization for high-dimensional time series prediction." Advances in neural information processing systems 29 (2016).

---

### Official Review · Reviewer_s3mh · 2023-07-14

**Soundness:** 3 good
**Presentation:** 3 good
**Contribution:** 3 good
**Rating:** 6
**Confidence:** 4

**Summary:**

This is a comprehensive work on a new extended variant of multivariate Singular Spectrum Analysis (mSSA), which manages to handle time series of deterministic trend/seasonality with AR stationary components, with rigorous theoretical guarantees. The algorithm is a natural extension of the variant of mSSA using the Page matrix. It further provides guarantees on the bound of estimation error of non-stationary components, the bound of AR parameter estimation error, and the bound of out-of-sample forecasting error. And its superior performance is shown in experiments on four datasets.


**Strengths:**

The extension of the algorithm naturally follows by adding the second step of AR estimation. It is impressive that this paper manages to show all these three bounds on estimation/forecasting errors. What is surprising that, for such a two-stage approach, the bound of out-of-sample forecasting error could be attained.


**Weaknesses:**

The numerical experiments could be richer to show the performance boundary of the proposed method, considering it is dealing with a general signal as the non-stationary plus the AR components. The types of non-stationary components may affects the performance; in particular, instead of the complicated one, the simple deterministic components that could be also be captured by (close to marginal stable) AR parameters may raise issues. And the signal-noise-ratio, the intensity of AR process compared to the deterministic, etc. may also be tested in experiments. All these simulation may help us to further understand the strengths and weaknesses of the proposed method. The reason why the reviewer expects more numerical testing comes from the concerns on the power of truncated SVD of Page matrix for approximately splitting the deterministic/non-stationary and the AR components. If this step fails completely, the whole would fail. We like to know its performance boundary and reliability.

Another concern may be taken into account by area chair. This paper is more typical in the fields of econometrics or statistics. The reviewer is not sure if it fits NeurIPS well.


**Questions:**

Besides the questions in the "Weaknesses" part, there could be more deserving the authors' attentions.

1. Is it correct that we consider such a simplified multivariate time series modelling problem that each channel/variable (i.e. y_n) is driven an independent AR process (x_n)? Together with Assumption 2.1, we actually can model each channel signal y_n separately. This also allows the stack of Page matrix for an extension from the univariate to the multivariate.

2. Following the previous question, the analysis for SAMOSSA for multivariate case has nothing essentially different from the univariate case, right? Maybe we missed some technical challenges.

3. In the proof of out-of-sample forecasting error bound, what is the particular issue that our problem has compared to the analysis in [3]?

4. For time-series forecasting, there has been plenty of SOTA methods using deep learning (like Informer, PatchTST, etc.). We are curious about the comparative results. Indeed yours may be less accurate than DL SOTA, while yours has additional explainability. The performance gap might tell the cost for explainable modelling.


**Limitations:**

We do not see any potential negative societal impact of this paper.

---

> ### Author Rebuttal · Authors · 2023-08-07
>
> We thank the reviewer for their constructive feedback and valuable questions. Below we address the specific questions and comments they raise.
>
> > The numerical experiments could be richer, if this step fails completely, the whole would fail. We like to know its performance boundary and reliability.
>
> We agree that it is important to characterize the performance of the algorithm under different settings. However, we maintain that our analytical approach provides a deeper understanding of the algorithm's behavior compared to adding more numerical experiments, which can only cover a limited number of scenarios.
>
> For example, although it can be conveyed more explicitly, the analysis we provide helps characterize the performance in terms of the signal-to-noise ratio. In particular, we capture the noise effect through the dependence on the term $\sigma_x$ in the various bounds; on the other hand, the signal (for which on can use the sum of the first $k=RG$ singular value of the stacked Page matrix of $f$ as a reasonable estimate) is assumed to obey a lower bound as the balanced spectra assumption states. We will add a discussion of how the various bounds can be stated as a function of the signal-to-noise ratio as suggested. Thank you for the great suggestion.
>
> > This paper is more typical in the fields of econometrics or statistics. The reviewer is not sure if it fits NeurIPS well.
>
> While we acknowledge the reviewer's concerns, we respectfully disagree that our paper does not fit NeurIPS. Time series papers, such as the examples cited in references [1-4], have been prominently featured at NeurIPS, demonstrating a range of technical flavors relevant to our work. [1] in particular is a paper of a very similar flavor (about mSSA for change point detection and is of similar technical depth).
>
> > Is it correct that we consider such a simplified multivariate time series modelling problem that each channel/variable (i.e. y_n) is driven an independent AR process (x_n)? Together with Assumption 2.1, we actually can model each channel signal y_n separately. This also allows the stack of Page matrix for an extension from the univariate to the multivariate.
> Following the previous question, the analysis for SAMOSSA for multivariate case has nothing essentially different from the univariate case, right? Maybe we missed some technical challenges.
>
> The analysis of SAMoSSA encompasses studying both the deterministic and stochastic components of the time series.
>
> For the deterministic part, the analysis of the multivariate case has its own challenges compared to that of the univariate case (i.e., SSA) and it characterizes how one can exploit assumption 2.1 to learn across different time series which results in the better scaling (w.r.t. N).
>
> For the stochastic part, given that we assume that each channel is driven by its own stochastic process, the analysis of the AR parameter identification for the multivariate case is the same as the univariate case.  Considering a VAR-like model (or others), where there exists some dependence between the stationary processes $x_1$,..., $x_N$, is indeed an interesting direction of future work.
>
> > In the proof of out-of-sample forecasting error bound, what is the particular issue that our problem has compared to the analysis in [3]?
>
> The difference between our setting and that of [3] is the noise settings, which raises the following difficulties when establishing the results of thm 4.4:
> 1. **Spectral properties of AR noise.** given that the noise is not i.i.d., both reconstructing and forecasting the deterministic part ($f_n$) required establishing an upper bound on the operator norm of the Page matrix of AR processes; a result that may be of interest in its own right (see Lemma A.2).
> 2. **Analysis of parameter identification of AR processes under arbitrary noise.** To establish the out-of-sample forecasting results for the observations $y_n$, we had to establish finite sample bounds for forecasting both $f_n$ and $x_n$ (whereas in the i.i.d. setting, $f_n$ was the only goal). This requires learning the AR parameters (system identification) under arbitrary and bounded observation noise (due to the reconstruction error).
> 3. **Dependence between learned parameters and future noise terms.** The forecasting analysis in [3] benefits from the i.i.d. assumption as learned parameters remain independent of future “noise” terms $x_n$. For example,  the fact that $\hat{\beta}$ is independent of future noise terms $x_n(t), t > T$ is key when proving the forecasting error theorem in [3].
>
> > We are curious about the comparative results. Indeed yours may be less accurate than DL SOTA, while yours has additional explainability. The performance gap might tell the cost for explainable modelling.
>
> Based on the collective feedback from reviewers, we added a new baseline to our experiments (DeepAR, see attached pdf), and we will attempt to add TRMF in our revised manuscript. Interestingly, DeepAR only outperforms SAMoSSA in the traffic dataset.
>
>
> We again thank the reviewer for their comments and constructive feedback. We hope the reviewer will take these clarifications into account in their revised scores.
>
> **References**
>
> [1] Alanqary, Arwa, Abdullah Alomar, and Devavrat Shah. "Change point detection via multivariate singular spectrum analysis." Advances in Neural Information Processing Systems 34 (2021): 23218-23230.
>
> [2] Mi, Xuelong, et al. "BILCO: An Efficient Algorithm for Joint Alignment of Time Series." Advances in Neural Information Processing Systems 35 (2022): 36270-36281.
>
> [3] Liu, Yong, et al. "Non-stationary transformers: Exploring the stationarity in time series forecasting." Advances in Neural Information Processing Systems 35 (2022): 9881-9893.
>
> [4] Wang, Zhiyuan, et al. "Learning latent seasonal-trend representations for time series forecasting." Advances in Neural Information Processing Systems 35 (2022): 38775-38787.

---

> > ### Comment · Reviewer_s3mh · 2023-08-16
> >
> > Thanks for your responses and sound clarifications on the points we concerned! We appreciate your detailed comments on our questions on the difference of yours from the univariate case and [3]. Answers to Q3 are clear and highly appreciated, thanks! And the added experiments are good. We think you have soundly addressed the issues we raised.
> >
> > For the answers to Q1+2, the "stochastic" part is easy to understand, and we agree on it. Excuse us for one more question, we didn't really get the "deterministic" part of your answer to Q1+2, which has been quoted below:
> > > For the deterministic part, the analysis of the multivariate case has its own challenges compared to that of the univariate case (i.e., SSA) and it characterizes how one can exploit assumption 2.1 to learn across different time series which results in the better scaling (w.r.t. N).
> >
> > Could you further tell us what exactly are the "challenges" or what are the new stuffs of math in your presented results or proof? Thanks a lot for your further clarification.

---

> > > ### Author Response · Authors · 2023-08-17
> > >
> > > We again thank the reviewer for their constructive feedback,  and we're pleased to learn that our responses have been found clear and sound.
> > >
> > > > Could you further tell us what exactly are the "challenges" or what are the new stuffs of math in your presented results or proof? Thanks a lot for your further clarification.
> > >
> > > We will address two aspects, to make sure that our contribution is appropriately communicated.
> > >
> > > **First: Under our settings, how does the analysis for estimating the deterministic part in multivariate case differ from the univariate case?**
> > >
> > > In addressing this, it's pivotal to highlight that the first step in the algorithm differs depending on the case in question. Specifically, in the multivariate case, the algorithm constructs a stacked Page matrix of the observations (refer to eq(5)). Conversely, in the univariate case, only a single Page matrix is constructed.
> > >
> > > Given that, the analysis of the two algorithms will naturally be different. For example, to establish that the estimation error bound in the multivariate case, we had to analyze both the rank of the *stacked* Page matrix induced by the deterministic components $f_1, \dots, f_N$ and the spectral properties of the *stacked* Page matrix induced by the stationary noise process $x_1, \dots, x_N$.  This clearly is different from analyzing the spectral properties of the Page matrix of a single component. This careful analysis for the stacked Page matrices results in a better scaling for the estimation error ($1/\sqrt{NT}$ v.s. $1/\sqrt{T}$ if the univariate algorithm is applied for each time series individually).
> > >
> > > **Second: How is our analysis for estimating the deterministic part  different from that of [3]?**
> > >
> > > The analysis in [3] assumes an independent noise process. Here, we assume that the noise is a correlated stationary AR process. To establish the consistency of estimating the deterministic part, we had to establish certain spectral properties of the stacked Page matrix induced by the stationary noise processes we assumed. This is precisely what we have done. The spectral properties of such random matrices with dependent entries should be of interest in its own right.
> > >
> > > We hope these two points address your concern. Please let us know If you have any further questions.

---

### Official Review · Reviewer_Xzdt · 2023-07-25

**Soundness:** 3 good
**Presentation:** 2 fair
**Contribution:** 3 good
**Rating:** 7
**Confidence:** 2

**Summary:**

This paper extends previous work on multivariate Singular Spectrum Analysis (mSSA) to observations with autoregressive (AR) noise. The method constructs a sliding window representation of the target univariate or multivariate time series called the Page matrix and learns the deterministic non-stationary component (with a linear model) from the singular value decomposition of this representation. The model then produces forecasts by first estimating the parameters of a linear model for the deterministic component followed by the AR parameter estimates of the AR model in the second step. The authors provide theoretical bounds for the error estimates under AR noise of this method, the AR model parameters under perturbation, and the out-of-sample forecasting error. Finally, the authors compare the performance on their method with other time series models (including the mSSA method that is the basis for the presented algorithm.

**Strengths:**

Originality

From an algorithmic point of view, the proposed method is a straightforward extension of the previously developed mSSA model to incorporate autoregressive noise and use a two-step model to learn it’s parameters (and the parameters of the deterministic part). Of course, I think the main contribution lies in the theoretical error bound results but I am not familiar enough with the theory to judge the originality on these and will leave this to other reviewers.


Quality

Again, I am not an expert on the underlying theory but so I cannot judge on the correctness of the theoretical results. The presented method is an interesting approach to time series forecasting and the theoretical guarantees are appealing. The empirical evaluation on the synthetic results confirm the established error bounds on that synthetic example and the paper also contains a (albeit brief) empirical evaluation with different baselines.

Clarity

The paper and its contribution are clearly written. However, the paper is very much written from the perspective Singular Spectrum Analysis point-of-view. I would have appreciated more background and discussion of related approaches in classical time series literature (which I will go into detail in Weaknesses).

Significance

Time series methods that perform well in practice and also have provable error bounds are not very common in the recent ML literature that has focused mostly on deep learning models that lack these guarantees. As such, the method presented in this paper is significant, particularly for high-stakes applications where provable error bounds are required.


**Weaknesses:**

The paper is brief on related work, especially on related approaches in classical time series forecasting (STL decomposition, simple exponential smoothing (SES) with drift). I think it would be useful to the reader to contrast the presented method to these approaches. Another class of related models are matrix factorization models that are not discussed either. Again, I think it would be useful to the reader to discuss this class of methods.

I understand that the main focus on the paper is on the derivation of the theoretical results and I appreciate the effort here. However, the quantitative evaluation is very brief and does not give much insight into the performance of the method relative to either closely related models (other than mSSA) or modern deep learning methods.

I would appreciate if the authors would at least compare their method that is conceptually or algorithmically similar. To this end, I would propose to compare against SES with drift and at least one matrix factorization method for forecasting (TRMF for example: https://dl.acm.org/doi/abs/10.5555/3157096.3157191). A comparison with deep learning baselines such as DeepAR or Fedformer (https://arxiv.org/abs/2201.12740) would also be interesting to understand how the method compares to currently used deep learning models. Simple baselines are also missing (seasonal naive for Traffic/Electricity; the naive method proposed in Bergmeir et al., KDD 2022 (https://link.springer.com/article/10.1007/s10618-022-00894-5) for Exchange).

I would also appreciate more insight on the non-stationary deterministic part learned by SAMoSSA. I’m actually wondering on the added benefit of that part on the Electricity and Traffic datasets, which are quite stationary (but this might be up to debate).


**Questions:**

Why is there such a large improvement from SAMoSSA over mSSA for Electricity? I would appreciate any insight on why the proposed method performs much better here.

**Limitations:**

Limitations are briefly discussed (and the discussion is sufficient in my opinion).

---

> ### Author Rebuttal · Authors · 2023-08-07
>
> We thank the reviewer for their detailed and constructive feedback. In the following sections, we address each of the questions and comments they have raised.
>
> > However, the quantitative evaluation is very brief and does not give much insight into the performance of the method relative to either closely related models (other than mSSA) or modern deep learning methods.
>
> As the reviewer correctly points out, the main focus of this paper is to address the missing theoretical characterization of commonly used multi-stage learning algorithms.  That said, based on the collective feedback from reviewers, we added a new baseline to our experiments (DeepAR, see attached pdf), and we will attempt to add TRMF in our revised manuscript.  Interestingly, DeepAR only outperforms SAMoSSA in the traffic dataset.
>
> > Why is there such a large improvement from SAMoSSA over mSSA for Electricity? I would appreciate any insight on why the proposed method performs much better here. I would also appreciate more insight on the non-stationary deterministic part learned by SAMoSSA. I’m actually wondering on the added benefit of that part on the Electricity and Traffic datasets, which are quite stationary (but this might be up to debate).
>
> We thank the reviewer for this great question, as it gives us the chance to clarify our results and highlight the advantage of our proposed method. To explain the improvement, let us first describe how the two forecast estimates (mSSA vs. SAMoSSA) are different. In mSSA, assuming the univariate case, the forecast estimate is
> $$\hat{y}(t+1) = \hat{f}(t+1)  =  \sum_i \hat{\beta}_i y(t+1-i)  $$
>
> Whereas in SAMoSSA, the forecast estimate is (using the same $\hat{\beta}$ above)
> $$ \hat{y}(t+1) = \hat{f}(t+1) +    \hat{x}(t+1)  =  \sum_i \hat{\beta}_i y(t+1-i)   + \sum_i \alpha_i  \tilde{x}_1(t+1-i) $$
>
> That is, mSSA as described in [3] overlooks any potential structure in the stochastic process $x(\cdot)$ as it assumes i.i.d mean-zero noise process, while in SAMoSSA the structure of $x(\cdot)$ is captured through the learned AR process. Given that, the difference in performance would be attributed to the structure in the (estimated) stochastic process $\hat{x}_(\cdot)$. That is, if there is an AR structure in $ \hat{x}(\cdot) = y(\cdot) - \hat{f}(\cdot)$, then we expect SAMoSSA to perform better.
>
>
> Interestingly, we indeed find this to be the case -- in the electricity dataset, the partial autocorrelation coefficient of  $\hat{x}(\cdot)$ at lag $1$ is significant (we focus on lag $=1$ for brevity and since we find it the highest coefficient on average). In particular, Figure 1 in the attached PDF, shows that on average (across the different univariate time series in the electricity dataset), the partial autocorrelation coefficient at lag $1$ equals $0.2$.
>
> We see a similar but weaker partial autocorrelation in the traffic datasets, with an average partial autocorrelation coefficient (at lag 1) of $0.1$. This could also explain why the improvement in the traffic dataset is relatively smaller.
>
>
> > The paper is brief on related work, especially on related approaches in classical time series forecasting (STL decomposition, simple exponential smoothing (SES) with drift). I think it would be useful to the reader to contrast the presented method to these approaches. Another class of related models are matrix factorization models that are not discussed either. Again, I think it would be useful to the reader to discuss this class of methods.
>
> We thank the reviewer for their great suggestions. We agree that the related work should discuss both topics and we will add them in the revised version.
>
>
> We again thank the reviewer for their comments and constructive feedback. We hope the reviewer will take these clarifications into account in their revised scores.

---

> > ### Comment · Reviewer_Xzdt · 2023-08-17
> >
> > I would like to thank the authors for their response, additional experiments and additional discussion of the related work in the revised manuscript. I am increasing my score. I still would like to ask the authors to consider SES with drift as a baseline in the revised paper.

---

> > > ### Author Response · Authors · 2023-08-17
> > >
> > > Thank you for acknowledging our efforts and the constructive suggestions. We recognize the potential value of SES with drift as a baseline. We will duly consider your suggestion as we finalize our paper.

---

### Official Review · Reviewer_rVQz · 2023-07-26

**Soundness:** 3 good
**Presentation:** 3 good
**Contribution:** 3 good
**Rating:** 6
**Confidence:** 3

**Summary:**

The paper discusses a two-stage algorithm for time series analysis, which involves estimating deterministic, non-stationary trend and seasonality components, followed by learning the residual stochastic, stationary components. The first stage involves using multivariate Singular Spectrum Analysis (mSSA) to estimate the non-stationary components, even in the presence of a correlated stationary Autoregressive (AR) component. The AR component is then learned from the residual time series in the second stage. The authors provide a finite-sample forecasting consistency bound for SAMoSSA, which is data-driven and requires minimal parameter tuning. The paper also presents empirical studies that validate the superior performance of SAMoSSA compared to existing baselines. Notably, SAMoSSA's ability to account for AR noise structure yields improvements ranging from 5% to 37% across various benchmark datasets. The authors also provide a detailed explanation of the model and assumptions used in their analysis, including the spatial and temporal structure of the time series and the properties of the AR processes. They then describe the algorithm for the univariate case and explain how it decomposes the observations into estimates of the non-stationary and stationary components.

**Strengths:**

1.	This paper assesses the importance and novelty of the proposed SAMoSSA methodology and discusses the paper's originality in the use of mSSA and AR processes to jointly learn the deterministic non-stationary and stationary stochastic components of time series.
2.	The paper addresses a gap in the literature, namely the theoretical underpinning of multi-stage learning algorithms involving deterministic and stationary components.
3.	The results demonstrate significant performance improvements over existing baseline approaches.


**Weaknesses:**

1.	The complexity of the paper could be a potential barrier, making it difficult for people without theoretical interests in the area of time series analysis to understand. The paper could provide clearer intuitions and simplifications alongside the complex definitions and theorems to make it more approachable.
2.	The empirical evaluation could be slightly limited. The variety of benchmark baselines used to demonstrate the efficacy of the method could be extended. The analysis of the results should be discussed more carefully, providing insight into why SAMoSSA outperforms other models. It is unclear whether the assumptions made in the paper hold in some of the typical practical scenarios like the electricity and health datasets.
3.	The paper should discuss in more depth where these assumptions come from and under what conditions they might not hold. It should also explore how the method could perform if some of the assumptions are violated. The method proposed seems tailored for a specific kind of problem and it is unclear how generalizable it is.


**Questions:**

The authors should provide simpler explanations or visual aids alongside the more complex mathematical definitions and proofs to make the paper more accessible. More extensive empirical evaluation should be carried out with a broader array of datasets to prove the efficacy of the method. The impact of the violation of some assumptions should be discussed, indicating potential limitations of the proposed method. The authors should provide more discussion on how this method could be generalized or modified to tackle different types of time series.

**Limitations:**

The authors discuss but do not highlight the potential limitations of this work.

---

> ### Author Rebuttal · Authors · 2023-08-07
>
> We express our gratitude to the reviewer for their positive and insightful feedback. In what follows, we address the specific questions and comments raised.
>
> > The authors should provide simpler explanations or visual aids alongside the more complex mathematical definitions and proofs to make the paper more accessible.
>
> Thank you for the feedback. While we have included some discussion and aimed to provide intuition for the various theorems/definitions, we will revise them to make them more approachable. Can the reviewer kindly help us identify particular definitions/theorems that were particularly complex and inaccessible?  Thank you again!.
>
> > More extensive empirical evaluation should be carried out.
>
> As the reviewer correctly points out,  the theoretical characterization of multi-stage learning algorithms in this setup is missing in the literature, and this is the main focus and deliverable of our paper.  That said, based on the collective feedback from reviewers, we added a new baseline to our experiments (DeepAR, see attached pdf), and we will attempt to add TRMF [1] in our revised manuscript. Interestingly, DeepAR only outperforms SAMoSSA in the traffic dataset.
>
>
> > The impact of the violation of some assumptions should be discussed, indicating potential limitations of the proposed method.
>
> We thank the reviewer for raising this important point. To make our contributions more practical, we will add a discussion of a diagnostic test (developed in [2]) that can verify whether the main assumptions we make are valid. This will hopefully indicate the limits of the proposed method and allow practitioners to know when to use or not use the method. The test verifies whether mSSA is likely to succeed, given the observed data.  In particular, the test measures the effective rank of the Page matrix associated with the multivariate time series with parameter $L$ ~ $\sqrt{NT}$. If the effective rank does not scale much slower than L then mSSA is unlikely to be effective.. We will add a reference and discussion of this test in the revised version.
>
>
>
> > The authors should provide more discussion on how this method could be generalized or modified to tackle different types of time series.
>
> In the limitation section, we identify two areas of future work that can generalize the current method. The first is extending the model to include non-stationary stochastic processes (e.g., integrated processes). The second is considering a VAR-like model where there exists some dependence between the stationary processes $x_1$,..., $x_N$.
>
>
> We again thank the reviewer for their comments, constructive feedback and suggestions. We hope the reviewer will take these clarifications into account in their revised scores.
>
> **References**
>
> [1] Yu, Hsiang-Fu, Nikhil Rao, and Inderjit S. Dhillon. "Temporal regularized matrix factorization for high-dimensional time series prediction." Advances in neural information processing systems 29 (2016).
>
> [2] Agarwal, Anish, Abdullah Alomar, and Devavrat Shah. "On multivariate singular spectrum analysis and its variants." ACM SIGMETRICS Performance Evaluation Review 50.1 (2022): 79-80.

---

> > ### Comment · Reviewer_rVQz · 2023-08-18
> >
> > I appreciate the authors' efforts to address most of my concerns. Based on the current version of the work as well as the discussion from other reviewers, I would like to keep my score.

---

### Author Rebuttal · Authors · 2023-08-07

We thank all the reviewers for constructive feedback. Here's a succinct highlight of our paper's key contributions, beyond our individual responses to reviewers:

The main contribution of this paper is to showcase the effectiveness of a simple multi-stage algorithm in time series forecasting. For which we do the following:
1. To establish its effectiveness in theory, we had to extend prior work in the following ways:
- We extend the analysis of mSSA in [3] to accommodate the case of correlated autoregressive noise, a prevalent noise structure in time series analysis. In particular, we establish that one can effectively estimate and forecast the *deterministic* part of the signal under correlated autoregressive noise.
- We extend the analysis of AR parameter estimation in [14] to accommodate the case when the process is observed under arbitrary bounded noise. Our results can be thought of as a robust generalization of [14] which derives similar results but without any observation noise. Thus, we establish that one can effectively forecast the *stochastic* part of the signal.

That is, we overcame two key challenges in analyzing the proposed multi-stage algorithm, significantly building upon recent prior work..

2. We showcase that empirically, through limited but representative datasets and baselines, that SAMoSSA outperforms mSSA. This showcases the effectiveness of the multi-stage process we propose and that the model we proposed is indeed reasonable (see Figure 1 in the rebuttal for further evidence). We believe that further extensions (e.g., considering VAR-like process for the stochastic component) can help achieve even better performances -- which we will consider in future work.

---

### Decision · Program_Chairs · 2023-09-21

**Decision:**

Accept (poster)

**Comment:**

This paper is about multivariate time series analysis. The authors consider N-dimensional time series y=(y_n)_{n=1}^N where each coordinate is modelled as a sum of a non-stationary deterministic component (f_n) and an AR(p_n) process x_n, as written in (1)-(2) with Assumption 2.1-2.4, 4.1-4.2. The goal is to estimate f_n, the parameters of x_n and to forecast the observation y_n. To address these challenges they design a novel two-stage algorithm based on multivariate singular spectrum analysis (mSSA). The resulting SAMoSSA technique is accompanied with finite-sample l_2 estimation error guarantee of the non-stationary component (Theorem 4.1) and the AR parameters (Theorem 4.2), and that of the forecasting (Theorem 4.4). Numerical results on both toy and 3 real-world datasets compared to existing baselines underpin the efficiency of the method.

Time series are omnipresent in data science. Developing principled techniques to analyze them is of fundamental interest to the machine learning community. The authors provide new valuable tools in this context, in a nicely written and well-organized manuscript.